# Sympathetic axonal sprouting induces changes in macrophage populations and protects against pancreatic cancer

Jérémy Guillot[1,12], Chloé Dominici[1,12], Adrien Lucchesi[1,12], Huyen Thi Trang Nguyen [1,2,12], Angélique Puget[1], Mélanie Hocine[1], Martha M. Rangel-Sosa [1], Milesa Simic[3], Jérémy Nigri [4], Fabienne Guillaumond [4], Martin Bigonnet[4], Nelson Dusetti[4], Jimmy Perrot[5], Jonathan Lopez[6,7,8], Anders Etzerodt [3,9], Toby Lawrence[3], Pierre Pudlo[10], Florence Hubert[10], Jean-Yves Scoazec [11], Serge A. van de Pavert [3], Richard Tomasini[4], Sophie Chauvet[1,13] & Fanny Mann [1,13✉]

Neuronal nerve processes in the tumor microenvironment were highlighted recently. However, the origin of intra-tumoral nerves remains poorly known, in part because of technical difficulties in tracing nerve fibers via conventional histological preparations. Here, we employ three-dimensional (3D) imaging of cleared tissues for a comprehensive analysis of sympathetic innervation in a murine model of pancreatic ductal adenocarcinoma (PDAC). Our results support two independent, but coexisting, mechanisms: passive engulfment of pre-existing sympathetic nerves within tumors plus an active, localized sprouting of axon terminals into non-neoplastic lesions and tumor periphery. Ablation of the innervating sympathetic nerves increases tumor growth and spread. This effect is explained by the observation that sympathectomy increases intratumoral CD163+ macrophage numbers, which contribute to the worse outcome. Altogether, our findings provide insights into the mechanisms by which the sympathetic nervous system exerts cancer-protective properties in a mouse model of PDAC.

[1] Aix Marseille Univ, CNRS, IBDM, Marseille, France. [2] University of Science and Technology of Hanoi (USTH), VAST, 18 Hoang Quoc Viet, Hanoi, Vietnam. [3] Aix Marseille Univ, CNRS, INSERM, CIML, Marseille, France. [4] Aix Marseille Univ, CNRS, INSERM, Institut Paoli-Calmettes, CRCM, Marseille, France. [5] Department of Anatomopathology, Lyon Sud University Hospital, Hospices Civils de Lyon, Lyon, France. [6] Department of Biochemistry and Molecular Biology, Lyon Sud University Hospital, Hospices Civils de Lyon, Lyon, France. [7] Faculty of Medicine Lyon-Est, Lyon 1 University, Université de Lyon, Lyon, France. [8] Cancer Research Center of Lyon, INSERM U1052, CNRS UMR5286, Lyon, France. [9] Department of Biomedecine, Aarhus University, Aarhus, Denmark. [10] Aix Marseille Univ, CNRS, Centrale Marseille, I2M, Marseille, France. [11] Department of Pathology, Gustave Roussy Cancer Campus, Villejuif, France. [12] These authors contributed equally: Jérémy Guillot, Chloé Dominici, Adrien Lucchesi, Huyen Thi Trang Nguyen. [13] These authors jointly supervised this work: Sophie Chauvet, Fanny Mann. ✉email: fanny.mann@univ-amu.fr

The peripheral nervous system (PNS) includes a large network of nerves that relays information back and forth between the brain and the body. Typically, a peripheral nerve contains thousands of nerve fibers, or axons, wrapped in bundles that defasciculate into individual axons and form complex branched networks within their target peripheral organ. Axons from the autonomic division of the PNS innervate and regulate most internal organs, glands, and blood vessels to maintain body homeostasis under basal and stress conditions. The autonomic nervous system also plays crucial roles in the regeneration process that restores organ structure and function after damage, leading to the concept of "nerve dependence" in tissue regeneration[1]. Recent insights have revealed a role for the autonomic nervous system in promoting tumorigenesis, which can sometimes be considered an uncontrolled tissue regeneration process[2]. The autonomic nervous system is divided into the sympathetic and parasympathetic nervous systems, whose effects on cancers depend on the type and stage. In prostate cancer, experimental ablation of adrenergic sympathetic nerves inhibits tumor initiation, whereas blocking cholinergic activity of parasympathetic nerves inhibits metastatic spreading at advanced stages of the disease[3]. The similar pro-tumor activity of the autonomic nervous system has been reported in gastric cancer, where parasympathetic nerve fibers play a main role in promoting tumor initiation and progression[4]. This has led to the notion that many cancers depend on nerves for development.

Pancreatic ductal adenocarcinoma (PDAC) is a cancer with a poor prognosis that develops from the exocrine part of the pancreas. Both parasympathetic and sympathetic nervous systems densely innervate the pancreas and regulate exocrine secretions of acinar and ductal cells[5]. The role of the autonomic nervous system in PDAC development has been addressed in recent studies that have revealed a more complex and unexpected picture when compared to other cancer models. Indeed, the interruption of parasympathetic innervation or activity strongly promotes PDAC progression in Kras-mutated mouse models, revealing an antitumor effect of parasympathetic cholinergic signaling[6,7]. The role of the sympathetic nervous system in PDAC has been studied in relation to stress. It was reported that chronic restraint stress promotes Kras-induced pancreatic tumorigenesis through the elevation of circulating adrenal-derived catecholamines (epinephrine and norepinephrine) and stimulation of β2 adrenergic receptor-dependent pancreatic epithelial growth[8]. This study reported that surgical removal of the celiac ganglionic plexus, that provides sympathetic efferent inputs to the pancreas, extended the survival of mice with established PDAC[8]. However, the surgical procedure may also have reduced pancreatic input from sensory neurons[9], which were shown to increase the growth of Kras-mutant spheres and stimulate PDAC initiation and progression[10–12]. Thus, although it has been proposed that noradrenergic signaling of pancreatic sympathetic nerves contributes to PDAC growth, this has yet to be formally demonstrated. This is particularly important because conflicting results from a study based on selective chemical sympathectomy proposed an opposite role for sympathetic nerve activity in mediating the antitumor effect of positive stress induced by enriched housing conditions in murine pancreatic cancer[13].

PNS control over tumorigenesis relies on its ability to innervate developing tumors, which are known to express large numbers of neurotrophic factors and axon guidance molecules[14]. PDAC in particular is a cancer in which significant neuroplastic changes have long been described and which exhibits large nerve bundles in sections of resected tumors from human patients[15,16]. Large nerves are also observed in murine PDAC tumors and an increased intratumoral density of sympathetic nerves has been reported in enlarged tumors of Kras-mutant mice subjected to chronic stress conditions[8,10]. Stress-mediated upregulation of nerve growth factor (NGF) secretion by tumor cells was proposed to attract sympathetic nerve growth via "tumor axonogenesis"[8]. However, a recent study proposed that axons innervating tumors may extend from new neurons generated by populations of precursor cells in the subventricular zone of the brain that would have to travel to the distant tumor through the bloodstream[17]. Such a mechanism, initially described as contributing to the innervation of prostate cancer, has been proposed to occur in other cancer types[17]. Nonetheless, the relative contribution of axonogenesis (i.e., the growth of axons from existing neurons) and neurogenesis (i.e., the de novo generation of neuronal cells) to neuroplastic changes accompanying the development and progression of PDAC remains to be explored.

In this study, we employ 3D imaging of optically cleared tissues to visualize sympathetic nerves and their terminal innervation in the mouse pancreas. This approach allows us to accurately quantify changes in sympathetic innervation patterns during PDAC progression and trace the origin of intratumoral nerves. Our findings disagree with the notion of newborn neuron integration in PDAC tumors and instead support two independent, but coexisting, mechanisms: passive engulfment of sympathetic nerves inside the tumor and active, localized sprouting of axon terminals into nonneoplastic lesions and tumor periphery. In addition, we examine how sympathectomy of pancreatic tumors affects disease progression and find that a lack of sympathetic influence promotes PDAC development and metastatic spread and decreases survival. This worst outcome results from infiltration of pro-tumoral CD163+ macrophages into the denervated tumors. Altogether, our findings suggest that, in contrast to the classical pro-tumor activity reported in other organs, plastic remodeling of sympathetic innervation constitutes a protective response against tumorigenesis in the pancreas.

## Results

**Preexisting sympathetic nerves become engulfed in pancreatic tumors**. To study how the structure and distribution of sympathetic nerves change during PDAC development, we employed tyrosine hydroxylase (TH; an enzyme involved in biosynthesis of norepinephrine) antibody staining and optical tissue clearing to visualize adrenergic innervation of the whole murine pancreas by 3D light-sheet fluorescent microscopy (LSFM). TH is expressed by all postganglionic sympathetic neurons of the celiac-superior mesenteric ganglion complex that supply the pancreas (Fig. 1a). Although the enzyme is also present in some sensory neurons of the dorsal root ganglia (DRG) that innervate the skin and pelvic viscera[18], retrograde tracing experiments indicated that no sensory afferent inputs to the pancreas expressed TH (Fig. 1b, c). TH-positive postganglionic sympathetic nerves emerged from the coeliac-superior mesenteric ganglion complex that entered the head of the pancreas through a main entry point (Fig. 1d), then divided into nerve fiber bundles (typically about 150–100 μm diameter) and extend along the major arteries supplying the pancreas, as previously reported[19,20] (Fig. 1e, f and Supplementary Movie 1). The distribution of sympathetic nerve trunks was quantified on equidistant optical sections through the entire pancreas. The results confirmed a nonhomogeneous distribution, with a high nerve density in the pancreatic head that decreases toward the tail region (Fig. 1g–j).

We next investigated sympathetic innervation in the LSL-Kras^{G12D/+}; Cdkn2a (Ink4a/Arf)^{lox/lox}; Pdx1-Cre (KIC) mouse model of pancreatic cancer. By 8 weeks of age, all KIC mice have developed locally invasive PDAC[21]. At this stage, the sympathetic nerve bundles diverging from the main entry point could be recognized as in controls, but the normal pattern of pancreatic

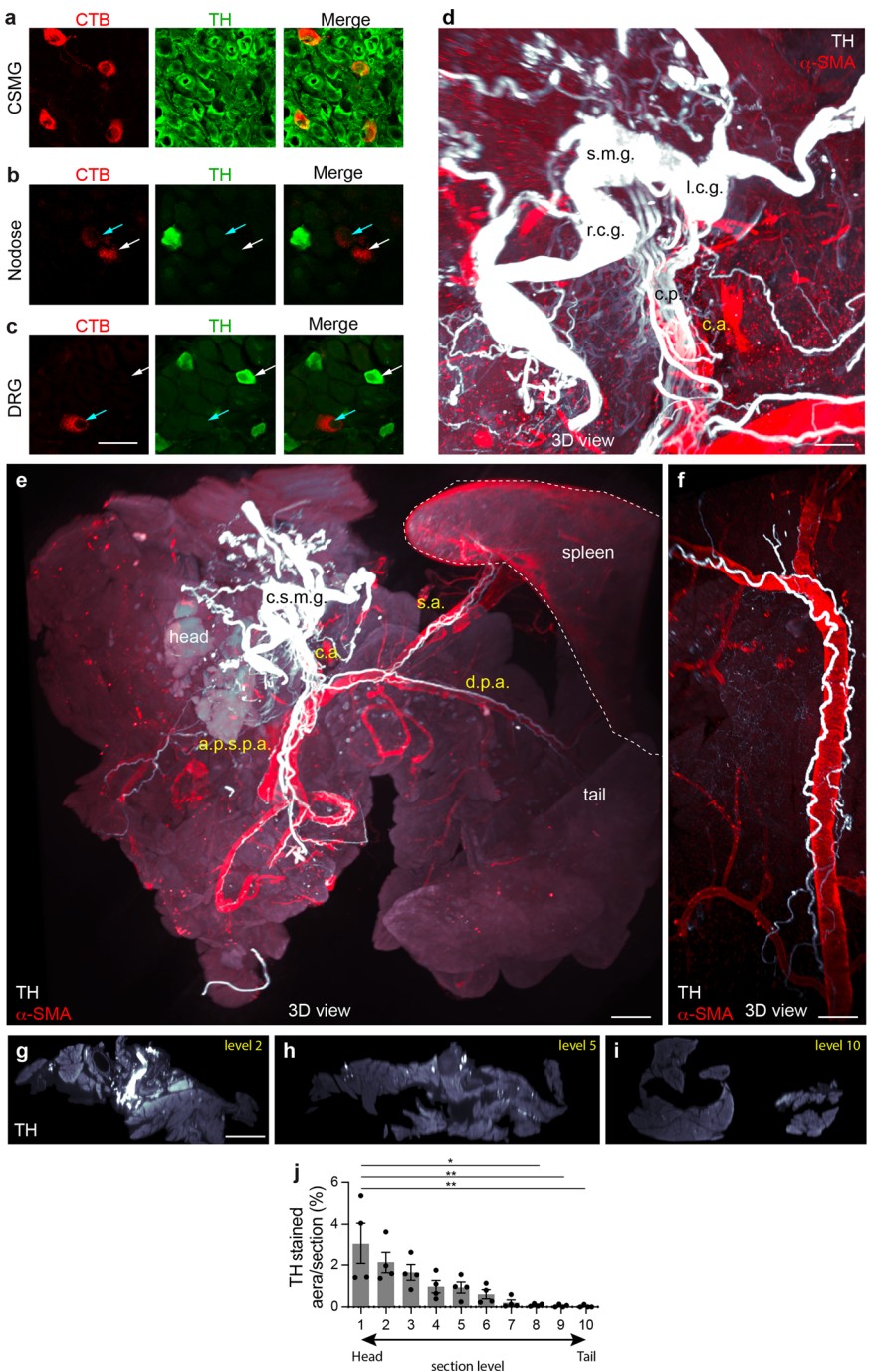

**Fig. 1 Origin and distribution of sympathetic nerves in the wild-type pancreas. a–c** Representative images of retrogradely labeled CTB[+] neurons projecting to the pancreas in tissues sections of the coeliac-superior mesenteric ganglion complex (CSMG) (**a**), nodose ganglia (**b**), and DRG (**c**) co-labeled with anti-TH antibody. In the coeliac-superior mesenteric ganglion complex, 100% of the CTB[+] neurons were TH[+] ($n = 3$ mice, 21 sections, 86 CTB[+] neurons). None of the CTB[+] sensory neurons in nodose ganglia or DRG were TH[+] ($n = 3$ mice, 4 DRGs, 29 sections, 109 CTB[+] neurons; and 3 nodose ganglia, 25 sections, 220 CTB[+] neurons). **d–f** Maximal intensity projection of 3D image stacks of an 8-week-old control murine pancreas labeled with anti-TH and anti-SMA. The coeliac-superior mesenteric ganglion complex consists of distinct ganglionic subunits (r.c.g, right celiac ganglion; l.c.g, left celiac ganglion; s.m.g., superior mesenteric ganglion) whose TH[+] efferents reach the pancreas through the celiac plexus (c.p.) surrounding the coeliac artery (c.a.) (**d**). After entering the pancreas, TH[+] sympathetic nerves travel with the main pancreatic arteries (d.p.a., dorsal pancreatic artery; a.p.s.p.a., antero and posterior superior pancreatoduodenal artery, s.a., splenic artery) (**e**, **f**). The spleen that was kept attached during dissection is outlined with a dotted line. One representative image from 3 mice is shown. **g–i** Images of TH[+] nerve sections on optical slices made at different levels from the head to the tail of an 8-week-old control pancreas. **j** Quantification of the percentage of TH[+] area on equidistant optical sections through the pancreas, from the nerve entrance in the head (level 1) to the tail (level 10). Data are presented as mean ± SEM. $n = 4$ mice. level 1 versus level 8, $P = 0.0313$; level 1 versus level 9, $P = 0.0087$; level 1 versus level 10, $P = 0.0032$ (Kruskal–Wallis test and Dunn's post hoc tests). Scale bars = 30 μm (**a–c**), 500 μm (**d**), 1 mm 200 μm (**e**), (**f**), and 1 mm (**g–i**). Source data are provided as a Source Data file.

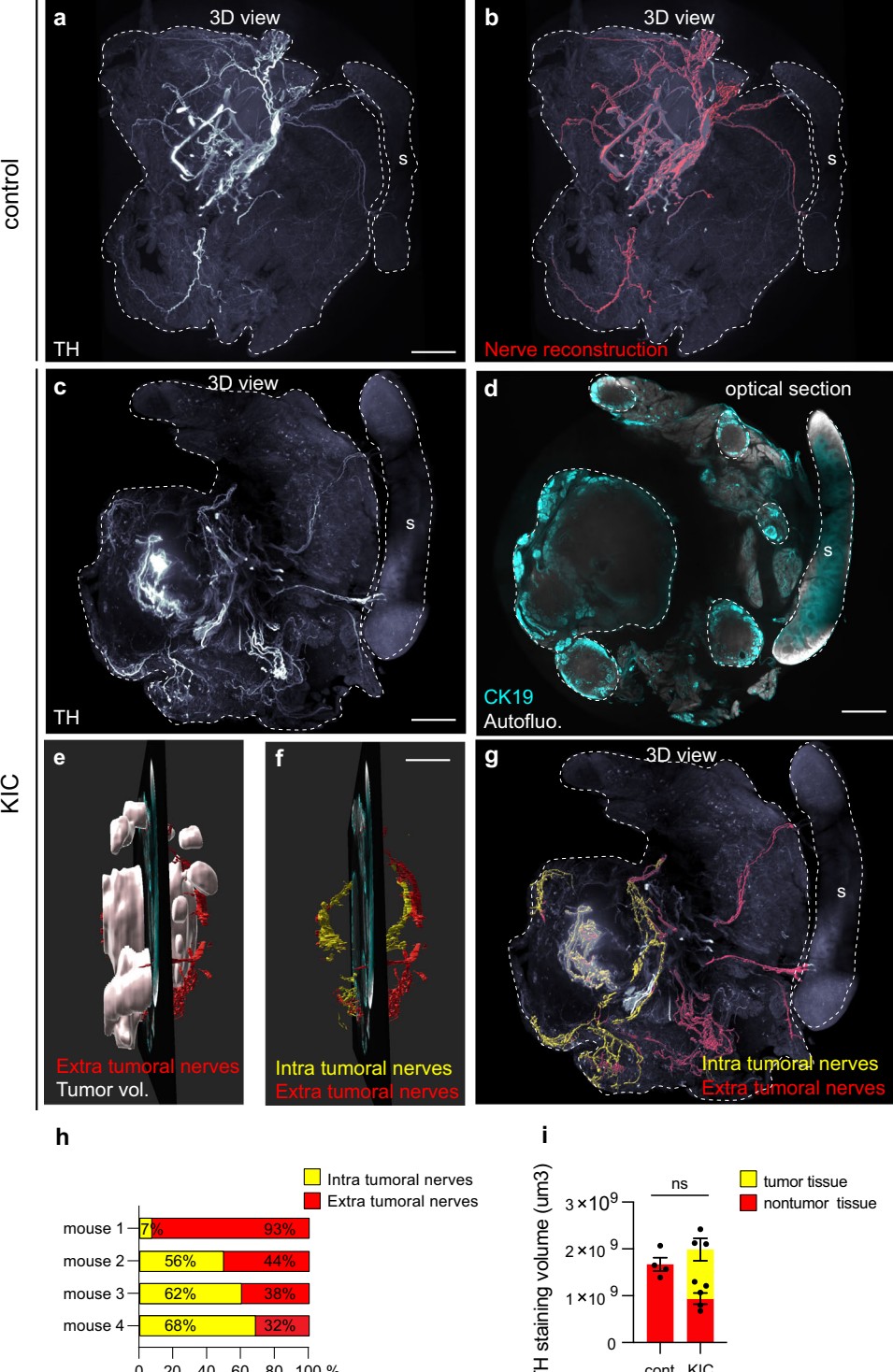

**Fig. 2 3D patterns of sympathetic nerve bundles in KIC tumors. a, b** Maximal intensity projection of 3D images of an 8-week-old control pancreas labeled with anti-TH (**a**). The reconstructed sympathetic nerve bundles have been highlighted in red (**b**). The spleen (s) that was kept attached during dissection is outlined with a dotted line. Images are representative of four mice analyzed. **c, d** Maximal intensity projection of 3D images of an 8-week-old KIC pancreas labeled with anti-TH (**c**) and anti-CK19 (**d**). In (**d**), tissue autofluorescence, imaged at an excitation wavelength of 488 nm, is shown in white. Images are representative of 4 mice analyzed. **e–g** after 3D reconstruction of the sympathetic nerves and tumor volumes (**e**), extratumoral and intratumoral nerves were artificially colored in red and yellow, respectively (**f, g**). **h** Quantification of the percentage of intratumoral and extratumoral TH+ nerves in four KIC pancreas. **i** Quantification of the total (extratumoral+intratumoral) TH+ nerve volume in control and KIC pancreas. Data are presented as mean ± SEM. $n = 4$ mice/group. Total TH in WT versus total TH in KIC, $P = 0.3429$; total TH in WT versus extratumoral TH in KIC, $P = 0.0286$ (Mann–Whitney test). Scale bars = 2 mm (**a–d, g**) and 3 mm (**e, f**). Source data are provided as a Source Data file.

innervation appeared severely disrupted by tumor development (Fig. 2a–c and Supplementary Movie 2). Previous studies have quantified intratumoral nerve density by measuring TH+ staining area on sections compared to wild-type pancreas. However, pancreatic TH+ nerve distribution is very heterogeneous (Fig. 1g–j). Therefore, the measured density in previous studies was biased since it depended on the level of sections within the organ and/or the location of the tumor. In order to accurately assess the innervation of KIC tumors, we reconstructed the sympathetic nerve trunks within the entire organ (excluding the celiac-superior mesenteric ganglia and extra-pancreatic nerves often preserved in whole-mount preparations), segmented the volume of the tumor, and then artificially colored the intratumoral nerve projections contained within that volume (Fig. 2d–g). Autofluorescence of the tissue (imaged at 488 nm of excitation) provided sufficient structural information to accurately identify the contours of the tumor, as confirmed by immunostaining for cytokeratin 19 (CK19) (Fig. 2d and Supplementary Fig. 1). We observed many sympathetic nerve trunks scattered in KIC tumors (Fig. 2g), confirming previous observations made in human samples and other transgenic models of PDAC[8,22]. When comparing the volume of intra- and extratumoral nerves, we found that the majority of samples (3/4) had a higher proportion of sympathetic nerves in the tumor than in the adjacent tissue (Fig. 2h). However, despite the obvious presence of intratumoral nerves, the total volume of TH+ nerves in KIC pancreas was similar to that in wild-type control tissues (Fig. 2i). Furthermore, by tracing the complete course of the TH+ nerve trunks, we were able to identify intratumoral nerves as portions of TH+ nerves also present in a control pancreas (Supplementary Fig. 2, for example, shows a segment of the splenic sympathetic nerve within a PDAC nodule). This overall preservation of TH+ nerves suggests that the sympathetic nerves reported in KIC tumors may not present new structures that have grown into the tumor, but rather correspond to preexisting nerve bundles that become embedded in the tumor during its development.

**Remodeling of sympathetic axon nerve terminals**. The above data suggest that the large sympathetic nerve bundles observed in pancreatic cancer are unlikely to result from tumor-induced axonogenesis. However, evidence supporting axonogenesis, particularly at the early stages of the disease, came from studies on transgenic models of PDAC, in which high densities of individual axon fibers are observed in small regions of the pancreas containing fibrosis or pancreatic intraepithelial neoplasia (PanIN) lesions[10]. These areas of hyperinnervation suggest that active axon growth may occur in a very localized manner. To address this, we employed LSFM to visualize the 3D distribution of TH-labeled sympathetic axonal endings within the pancreatic parenchyma. The main sympathetic nerve trunks described above defasciculated and splayed out into a meshwork of smaller bundles that entered each individual lobule of the pancreas (Fig. 3a). In control tissues, a dense network of fine sympathetic axonal fibers, which innervate the entire lobule, was observed (Fig. 3a–c). Co-staining of blood vessels with antibodies against Plasmalemma vesicle-associated protein (clone MECA-32) or Platelet endothelial cell adhesion molecule (PECAM) indicated that TH+ fibers were aligned with the capillaries that supply blood to the acinar parenchyma, and on which they formed synapses as revealed by expression of the presynaptic marker synaptophysin 1 (Fig. 3d–h). These results confirm previous observations of sympathetic innervation of the periacinar capillary bed[20,23]. In pancreatic lobules of 6-week-old KIC mice, we observed "hot spots" of sympathetic hyperinnervation not present in control pancreata, while innervation of the rest of the tissue was similar to

that of controls (Fig. 3i, j). These hotspots corresponded to dense networks of sympathetic fibers surrounding PanIN lesions, as revealed by autofluorescence signature analysis of the tissue (Fig. 3k and Supplementary Fig. 1) and immunoreactivity to CK19 but not insulin (Fig. 3l–o). These findings indicate that changes in the sympathetic innervation of the pancreas occur early in the development of PDAC and may involve substantial growth and remodeling of individual nerve fiber terminals.

**Sympathetic axon terminals sprout independently of blood vessels**. To thoroughly characterize how patterns of sympathetic nerve terminals change throughout successive stages of PDAC progression, we developed an image analysis pipeline that allowed a reconstruction of axonal and vascular networks from 3D LSFM images. We analyzed a dozen parameters describing axon morphology and proximity interactions with blood vessels (Supplementary Figs. 3 and 4 and Supplementary Table 1). Figure 4a and b shows original images and reconstructions of neuronal and vascular networks in acinar tissue of a control pancreas and a histologically normal (asymptomatic) region of a 6-week-old KIC pancreas. No major difference was observed, as the two tissues appeared similar in axonal and vascular density and the two systems were aligned next to each other, as evidenced by the reconstruction of their surfaces of contact. Moreover, the calculated axonal and vascular parameters were almost identical between controls and asymptomatic tissues with Z-scores close to 0 (Fig. 4g and Supplementary Data 1).

On the other hand, sympathetic innervation was higher in pancreatic regions comprising nodular fibrosis (NF), acinar-to-ductal metaplasia (ADM), and PanIN lesions, which will be hereafter collectively referred to as "noninvasive pancreatic neoplastic lesions", compared to control tissues (Figs. 4c–e). Quantification confirmed increased densities of axons with more, but smaller, axon branches (Fig. 4g), indicating that hyperinnervation of these regions was due to localized sprouting of new branches. Whereas sympathetic axons in healthy tissues were intimately associated with blood vessels, the contact surfaces between axons and vessels in noninvasive pancreatic neoplastic lesions appeared smaller (Figs. 4c–e). Quantitative analysis of proximity contacts between remodeling axons and blood vessels showed an increased fragmentation level, with several smaller-sized contact zones (Fig. 4g). Overall, a lower percentage of sprouting axon surface area was in contact with blood vessels (% Contact axon/BV) compared with that of control conditions (Fig. 4g). This decrease could not be explained by decreased vascularization, as blood vessel density was not significantly altered between the examined regions—although blood vessels appeared structurally abnormal and enlarged in neoplastic lesions (Fig. 4c–e). On the other hand, the percentage of vascular surface in contact with axons (% Contact BV/axon) remained unchanged across different regions, indicating that the reduction in axon/vessel contact size observed in the lesion regions may have been offset by the opposing increase in axonal sprouting and density.

Unbundled sympathetic axons were also detected in well-differentiated glandular regions of KIC tumors (Fig. 4f). These innervated PDAC regions were located on the periphery of tumor nodules. Well-differentiated PDAC regions had a slightly higher (but not statistically significant) sympathetic axonal density than that of control tissues and exhibited changes in axon-vessel contacts similar to those observed in noninvasive neoplastic pancreatic lesions (Fig. 4g).

Finally, using unsupervised hierarchical clustering and principal component analysis (PCA), we identified three different patterns of sympathetic innervation that distinguished the consecutive stages of PDAC progression from control and

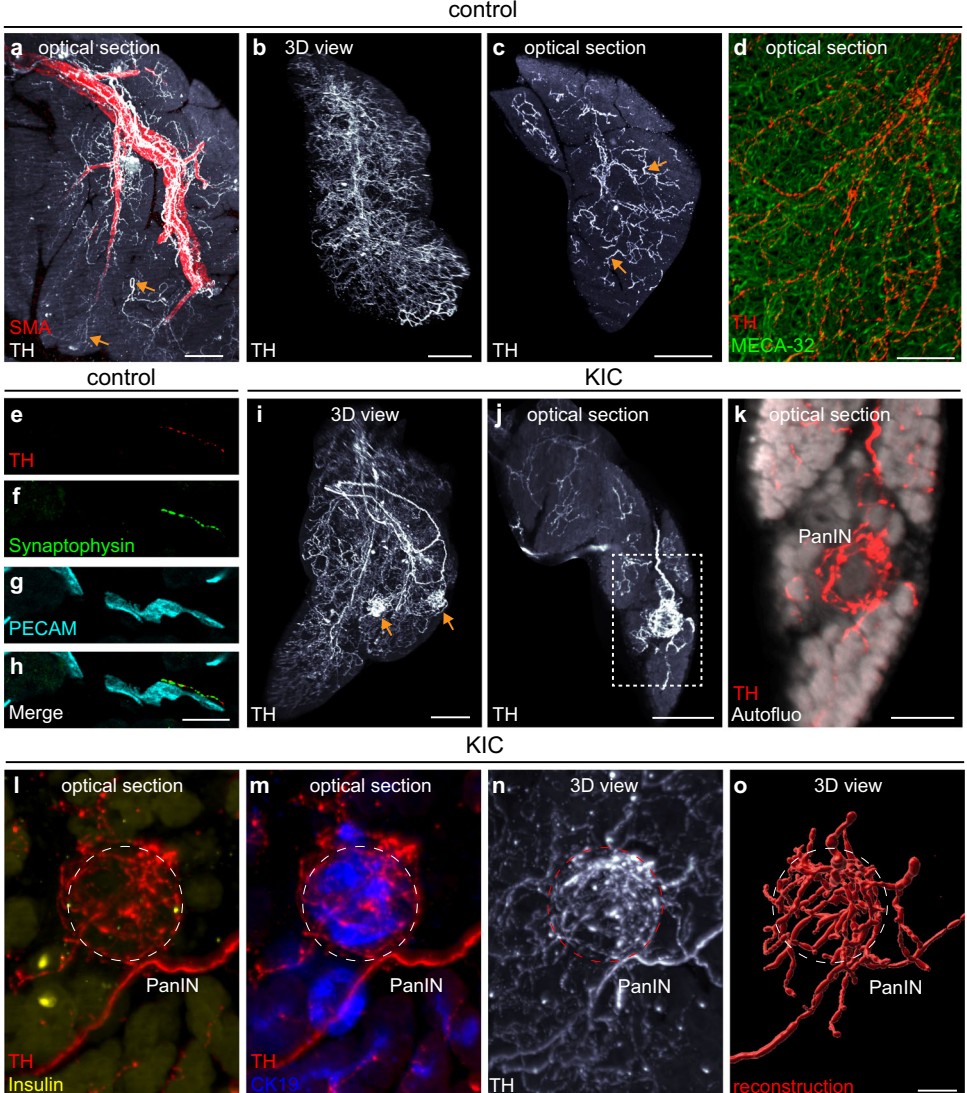

**Fig. 3 Hotspots of sympathetic innervation in KIC pancreas. a–c** Maximal intensity projection of 3D image stacks and optical sections of pancreatic lobules from control mice immunostained with anti-TH (**a–c**) and anti-SMA (**a**) antibodies to label artery-associated sympathetic nerves (**a**) and their terminals in the pancreatic parenchyma (arrows in **a** and **c**). **d** Optical section through a pancreatic lobule double-labeled with anti-TH and MECA-32 antibodies. **e–h** Immunolabeling for TH (**e**, **h**), synaptophysin 1 (**f**, **h**), and PECAM (**g**, **h**) on a section through the acinar parenchyma of a control pancreas. **i–k** 3D reconstruction (**i**) and optical section images (**j–k**) of a KIC pancreatic lobule showing "hotspots" of innervation by TH$^+$ sympathetic axons (arrows in **i**). Tissue autofluorescence, imaged at an excitation wavelength of 488 nm, indicates the presence of PanIN lesions (**k**). **l, m** Optical section images of a TH$^+$ hotspot" co-labeled with anti-insulin (**l**) or anti-CK19 (**m**) antibodies. **n, o** 3D view of the original anti-TH staining (**n**) and reconstruction (**o**) of the sympathetic axon network around a PanIN lesion. Images are representative of four mice of each genotype. Scale bars = 500 μm (**a**), 300 μm (**b, i**), 200 μm (**c, j**), 40 μm (**d**), 30 μm (**e–h**), 100 μm (**k**), and 30 μm (**l–o**).

asymptomatic tissues (group 1) to noninvasive neoplastic lesions (group 2) and invasive PDAC (group 3) (Fig. 4h).

To further examine these findings, we performed similar experiments and analyses on *LSL-Kras^{G12D/+}; LSL-Trp53^{R172H/+}; Pdx1-Cre* (KPC) mice. KPC mice begin to develop PDAC at 8 weeks and have a median survival of about 5 months[24], which allowed us to study the innervation of tumors that developed over longer periods of time compared with that of KIC mice. In the pancreas of 14-week-old KPC mice, hyperinnervation of noninvasive neoplastic pancreatic lesions and changes in axon-vessel contacts similar to those reported in KIC mice were observed (Supplementary Fig. 5A–E, G). However, the axonal density measured in areas of well-differentiated PDAC was higher than that of the KIC tumors (Supplementary Fig. 5F–G and Supplementary Data 1). Therefore, in the KPC mouse model,

PDAC clustered with the highly innervated noninvasive neoplastic pancreatic lesions in PCA (Supplementary Fig. 5H). Together, these data indicate that early and progressive sprouting and growth of sympathetic axon branches is a common characteristic of PDAC.

**No evidence of neurogenesis in PDAC.** A recent study on prostate cancer proposed that doublecortin (DCX)-expressing progenitor cells from the brain are transported via the blood-stream to tumors, where they contribute to the formation of new neurons and innervation of tumor tissues[17]. We, therefore, assessed the presence of TH$^+$ cell bodies in cleared samples and classical tissue sections of KIC pancreas. No TH$^+$ neuronal cell bodies were observed in normal acinar regions, consistent with the known organization of the sympathetic postganglionic

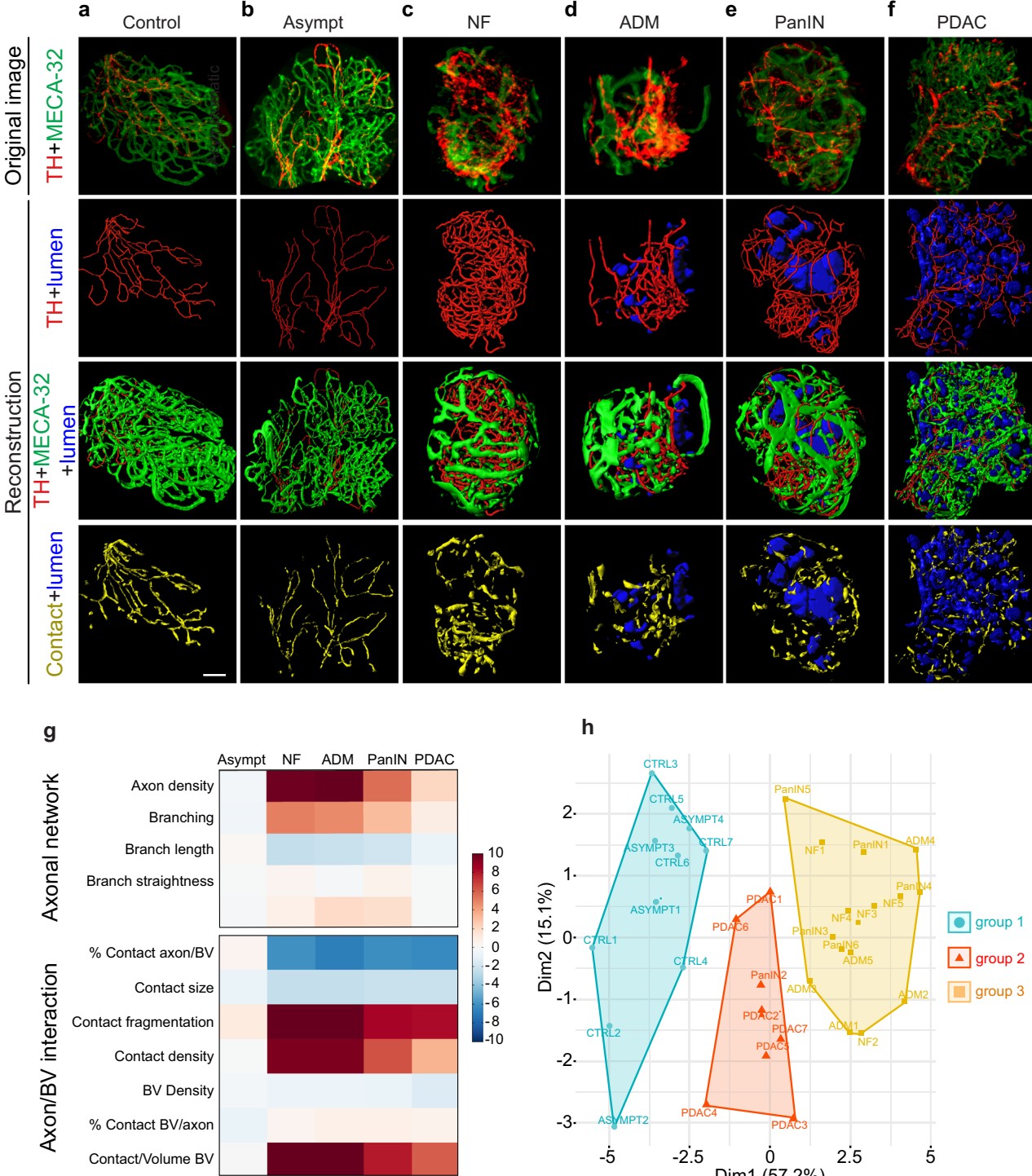

**Fig. 4 3D visualization and statistical analysis of sympathetic axon and blood vessel networks in the KIC pancreas. a–f** Representative images of 6-week-old control (**a**) or KIC (**b–f**) pancreata immunostained with anti-TH and MECA-32 antibodies (first row). 3D reconstructions of sympathetic axons (red, second row), blood vessels (green, third row), and axon/vessel surface contacts (yellow, fourth row) in normal acinar tissue (**a**), asymptomatic (Asympt) acinar tissue (**b**), NF (**c**), ADM (**d**), PanIN (**e**), and a well-differentiated PDAC region (**f**). Lumen of the epithelial lesions are represented in blue. Images are representative of 4 control and 5 KIC mice. **g** Heatmap of the Z-scores calculated for each of the 12 variables describing the architecture of sympathetic axons and their relationship with blood vessels. Values from 4 Asympt, 5 NF, 5 ADM, 6 PanIN, and 7 PDAC samples of 6-week-old KIC mice ($n = 5$) were compared with those of 7 normal acinar regions in age-matched control mice ($n = 4$). **h**, Factor map of the PCA performed on 34 tissue samples and 12 variables. Three cluster groups were identified corresponding to control and asymptomatic tissues (group 1, blue), noninvasive neoplastic pancreatic lesions (group 2, orange), and invasive tumor lesions (group 3, red). Scale bar = 50 μm. Source data are provided as a Source Data file.

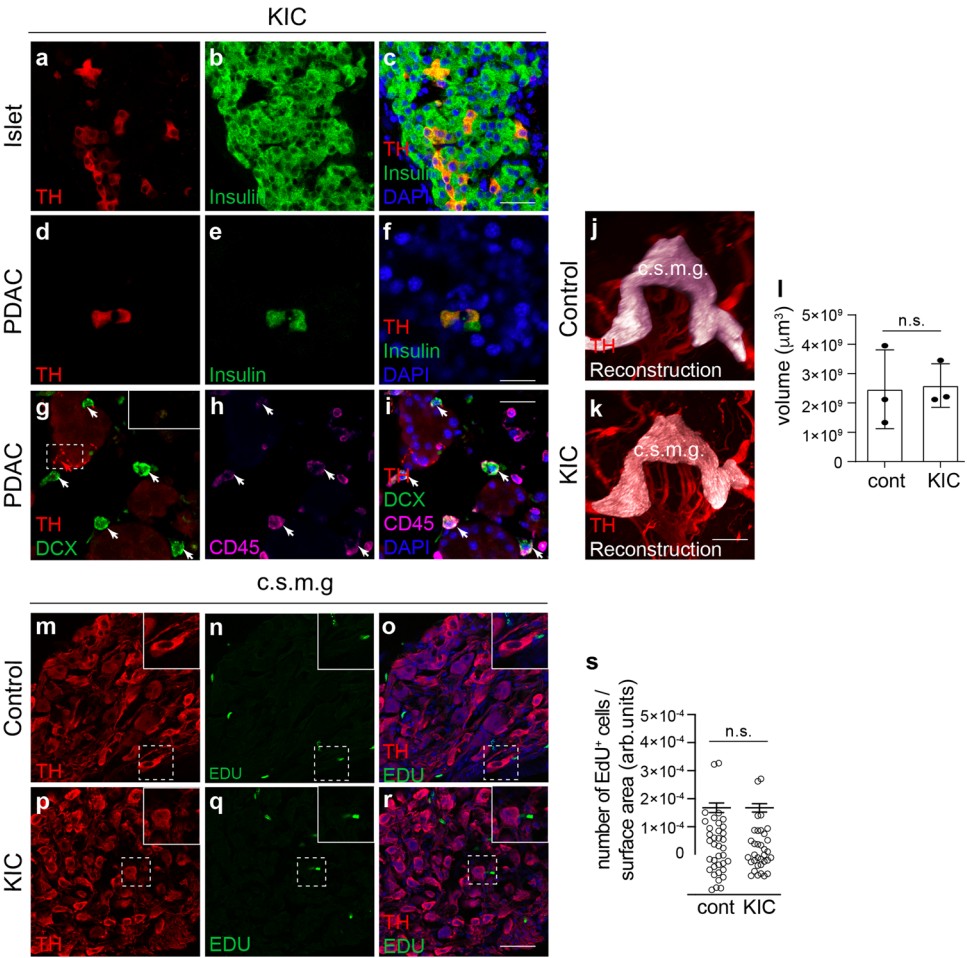

**Fig. 5 Lack of neurogenesis in KIC tumors and sympathetic ganglia. a–f** Sections through the pancreas of 6-week-old KIC mice double-labeled with anti-TH and anti-insulin antibodies. Images show TH+/insulin+ β-cells in an intact islet (**a–c**) and scattered inside the tumor (**d–f**). **g–i** PDAC sections from 6-week-old KIC mice triple-labeled with anti-TH, anti-DCX, and anti-CD45 antibodies. The inset in (**g**) shows TH staining of an axon as a positive control. Images are representative of three mice analyzed. **j, k** Representative 3D view of the coeliac-superior mesenteric ganglia (c.s.m.g.) of 8-week-old control (**j**) and KIC (**k**) mice after labeling with anti-TH (red) and ganglia volume reconstruction (white). **l** Quantification of the volume of the coeliac-superior mesenteric ganglion complex in control and KIC mice. Data are presented as mean ± SEM. n = 3 mice/group. P = 0.08 (Mann–Whitney test). **m–r** Visualization of TH+ neurons and EDU+ cells in the coeliac-superior mesenteric complex (c.s.m.g.) of 5.5-week-old control (**m–o**) or KIC (**p–r**) mice. No EDU+ cells expressed TH. **s** Quantification of the number of EDU+ cells per surface area. Data are presented as mean ± SEM. WT: n = 3 mice, 37 sections and 273 EDU+ cells; KIC: n = 3 mice, 34 sections and 265 EDU+ cells. P = 0.9407 (Mann–Whitney test). Scale bars = 20 μm (**a–i** and **m–r**) and 400 μm (**j–k**). Source data are provided as a Source Data file.

innervation of the pancreas. TH+ somata were detected in islets of Langerhans and corresponded to endocrine β-cells expressing insulin, as previously reported (Fig. 5a–c)[25,26]. Within the PDAC regions rare scattered TH+ cell bodies were observed but these also corresponded to insulin-expressing endocrine cells (Fig. 5d–f). Meanwhile, many DCX+ cells were present in tumors but none were immunopositive for TH. Instead, all DCX+ cells co-expressed the marker CD45 (Fig. 5g–j), and thus represent hematopoietic cells rather than neural precursors or immature neurons. Finally, to determine whether hyperinnervation of pancreatic lesions may result from neurogenesis in the celiac-superior mesenteric ganglion complex, 5-ethynyl-2′-deoxyuridine (EdU) was given 3 times over 10 days during early stages of neoplastic transformation. No difference in the volume of the celiac-superior mesenteric ganglion complex (Fig. 5j–l) and in the number of EdU+ cells was found in the coeliac-superior mesenteric ganglion complex of KIC mice compared to controls (Fig. 5m–s). The EdU+ cells corresponded to PECAM+ endo-thelial cells, Vimentin+ fibroblasts and Sox10+ glial cells, and may reflect normal cell and tissue turnover (Supplementary

Fig. 6). In contrast, no TH+ sympathetic neurons had incorpo-rated the proliferation marker (Fig. 5m–r). The results did not reveal a burst of proliferation, although our approach does not rule out a slower process of neurogenesis, and instead indicate that KIC tumors are innervated via the growth and branching of existing axon terminals.

To further test this notion, we employed orthotopic patient-derived xenografts (PDXs) to assess the ability of graft tumors to recruit host sympathetic axon terminals. We used two defined subtypes of PDX tumors pre-characterized as "classical" and "basal-like" in a previous multiomic analysis[27]. LSFM imaging of TH-labeled axons in undifferentiated basal-like PDXs revealed a lack of sympathetic innervation (n = 4 PDX tumors derived from two human patients). In contrast, TH-labeled fibers were detected in classical PDXs (n = 4 PDX tumors derived from two human patients). Virtual dissection and 3D reconstruction of the projection networks revealed that the observed intratumoral sympathetic axons were distal extensions of fibers that innervated the adjacent gastrointestinal tract (Fig. 6a–f). A clear demarcation between PDX and mouse pancreatic tissue was always visible,

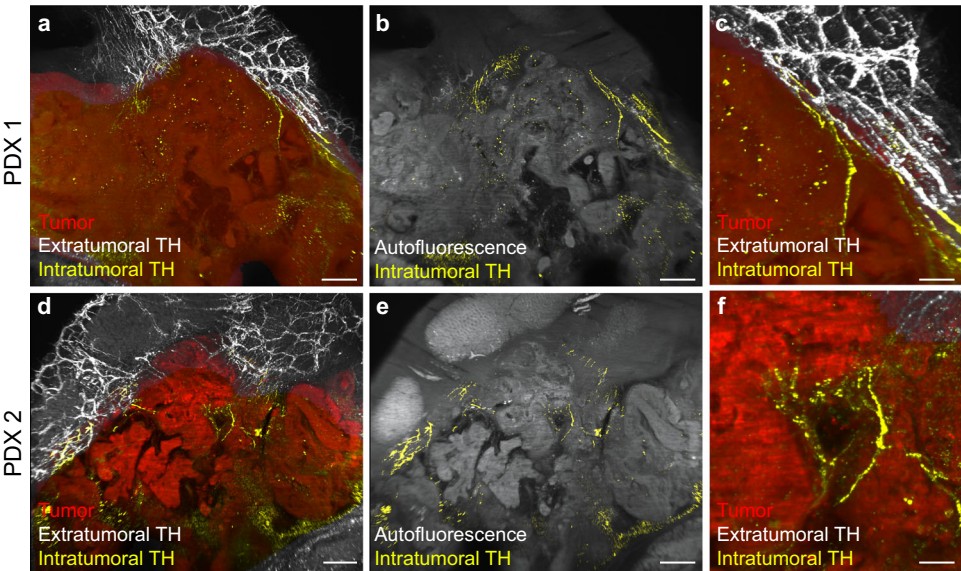

**Fig. 6 Sympathetic innervation of PDX tumors. a–f** Maximal intensity projection of 3D image stacks of PDX tumors derived from two different patients [PDX1 (**a–c**), PDX2 (**d–f**)] and immunostained with anti-TH antibody. Segmentation of PDX tumors is shown in red, intratumoral sympathetic fibers in yellow, and extratumoral fibers in white. Images are representative of four mice analyzed. Scale bars = 200 µm (**a**, **b**, **d**, **e**), and 50 µm (**c**, **f**).

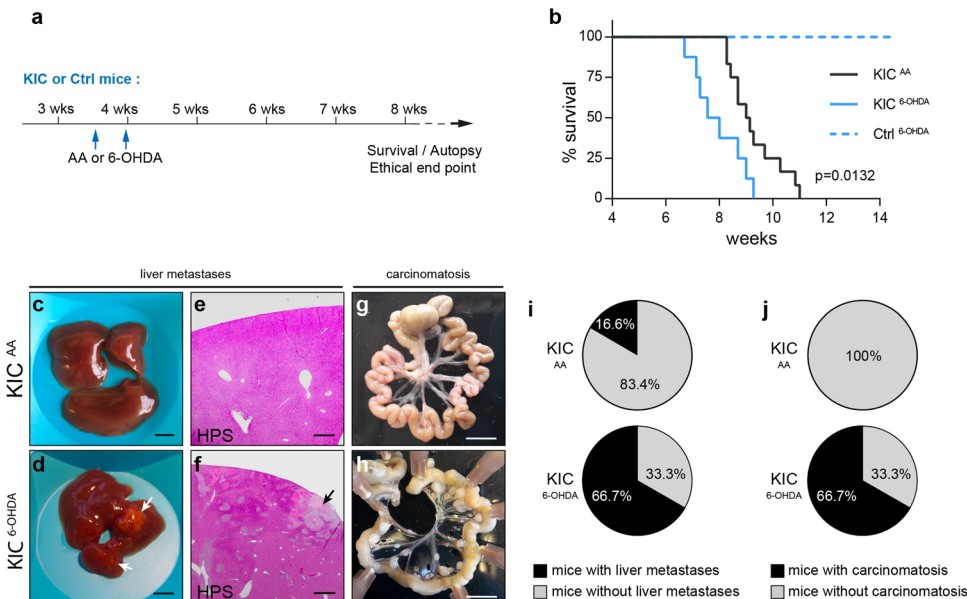

**Fig. 7 Reduced survival and increased metastatic spread in sympathectomized KIC mice. a** Outline of the experiment. **b** Kaplan–Meier curve comparing overall survival of control mice treated with 6-OHDA (Ctrl[6-OHDA], $n = 10$) and KIC mice treated with AA (KIC[AA], $n = 12$) or 6-OHDA (KIC[6-OHDA], $n = 8$). For KIC[AA] versus KIC[6-OHDA]: $P = 0.0132$ (Log-rank test) and hazard ratio B/A = 2.711 (A = KIC[AA] and B = KIC[6-OHDA]). **c–h** Representative pictures of livers (**c**, **d**), HPS-stained liver sections (**e**, **f**), and intestine (**g**, **h**) of AA- or 6-OHDA-treated KIC mice collected at autopsy. The arrows point to macrometastases. **e**, **f** show representative images of three mice analyzed per treatment group. **i–j** Pie charts showing the percentage of AA- and 6-OHDA-treated KIC mice without and with liver metastasis (**i**) and carcinomatosis in the intestine mesentery (**j**) at death (KIC[AA], $n = 9$; KIC[6-OHDA], $n = 6$). Scale bars = 5 mm (**c**, **d** and **g**, **h**), and 500 µm (**e**, **f**). Source data are provided as a Source Data file.

which excluded a simple engulfment of mouse fibers. These data confirm that both murine and PDX tumors are capable of stimulating growth and recruiting preexisting sympathetic axons from surrounding tissues.

**Sympathectomy accelerates tumor growth and metastasis and decreases survival.** The neuroplastic changes described are likely to play a significant role in PDAC progression. To test this, we used the neurotoxin 6-hydroxydopamine (6-OHDA) to induce

peripheral chemical sympathectomy in KIC mice (Fig. 7a). Mice received two intraperitoneal (i.p.) injections of 6-OHDA or ascorbic acid (AA) vehicle solution between 3.5 and 4 weeks of age, i.e., after the sympathetic-dependent maturation of endocrine pancreatic functions and before premalignant lesions develop[21,25]. 6-OHDA treatment selectively eliminated 83.9% of TH[+] axons residing in acinar tissue, without affecting vesicular acetylcholine transporter (VAChT)-positive parasympathetic fibers or blood vessel density (Supplementary Fig. 7A–J). We found that 6-OHDA treatment significantly reduced KIC mouse survival compared with

vehicle-treated mice (median survival, KIC$^{AA}$: 9.1 weeks, KIC$^{6-OHDA}$: 7.8 weeks; Fig. 7b). Chemical sympathectomy had no effect on the survival of non-tumor-bearing mice during the same period of time (Fig. 7b). Notably, KIC mice rarely developed macrometastatic disease[21], whereas autopsy at death revealed high percentages of 6-OHDA-treated mice with hepatic metastasis (KIC$^{AA}$: 16.7%, KIC$^{6-OHDA}$: 77.8%) or peritoneal carcinomatosis (KIC$^{AA}$: 8.3%, KIC$^{6-OHDA}$: 77.8%) (Fig. 7c–j).

To further assess these results, we performed surgical sympathectomy by severing the nerves entering the pancreas, which contained a mix of sympathetic and sensory fibers (Supplementary Fig. 7A–J). Compared with sham-operated animals, KIC mice that underwent pancreatic denervation showed decreased survival (median survival, KIC$^{Sham}$: 9.0 weeks, KIC$^{SympX}$: 8.1 weeks, Supplementary Fig. 7K, L) and a higher proportion of liver metastases (KIC$^{Sham}$: 33.3%, KIC$^{SympX}$: 66.7%) and carcinomatosis (KIC$^{Sham}$: 11.1%, KIC$^{SympX}$: 33.3%) (Supplementary Fig. 7M–R).

Next, to precisely analyze the effect of the sympathetic nervous system on primary tumor growth, we performed chemical sympathectomy of KIC mice with 6-OHDA at 3.5–4 weeks and comprehensively analyzed tumors from animals at 6.5 weeks when the tumors became palpable (Fig. 8a). Total pancreatic weight increased in sympathectomized KIC mice compared with vehicle-treated KIC animals (Fig. 8b–d). Pathological evaluation of the denervated pancreas revealed that the area occupied by the tumor increased by 30% at the expense of histologically normal tissue (Fig. 8e–g). These larger tumors were associated with increased cell proliferation (Fig. 8h, i, p), increased fibrosis (Fig. 8j, k, q), reduced vascular density (Fig. 8l, m, r), and increased hypoxia (Fig. 8n, o, s). All these changes are indicative of more advanced or aggressive malignancy. Altogether, these results demonstrate that the sympathetic nervous system exerts a protective function during the early development and progression of pancreatic cancer in the KIC mouse.

**An intact immune system is required for tumor response to sympathectomy**. The promotive effect induced by sympathetic denervation on tumor growth was further tested in a syngeneic orthotopic mouse model of PDAC. We used luciferase-expressing epithelial cancer cells isolated from primary KIC (PK4A-Luc cells) or KPC (R211-Luc cells) tumors[28,29]. The in vitro growth of PK4A-Luc cells was unaffected by exogenous addition of nor-epinephrine ($10^{-6}$–$10^{-8}$ M), the main neurotransmitter of the sympathetic nervous system (Fig. 9a). In contrast, norepinephrine ($10^{-6}$ M) exerted a small but significant suppression of R211-Luc cell growth, which was blocked by the beta1-blocker atenolol and only partially blocked by the beta2-blocker butoxamine (Fig. 9b). PK4A-Luc cells were injected into the pancreas of 6-OHDA-lesioned or vehicle-treated (AA) syngeneic mice and longitudinal monitoring of tumor development were assessed by biolumines-cence imaging (Fig. 9c, d). For each mouse, tumor growth was described by a Gompertz equation (Fig. 9e, f) and a Bayesian hierarchical model was designed to compare the sympathecto-mized mice to the control group (see "Methods"). Figure 9g shows the 2D-posterior probability distributions for the estimated para-meters of the Gompertz equation: log $b$ (logarithm of the max-imum bioluminescence value, or plateau) and $a$ (speed at which the logarithm of the growth curve reaches its maximum value). The two groups appeared clearly separated, with tumors growing to a larger size (higher mean log $b$ value) in the 6-OHDA-treated groups (Fig. 9g). Similar results were obtained using R211-Luc cells: tumors of the sympathectomized mice grew exponentially during a longer period of time (lower mean $a$ value), to a larger size (higher mean log $b$ value) when compared to controls

(Fig. 9h). Given the potential function of neural signals in mod-ulating immunity, we next tested whether the response to sym-pathectomy involves the immune system by transplanting R211-Luc cells into the pancreas of immunodeficient athymic nude mice (Fig. 9i). Under these conditions, tumor growth was no longer different between the AA and 6-OHDA groups, indicating that the tumor response to sympathectomy requires an intact host immune system. Taken together, these data indicate that while direct noradrenergic signaling from the sympathetic nervous system to cancer cells may contribute in part to the inhibitory activity of the sympathetic nervous system on PDAC development in the KIC mouse, the tumor response to sympathectomy is more likely mediated indirectly by changes in the immune environment.

**Sympathectomy increased intratumoral CD163$^+$ macrophages**. To begin to elucidate the immune mechanisms underlying the effects of sympathectomy, we performed digital spatial profiling (DSP) to quantify 30 immune-related proteins in tumors from control (AA) and sympathectomized (6-OHDA) KIC mice (see "Methods"). Regions of interest (ROIs) were selected at the per-iphery of the tumors where sympathetic fibers are normally located, and each ROI was segmented into immune-dominant and tumor-dominant regions based on CD45 and pan-cytokeratin immunofluorescent staining, respectively. Data from immune-dominant ROIs, revealed the myeloid marker MHCII as the only protein upregulated following sympathectomy (Supple-mentary Data 2). In tumor-dominant ROIs, however, several proteins appeared to be upregulated, including MHCII, OX40L, and the macrophage marker CD163 (Fig. 10a and Supplementary Data 2). CD163$^+$ tumor-associated macrophages (TAMs) repre-sent a pro-tumorigenic fraction of TAMs that can suppress T-cell-mediated antitumor immunity[30]. Whereas T-cell subtype markers (CD3e, CD8a, CD4, and forkhead box P3 (FoxP3)) did not vary between sympathectomized and control samples, in contrast, the immune checkpoint CTLA4 (cytotoxic T-lymphocyte-associated antigen 4) emerged as the protein most significantly upregulated by 6-OHDA treatment (Fig. 10a). Fur-thermore, a positive and significant correlation was found between CD163 and CTLA4 expression in the epithelial tumor compartment of sympathectomized tumors (Fig. 10b).

To confirm the change in macrophage populations, we immunolabeled KIC tissue sections with CD163 and the pan-macrophage markers F4/80. CD163$^+$ macrophages accounted for nearly half of the macrophages found in asymptomatic regions of KIC pancreas (Supplementary Fig. 8A–D). In these same regions, administration of 6-OHDA did not alter the density of CD163$^+$ or F4/80$^+$ macrophages (Supplementary Fig. 8A–D). In contrast, a significant increase in CD163$^+$ macrophage number was observed in noninvasive pancreatic lesions (ADM and PanIN) of 6-OHDA-treated pancreata, while the total number of F4/80 macrophages remained constant (Supplementary Fig. 8E–L). Sympathectomy was also associated with an important change in the distribution of CD163$^+$ macrophages that appeared to infiltrate PDAC tumors (Fig. 10c, d). These observations were supported by the increase in intratumoral CD163 expression levels and CD163/F4/80 ratio in 6-OHDA-treated PDAC compared with controls (Fig. 10e–g).

A close association between F4/80$^+$CD163$^-$ macrophages and sympathetic innervation was found in both noninvasive and invasive pancreatic lesions (Supplementary Figs. 8E, I and 9). We thus assessed whether the sympathetic neurotransmitter norepi-nephrine could regulate Cd163 expression levels in macrophages. Pancreatic CD45$^+$F4/80$^+$CD163$^+$ macrophages were isolated from control mice (Supplementary Fig. 8M, N) and cultured with different concentrations of norepinephrine. Cd163 mRNA was

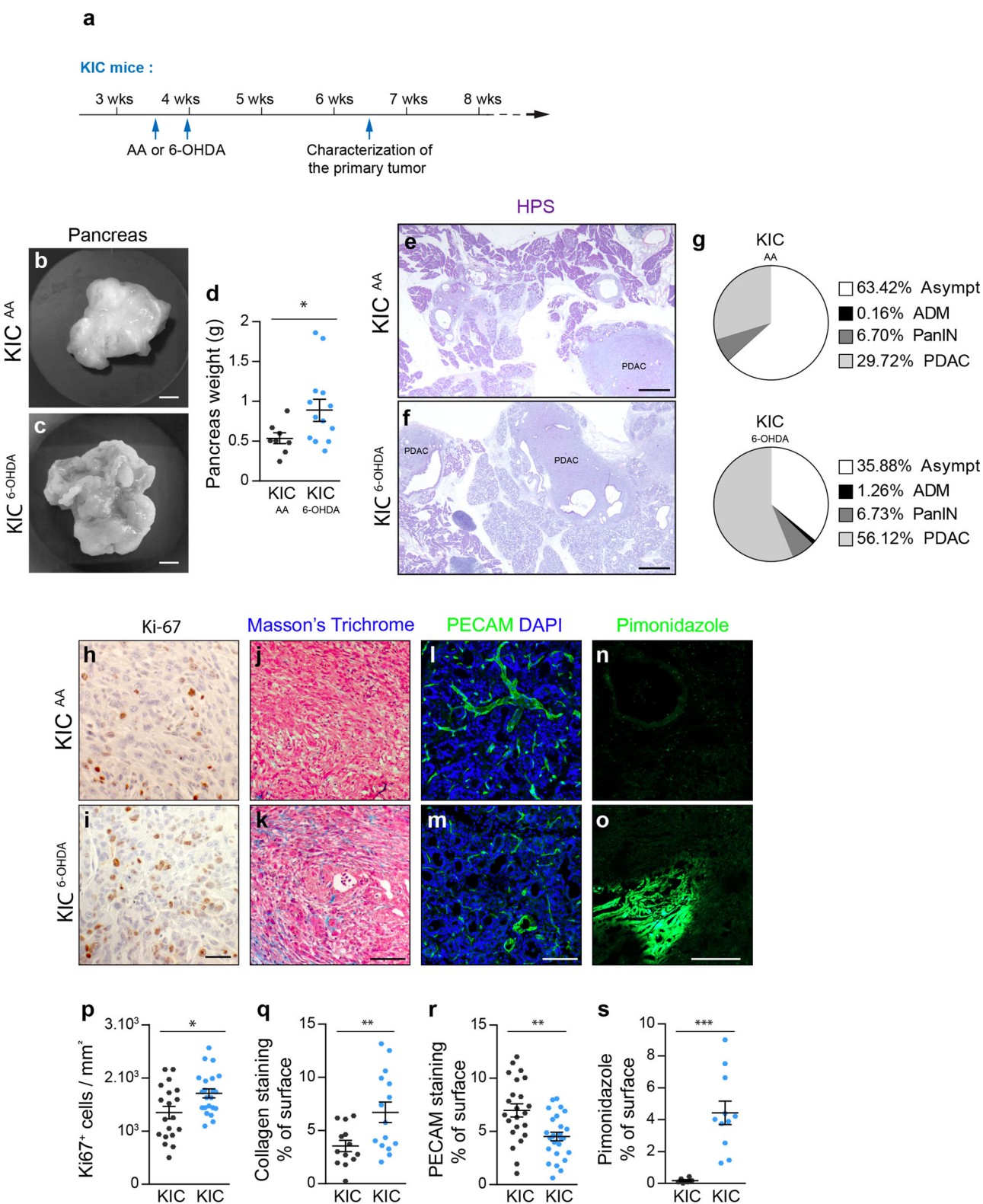

significantly downregulated in macrophages exposed to the higher dose ($10^{-6}$ M) of norepinephrine (Fig. 10h). Taken together, these results indicate an immunomodulatory function of norepinephrine release by sympathetic nerve terminals that would contribute to limit the proportion of pro-tumorigenic and immunosuppressive CD163$^+$ macrophages in pancreatic pre-neoplastic and tumor tissues.

**Depletion of CD163$^+$ macrophages rescued the effect of sympathectomy.** CD163 expression of TAMs has been linked to poor prognosis in a number of human cancers. In an analysis of macrophage populations in PDAC specimens from 34 patients treated at the North Hospital of Marseille (France), we found that high intratumoral expression of CD163, but not of the pan-macrophage marker CD68, was associated with shorter disease-

**Fig. 8 Accelerated tumor growth in sympathectomized KIC mice. a** Outline of the experiment. **b, c** Representative pictures of tumoral pancreata from 6.5-week-old KIC mice treated with AA (**b**) or 6-OHDA (**c**). **d** Scattered dot plot of pancreatic weight from 6.5-week-old KIC mice treated with AA (KIC$^{AA}$, $n = 9$) or 6-OHDA (KIC$^{6\text{-OHDA}}$, $n = 13$). $P = 0.0348$ (unpaired $t$ test with Welch's correction). **e, f** Bright-field images of HPS-stained pancreatic sections from 6.5-week-old KIC mice treated with AA or 6-OHDA. Images are representative of 3 mice analyzed per treatment group. **g** Pie charts showing the percentage of surface covered by asymptomatic tissue (Asympt), ADM, PanIN, and PDAC in pancreatic sections from 6.5-week-old KIC mice treated with AA ($n = 3$ mice, 12 sections) or 6-OHDA ($n = 3$ mice, 12 sections). **h–o** Representative images of immunohistochemistry for Ki-67 (**h, i**), Masson's trichrome staining (**j, k**), immunofluorescence for PECAM (**l, m**), and pimonidazole staining (**n, o**) in PDAC sections from 6.5-week-old KIC mice treated with AA or 6-OHDA. **p–s** Scattered dot plots of the densities of Ki-67$^+$ cells (**p**), collagen (**q**), PECAM$^+$ vessels (**r**), and pimonidazole$^+$ hypoxic areas (**s**) in PDAC sections of 6.5-week-old KIC mice treated with AA or 6-OHDA. **p** $P = 0.0145$, unpaired $t$ test with Welch's correction (KIC$^{AA}$, $n = 3$ mice, 19 images; KIC$^{6\text{-OHDA}}$, $n = 3$ mice, 22 images). **q** $P = 0.0085$, unpaired $t$ test with Welch's correction (KIC$^{AA}$, $n = 3$ mice, 13 images; KIC$^{6\text{-OHDA}}$, $n = 3$ mice, 15 images). **r**, $P = 0.0014$, unpaired $t$ test (KIC$^{AA}$, $n = 6$ mice, 23 images; KIC$^{6\text{-OHDA}}$, $n = 5$ mice, 26 images). **s** $P = 0.0002$, Mann–Whitney test (KIC$^{AA}$, $n = 1$ mouse, 6 images; KIC$^{6\text{-OHDA}}$, $n = 2$ mice, 11 images). Data are represented as median ± SEM. Scale bars = 5 mm (**b, c**), 500 μm (**e, f**), 50 μm (**h, i**), 100 μm (**j, k**), and 200 μm (**l-o**). Source data are provided as a Source Data file.

free survival (CD163$^{low}$: 17.5 months, CD163$^{high}$: 11.5 months; Fig. 10i, j), confirming a previous report[31]. Based on these findings, we therefore decided to test the contribution of CD163$^+$ macrophages to the poor survival of sympathectomized KIC mice. To this end, we used doxorubicin-loaded liposomes coated with CD163 antibody (DxR lipo) that selectively eradicate CD163$^+$ TAMs, without affecting other TAM populations[32,33]. Liposome administration via the retro-orbital route in KIC mice led to targeted depletion of 68.5% of the CD163$^+$ TAMs in PDAC, whereas control liposomes non-loaded with doxorubicin (Ctrl lipo) exhibited no effects (Supplementary Fig. 10). Sympathectomized KIC mice were treated with either DxR lipo or Ctrl lipo three times per week and survival was monitored (Fig. 10k). Ablation of CD163$^+$ macrophages improved survival of sympathectomized mice (median survival, KIC$^{6\text{-OHDA/Ctrl lipo}}$: 6.8 weeks, KIC$^{6\text{-OHDA/DxR lipo}}$: 8.4 weeks), which was restored to the same level as non-denervated KIC mice (Fig. 10l). Taken together, these findings support a role for CD163$^+$ macrophages as mediators of the pro-tumoral effect of sympathectomy in PDAC.

## Discussion

A typical feature of pancreatic carcinoma is the presence of intratumoral nerves, often hypertrophic and in greater numbers than in normal pancreatic tissue[34]. Pancreatic cancer is therefore thought to exert neurotrophic properties that induce the formation and growth of new nerves into the tumor. In contrast to this model, this study showed that healthy and tumor pancreata contain similar amounts of sympathetic nerves and those nerve trunks found within KIC tumors are preexisting pancreatic nerves. These nerves have been engulfed by the tumor and their presence is therefore not the result of an active process of nerve growth and plasticity. Our data on KIC mice do not exclude that sympathetic nerve expansion occurs in the human PDAC or in other mouse models, however, de novo nerve formation in PDAC remains to be formally demonstrated. 3D imaging highlighted the heterogenous distribution of sympathetic nerves within the pancreas. The frequent presence of intratumoral nerves in human PDAC may thus reflect the fact that the head of the pancreas, where major arteries and nerves enter the organ, is by far the most common tumor site for pancreatic cancer[35]. Moreover, in experimental PDAC tumors, increased intratumoral nerve density was reported after treatments causing a concomitant increase in tumor growth[8]. These larger tumors that have invaded most of the organs are more likely to contain embedded preexisting nerves. Thus, tumor size and location are parameters that may account for differences in intratumoral nerve density, without the growth of new nerves being involved. This is a critical point that should be considered in studies evaluating the prognostic value of intratumoral nerve density in cancer.

While sympathetic nerves do not appear to undergo active changes in KIC tumors, we found a significant increase in terminal axon density in noninvasive neoplastic lesions and, to a lesser extent, in well-differentiated regions of the tumors. It is unlikely that these axons emerged from new intra-pancreatic neurons generated by brain-derived precursors as proposed[17], as we found no TH$^+$ neuronal cell bodies in the pancreas and tumors. We did, however, detect DCX-expressing cells in pancreatic tumors. In addition to being a neurogenesis marker, the microtubule-associated protein DCX is also expressed outside neurogenic niches by hematopoietic cells, including CD8$^+$ T cells[36]. We, therefore, propose that the DCX$^+$/CD45$^+$ cells observed in PDAC correspond to infiltrating immune cells and do not reflect intratumoral neurogenesis.

Sympathetic innervation of pancreatic lesions would therefore result from sprouting of new axon collaterals from axon terminals that normally innervate blood capillaries located close to the lesioned areas. This is supported by our data showing an increased number of short axonal branches in both noninvasive neoplastic and invasive pancreatic lesions in both KIC and KPC mice. Sprouting responses from sympathetic axons have been documented in several chronic inflammatory diseases of the skin, joints, and mucosa[37–39]. PDAC is a typical example of inflammation-linked cancer. The persistent chronic micro-inflammation induced by oncogenic Kras that drives the initiation of precancerous lesions[40] may be at the origin of abnormal axonal sprouting and sympathetic hyperinnervation. It is therefore likely that ADM induced in chronic pancreatitis are also areas of sympathetic hyperinnervation, similar to Kras-induced ADM. The mechanisms linking inflammation to sympathetic neural remodeling remain to be fully elucidated, though a role for NGF has been proposed[41]. Sprouting may also be regulated by axon guidance molecules. Indeed, genomic alterations in axon guidance genes (including Slit/Robo and Semaphorin pathways) in human patients with PDAC as well as increased expression in mouse models of early neoplastic transformation and invasive PDAC have been identified[42]. Some axon guidance molecules can regulate adult sympathetic axon growth in cell culture systems, but their involvement in the innervation of pancreatic lesions is yet to be demonstrated in vivo.

Our results revealed an intratumoral heterogeneity in the distribution of axonal sprouts, which appears correlated with the degree of histological tissue differentiation. Indeed, sympathetic axon terminals were restricted in well-differentiated glandular areas, often located at the periphery of tumor foci, while undifferentiated regions of tumors with dense desmoplastic stroma were completely devoid of infiltrated TH$^+$ nerve endings, suggesting that these regions may be non-permissive and/or too compact for significant axon growth. Interestingly, previous PDAC classification studies identified at least two main clinically

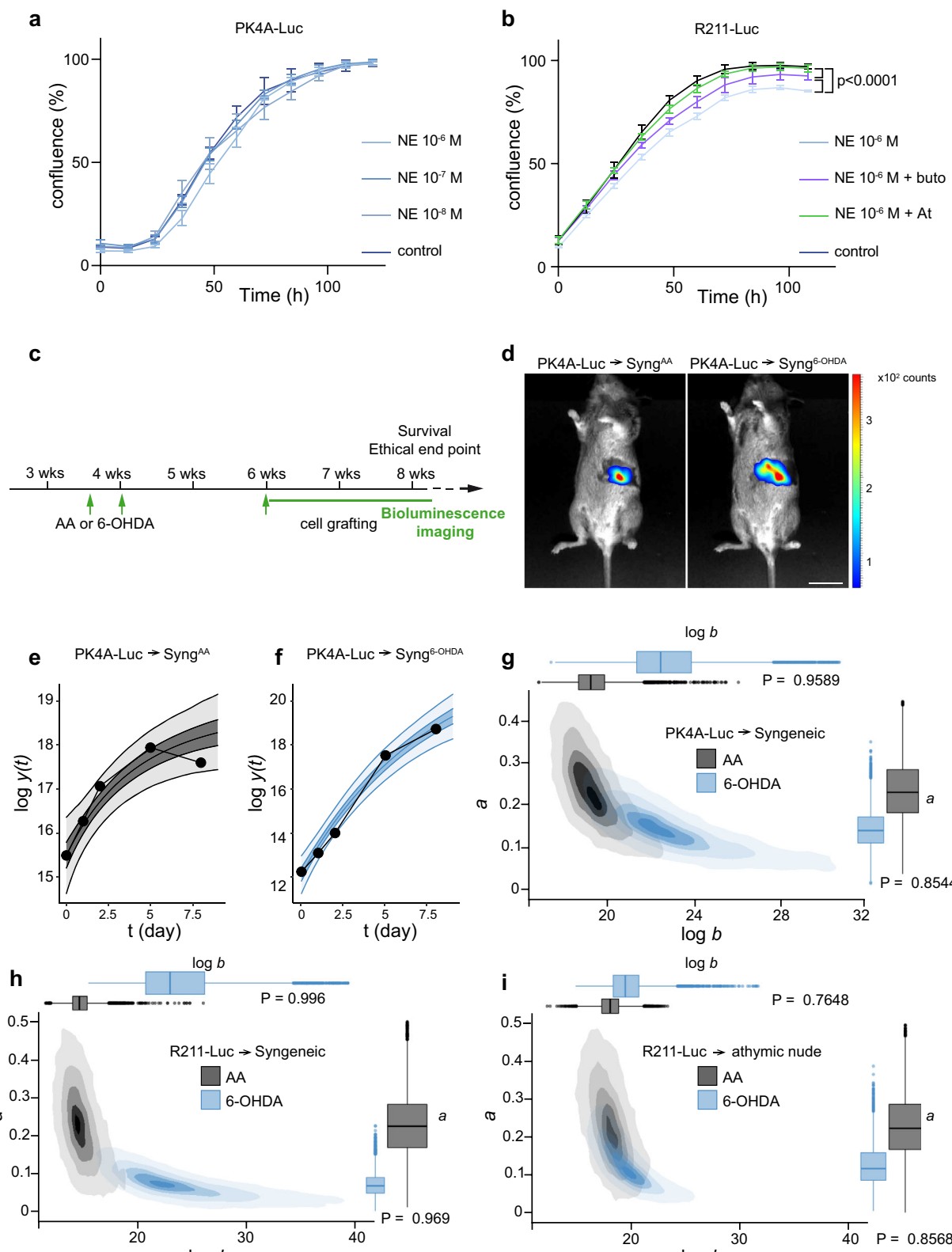

relevant subtypes, which also differ in terms of histological differentiation grades; classical tumors are more often well-differentiated and basal tumors are more often poorly differentiated[43]. The present finding that only "classical" PDX tumors were able to attract host sympathetic axon growth suggests that different subtypes of human PDAC may have distinct neural environments that contribute to their different clinical outcomes.

This study revealed a cancer-protective function of the sympathetic nervous system in the KIC mouse model of PDAC. This is reminiscent of the beneficial effect the sympathetic nervous system exerts on pancreatic tumor development in mice exposed

**Fig. 9 Effects of sympathectomy in orthotopic transplantation models. a**, **b** Cell confluence measured using IncuCyte live-cell imaging for PK4A-Luc (**a**) and R211-Luc (**b**) cells incubated with different concentrations of norepinephrine (NE), with or without atenolol (At) or butoxamine (buto). Data are presented as mean ± SEM. **a** $P = 0.9298$; **b** control versus NE, $P < 0.001$, control versus NE+ buto, $P < 0.001$, NE versus NE+buto, $P < 0.001$ (two-way ANOVA). $n = 3$ samples from three independent experiments. **c** Outline of the in vivo experiments. **d** Bioluminescence images of AA- and 6-OHDA-treated FVB/C57BL/6 mice 8 days after orthotopic grafting of PK4A-Luc cells. **e**, **f** Examples of growth curves for two representative mice from the control (**e**) and sympathectomized (**f**) groups engrafted with PK4A-Luc cells. The x axis represents time (in days), the y axis represents log-luminescence. Bioluminescence data (black dots) were fitted to a Gompertz growth curve function. Shades represent the credible regions of posterior probability 0.5 (dark tone) or 0.95 (light tone) of two mouse-level growth curves. **g–i** Comparison of the 2D-posterior distributions of the group-level shape parameters (log $b$ and $a$) of the two groups (AA in gray, 6-OHDA in blue) after syngeneic orthotopic transplantation of PK4A-Luc (**g**) and R211-Luc cells (**h**), or after transplantation of R211-Luc cells in athymic nude mice (**i**). Shades indicate high posterior density credible regions of probabilities 0.05, 0.25, 0.5, 0.75, 0.95 (from dark to light tone). The boxplots in the margins of the graph represent the marginal posterior distributions of the two parameters (log $b$ and $a$) in both populations (AA and 6-OHDA). The center line estimates the median of the posterior distribution, the box limits estimate the first and third quantiles of the posterior distribution. Each whisker is of length 1.5 times the interquartile range and is shortened when no draw from the posterior reaches or exceeds the limit of this whisker. On the other hand, draws that fall outside the whiskers are represented as dots. $P$ indicates the posterior probability of the difference between the two groups. Scale bar = 1 cm. Source data are provided as a Source Data file.

to positive stress by enriched housing conditions[13]. In the same study, chemical 6-OHDA sympathectomy also showed a tendency to promote transplanted pancreatic tumor growth in mice reared under standard environments. In contrast, the reduced survival of sympathectomized KIC mice differs from the study by Renz et al. in which surgical sympathectomy was associated with longer overall survival in the transgenic KPC model[8]. This inconsistency may be explained by the time point at which denervation was performed. In our study, denervation was performed at 3.5–4 weeks of age before invasive tumor formation (which occurs by 6 weeks of age in KIC mice[21]), whereas Renz et al. performed surgical denervation on animals whose tumors had reached a size of 20–60 mm[3]. Here we showed that selective ablation of sympathetic neurons by 6-OHDA mimics the effects of early surgical denervation. Since the surgical procedure consists or resecting mixed sympathetic-sensory nerves, this result indicates a prominent function of the sympathetic nervous system in the early steps of PDAC development. This may be consistent with the fact that the exocrine pancreas initially receives dense sympathetic innervation, whereas sensory neurons are restricted around the large blood vessels and islets but absent from the acinar tissue[23]. Early sympathectomy could prevent the remodeling of sympathetic endings that leads to hyperinnervation of ADM and PanIN lesions, and thus accelerate the progression of precursor lesions toward the invasive form of the disease. On the other hand, sympathectomy after tumor establishment may bypass this early function and instead reveal a later tumor growth-promoting function of the sympathetic nervous system. Such a function, however, is not supported by our bioluminescent imaging data, which still demonstrate a consistently inhibitory function of the sympathetic nervous system on the growth of grafted tumors. Thus, the reported effect of surgical denervation on established tumors could result from the elimination of the sensory component of the mixed sympathetic-sensory nerves. Indeed, previous studies reported that selective depletion or inactivation of sensory nerve fibers reduced tumor growth and prolonged overall survival in KPC mice[11,44]. Such a different contribution of sympathetic and sensory fibers to the outcomes of early and late surgical denervation echoes a previous report of a switch in the proportions of autonomic and sensory fibers in pancreatic nerves during the progression of human pancreatic cancer[22]. However, because sensory innervation patterns have not been fully characterized in *Kras* mouse models, further work is needed to validate this model and understand how the pancreatic neuro-environment might evolve from anti-tumoral to pro-tumoral during PDAC development.

While selective sympathetic denervation inhibits pancreatic tumorigenesis, an opposite accelerating effect on tumor development has been reported in several other cancer types, including prostate, breast, and liver cancers[3,45,46]. A determining factor could be the type of sympathetic neurons involved. A recent single-cell RNA-seq study of sympathetic neurons from the thoracic ganglion chain revealed the existence of five subtypes of TH-expressing neurons[47], indicating a greater degree of diversity and functional specialization of sympathetic neurons than previously thought. Whether this could explain their different activities during tumorigenesis awaits further investigation. Alternatively, divergent tumoral responses to early sympathetic ablation may depend on the specific type of cancer studied. In prostate cancer, increased sympathetic axon density is accompanied by increased axon/blood vessel interactions, while adrenergic signaling to endothelial cells promotes angiogenesis and thus cancer growth[48]. In pancreatic cancer, the sprouting of sympathetic axons is also likely to stimulate angiogenesis through increased local paracrine signaling to endothelial cells. In agreement, we found that sympathectomized PDAC tumors displayed reduced intra-tumor vascularization. While the primary angiogenic function of the sympathetic nervous system may be a common feature in cancer, decreased tumor vascularity after denervation, however, may not have the same impact in PDAC as in other cancers. Indeed, PDAC is typically a hypovascular tumor —due to its desmoplastic stroma—that is well adapted to hypoxic conditions[49]. Reduced vascularity, which is a favorable factor in some cancers, is on the contrary associated with poorer survival in PDAC[50]. Thus, PDAC response to sympathectomy may be shaped by its particular microenvironment.

Lastly, we identified a crucial function of the sympathetic nervous system in controlling the tumor immune environment. In KIC mice, pancreatic sympathectomy induced an increase in CD163+ macrophage density, which begins during the earliest stages of cancer progression, and later infiltration into PDAC tumors. Importantly, this "reprogramming" of macrophages mediated the pro-tumor response to pancreatic sympathectomy, as selective depletion of the CD163+ TAM population—which represents only a small proportion of all TAMs—was sufficient to rescue the overall survival of sympathectomized KIC mice. Thus, sympathetic axon sprouting in PDAC and its precursor lesions may result in a local increase in the release of norepinephrine, which can directly influence pancreatic macrophages, reducing CD163 expression and the acquisition of a pro-tumor phenotype.

The mechanism by which the CD163+ TAM subset promotes tumor progression has recently been demonstrated in mouse models of melanoma, where the CD163+ TAMs play a key role in suppressing T-cell-mediated antitumor immunity[51]. In line with this finding, we have shown here that the tumor response to sympathectomy is abolished in athymic nude mice, implicating

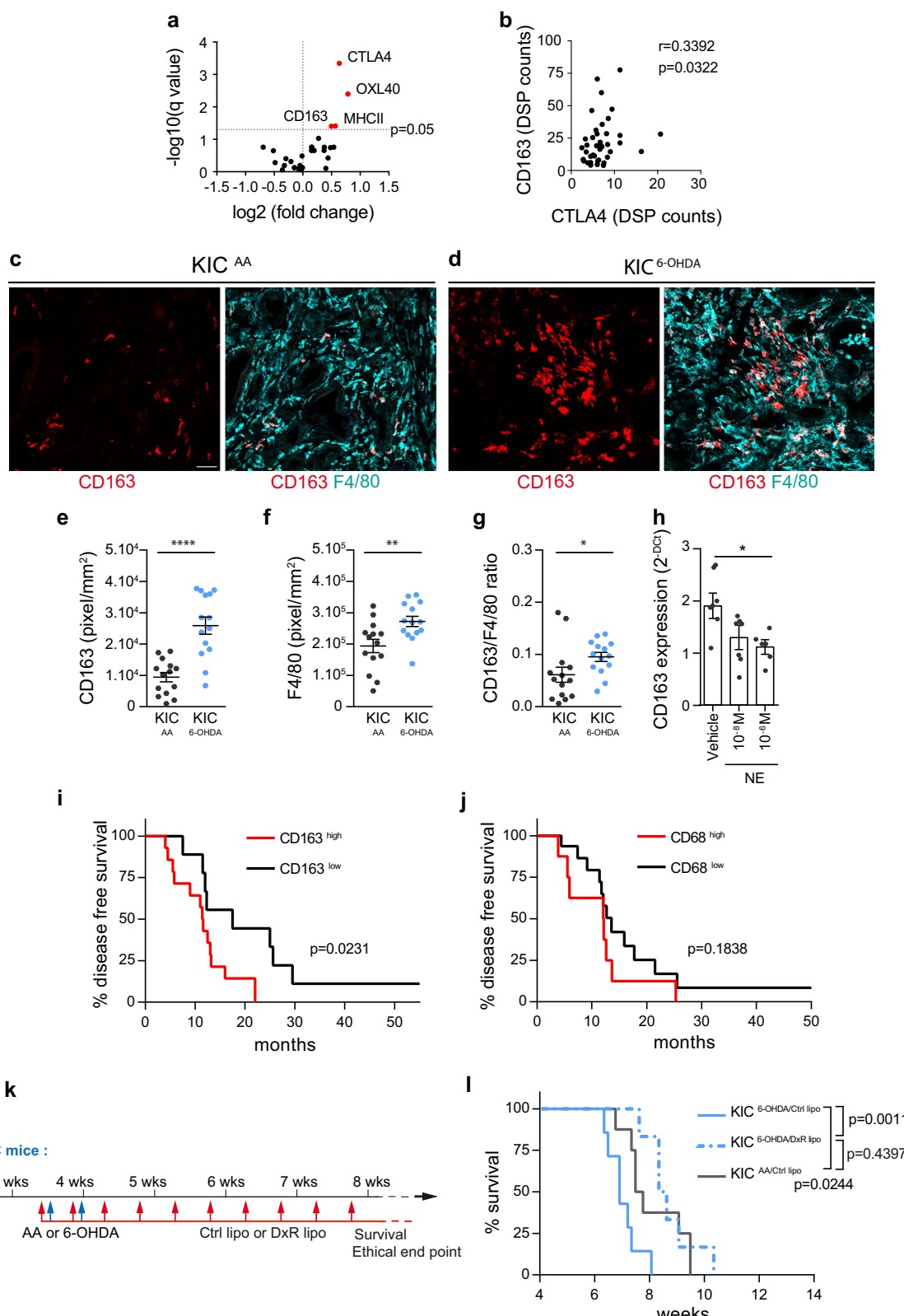

T cells in this response. Additional evidence comes from DSP analyses that revealed increased expression of the immune checkpoint CTLA4 in sympathectomized KIC tumors, correlated with an increase in CD163. A similar correlation between CD163 and CTLA4 has been previously reported in gastric cancer[52]. It is therefore possible that, in PDAC, a high level of CD163+ TAMs exert immunosuppressive activity by inducing CTLA4 expression

in tumor-infiltrated T cells. In support of this idea, TAMs isolated from human renal cell carcinoma were previously able to induce CTLA4 expression in CD4+ T cells[53].

In conclusion, we demonstrated that in KIC mice sympathetic innervation of noninvasive and invasive PDAC lesions occurs independently of neurogenesis via the sprouting of sympathetic axonal endings that normally supply the exocrine pancreatic

**Fig. 10 CD163[+] macrophages mediate the effect of sympathectomy. a** Volcano plot of DSP data in tumor-dominant ROIs from 6.5-week-old KIC mice treated with AA ($n = 5$ mice) or 6-OHDA ($n = 4$ mice). Markers significantly upregulated in 6-OHDA treated tumors are shown in red. **b** Correlation between CTLA4 and CD163 expression in tumor-dominant ROIs in KIC mice treated with 6-OHDA. $n = 4$ mice, $r = 0.3392$, $P = 0.0322$ (Spearman test). **c, d** Representative images of F4/80[+] and CD163[+] macrophages in PDAC of 6.5-week-old KIC mice treated with AA (**c**) or 6-OHDA (**d**). **e–g** Densities of CD163[+] macrophages (**e**; $P < 0.0001$, unpaired $t$ test with Welch's correction), F4/80[+] macrophages (**f**; $P = 0.0073$, unpaired $t$ test), and CD163/F4/80 ratio (**g**; $p = 0.0141$, Mann–Whitney test) in PDAC of 6.5-week-old KIC mice treated with AA (KIC[AA], $n = 4$ mice, 14 images) or 6-OHDA (KIC[6-OHDA], $n = 3$ mice, 14 images). Data are presented as median ± SEM. **h**, *Cd163* mRNA expression levels in pancreatic CD45[+]F4/80[+]CD163[+] macrophages treated or not with norepinephrine (NE). $n = 3$ samples tested in duplicate. Data are presented as mean ± SEM. Vehicle versus NE $10^{-8}$ M, $P = 0.1794$; Vehicle versus NE $10^{-6}$ M, $P = 0.0392$ (Kruskal–Wallis test and Dunn's post hoc tests). **i** Overall survival of human patients with PDAC and high intratumoral CD163 ($n = 14$) or low CD163 ($n = 9$) expression levels. $P = 0.0231$ (log-rank test), hazard ratio A/B = 3.428 and B/A = 2.445 (A = low CD163 and B = high CD163). **j** Overall survival of human patients with PDAC and high intratumoral CD68 ($n = 8$) or low CD68 ($n = 17$) expression levels. $P = 0.1838$ (log-rank test), hazard ratio A/B = 0.5552 and B/A = 1.801 (A = low CD68 and B = high CD68). **k** Outline of the experiment quantified in (**l**). **l** Survival of AA-treated KIC mice injected with Ctrl lipo (KIC[AA/Ctrl lipo], $n = 8$) and 6-OHDA-treated KIC mice injected with Ctrl lipo (KIC[6-OHDA/Ctrl lipo], $n = 7$) or DxR lipo (KIC[6-OHDA/DxR lipo], $n = 6$). KIC[AA/Ctrl lipo] vs. KIC[6-OHDA/Ctrl lipo], $P = 0.0244$ (log-rank test) and hazard ratio B/A = 2.778 (A = KIC[AA/Ctrl lipo] and B = KIC[6-OHDA/Ctrl lipo]). KIC[6-OHDA/Ctrl lipo] vs. KIC[6-OHDA/DxR lipo], $P = 0.0011$ (log-rank test) and hazard ratio A/B = 4.173 (A = KIC[6-OHDA/Ctrl lipo] and B = KIC[6-OHDA/DxR lipo]). KIC[AA/Ctrl lipo] vs. KIC[6-OHDA/DxR lipo], $P = 0.4397$ (log-rank test) and hazard ratio A/B = 1.437 (A = KIC[AA/Ctrl lipo] and B = KIC[6-OHDA/DxR lipo]). Scale bars = 500 μm. Source data are provided as a Source Data file.

capillaries. Moreover, we showed that sympathetic axons slow pancreatic tumor progression through local suppression of CD163[+] macrophage subsets at lesion sites. Recent studies reported a similar inhibitory role of the parasympathetic nervous system in PDAC, which is exerted in part by the suppression of myeloid cells[6,7]. Tumor formation and progression have been widely regarded as dependent on autonomic nerve function, stimulating interest in the clinical investigation of inhibitors of adrenergic and/or muscarinic cholinergic signaling in the treatment of certain cancers. However, our findings indicate a higher level of functional diversity of autonomic nerves in cancer than previously thought and suggest that blocking autonomic nerve function is not a therapeutic option for PDAC. The challenge is to exploit these insights into autonomic nervous system regulation of PDAC immune infiltrate and translate it into clinical strategies.

## Methods

All animal procedures were conducted in accordance with the guidelines of the French Ministry of Agriculture (approval number F1305521) and approved by the Ethics Committee for animal experimentation of Marseille – CEEA-014 (approval number APAFIS#1325-2016120211301815v1, APAFIS#17278-2018102514126851 V3, and APAFIS#17026-2018092610362806 v6). Tumor burden was assessed using a body condition score[54] and the maximum tumor burden allowed by the Ethics Committee was not exceeded in all experiments. Human tissue samples were obtained according to a protocol approved by the Regional Ethical Review Board of CRO2 - Center for Research in Oncobiology and Oncopharmacology (approval number DC-2013-1857). Only histopathological samples for which patients had consented to inclusion and storage in the biobank were included in the study.

**Mouse strains**. Wild-type mice had a C57BL/6 background or a mixed FVB/C57BL/6 background (Janvier Labs, Le Genest-Saint-Isle, France). Outbred Balb/c (Rj:A-THYM-*Foxn1*[nu/nu]) and Swiss (Crl:NU(Ico)-*Foxn1*[nu]) nude mice were obtained from Janvier Labs and Charles River Laboratories France (Ecully, France), respectively. KIC (*LSL-Kras*[G12D/+]; *Cdkn2a*[lox/lox]; *Pdx1-Cre*) and KPC (*LSL-Kras*[G12D/+]; *LSL-Trp53*[R172H/+]; *Pdx1-Cre*) mutant mice were obtained by intercrossing *LSL-Kras*[G12D/+] [55], *Pdx1-Cre*[56], and *Cdkn2a*[lox/lox][21] or *LSL-Trp53*[R172H/+] [24] mice. Animals were housed under controlled conditions (12 h light/dark cycle; humidity 45–65%; room temperature 23 ± 2 °C) and provided ad libitum access to water and food.

**Histology and immunohistochemistry**. Hematoxylin phloxine saffron stain (HPS) and Masson's Trichome stain (cat. #25088; Polyscience, Warrington, PA) were performed on 5-μm-thick paraffin sections. Pimonidazole staining was performed using the Hydroxyprobe[TM]-1 Omni Kit (cat. #HP3-1000Kit; Hypoxyprobe, Burlington, MA) according to the manufacturer's instructions. For immunohistochemistry, cryostat tissue sections were washed in PBS, blocked for 1 h in blocking solution (2% donkey normal serum, 2% bovine serum albumin, and 0.01% Triton X-100 in PBS), and incubated overnight at 4 °C with primary antibodies diluted in blocking solution. After several washes in PBS, sections were incubated for 2 h with secondary antibodies. Biotinylated antibodies were detected using the Vectastain ABC peroxidase kit (cat. #PK-6100; Vector Laboratory). The antibodies used are listed in Supplementary Table 2.

**Whole-mount immunostaining and clearing procedure**. Intact pancreata were immunostained and cleared following the iDISCO[+] protocol[57]. Briefly, mice were anesthetized by intraperitoneal injection of 100 mg/kg ketamine (Imalgene; Merial, Lyon, France) and 10 mg/kg xylazine (Rompun; Bayer, Leverkusen, Germany) and intracardiacally perfused with 20 mL of phosphate-buffered saline (PBS) followed by 30 mL of 4% paraformaldehyde (PFA) in PBS. Dissected pancreata were post-fixed overnight at 4 °C in 4% PFA. Samples were dehydrated using graded series of methanol solution (20%, 40%, 60%, 80%, 100% methanol, diluted in PBS) 1 h each at room temperature (RT) then bleached in methanol/5% hydrogen peroxide overnight at 4 °C. Samples were rehydrated using graded series of methanol solution, permeabilized in 20% dimethylsufoxyde (DMSO), 0.16% triton X and 18.4 g/L of glycine in PBS for 2 days at 37 °C, and incubated in blocking buffer: PTwH (0.2% Tween-20, 10 mg/L heparin in PBS), 5% DMSO and 3% donkey serum for 3 days at 37 °C. Tissues were incubated with primary antibodies at 37 °C for 1 week, washed in PTwH, and incubated with secondary antibodies for 2–3 days at 37 °C. The antibodies used are listed in Supplementary Table 2. In some cases, nuclear counterstaining was performed by incubation in TO-PRO[TM]-3 Iodide (642/661; cat. #T3605; Thermo Fisher Scientific, Waltham, MA) diluted in DMSO (1:1000) for 1 h 30 min at RT after incubation with the secondary antibody. Samples were washed in PTwH, dehydrated in graded methanol series, and equilibrated in 66% dichloromethane/33% methanol overnight at RT. Delipidation was completed by immersion in 100% dichloromethane for 20 min. Finally, samples were immersed in dibenzyl ether to homogenize the refractive indices between the tissue and the imaging medium.

**LSFM and image analysis**. Cleared samples were imaged on a light-sheet fluorescent microscope (Ultramicroscope II, Miltenyi Biotec, Bergisch Gladbach, Germany) equipped with a 5.5 sCMOS camera (Andor Neo) and a 2×/0.5 objective lens (MVPLAPO 2x) fitted with a 5.7 mm working distance dipping cap. Version 5.1.328 of the Imspector Microscope controller software was used. 3D volume images were generated using Imaris ×64 software (version 9.2.1 and 9.3.0; Bitplane, Zurich, Switzerland). Stack images were first converted to Imaris files (.ims) using Imaris File-Converter. 3D reconstruction of samples was then performed using 3D view in Imaris.

To quantify sympathetic nerves in the whole pancreas, TH[+] nerve trunks were reconstructed using the Imaris Surface tool and their volume ($V_{TH\ TOTAL}$) was collected in the Statistics tab. The contours of the pancreas and PDAC nodules were drawn with the Surface tool using the autofluorescence signal of the tissue and the volumes collected ($V_{Panc}$ and $V_{PDAC}$). The signal and volume of the previously reconstructed nerves contained in the segmented tumors were extracted ($V_{INTRA}$). Intratumoral nerve density was calculated as the ratio $V_{TH\ INTRA}/V_{PDAC}$ and extratumoral nerve density as $(V_{TH\ TOTAL} - V_{TH\ INTRA})/(V_{Panc} - V_{PDAC})$. Changes in nerve distribution across the pancreas were measured on 10 equidistant 180-μm optical sections covering the entire pancreas from the nerve entrance in the pancreatic head to the tail. The percentage of nerve staining was calculated as the TH signal area divided by the cross-sectional area of the pancreas, multiplied by 100. The pipeline for quantitative analysis of sympathetic terminals in pancreatic lesions is detailed in Supplementary Figs. 3 and 4, and in Supplementary Table 1. The coealic-superior mesenteric ganglia were reconstructed by delineating the region occupied by TH[+] cell bodies with the "Surface" tool and the volume was extracted. To illustrate the data, the samples could be optically sliced from any angle using the Orthoslicer or Obliqueslicer tools. In some cases, a clipping plane was created to allow better visualization of the structures of interest. 3D pictures movies were generated using the Snapshot and Animation tools. Where necessary, images were cropped and their brightness adjusted evenly using Photoshop CS6 (Adobe, San Jose, CA).

**PDX tumor models**. The establishment and maintenance of PDX via successive subcutaneous transplantation have been previously described[27]. Freshly resected xenograft (5 mm³) was sewn onto the pancreas of 8-week-old Swiss nude mice anesthetized with 2% isoflurane in oxygen. Mice were monitored twice a week and sacrificed when a palpable tumor nodule was detected. For each PDX, a total of four mice were transplanted.

**Denervation experiments**. For chemical sympathectomy, 6-hydroxydopamine (6-OHDA; cat. #H4381; Sigma-Aldrich, St. Louis, MO) was dissolved in vehicle solution (0.1% ascorbic acid [cat. #A4403; Sigma-Aldrich]) in 0.9% NaCl (saline) solution. KIC mice or control littermates received two i.p. injections of 6-OHDA (100 mg/kg at 3.5 weeks, and 250 mg/kg three days later) or vehicle solutions. For surgical sympathectomy, 3.5-week-old KIC mice or control littermates were anesthetized with 100 mg/kg ketamine and 10 mg/kg xylazine, while 100 μL of saline solution was injected subcutaneously to prevent dehydration during the procedure. Lidocain (10 mg/kg; Lurocaïne; Vétoquinol, Lure, France) was injected subcutaneously along the incision line and a midline laparotomy was performed to visualize the coeliac plexus. Nerve bundles surrounding the superior mesenteric artery were severed using microdissection forceps under a binocular microscope (Leica, Wetzlar, Germany). After nerve sectioning, the abdominal cavity was washed with pre-warmed saline solution at 37 °C, after which abdominal muscle and skin closure was performed using 6-0 (cat. #0320543; Ethicon, Cincinnati, OH) and 4-0 vicryl sutures (cat. #0320570), respectively. Buprenorphine (0.1 mg/kg; Bupaq; Virbac, Carros, France) was injected intraperitoneally after surgery and the following day. Sympathectomized and control animals were sacrificed at 6.5 weeks of age to collect primary tumors or euthanized at the endpoint requested by the Ethics Committee for survival assays.

**Orthotopic transplantation models**. Murine PK4A-Luc (RRID:CVCL_WB21) and R211-Luc[58] cells were cultivated in DMEM GlutaMAX medium (cat. #10566016; Gibco; Thermo Fisher Scientific) supplemented with 10% fetal bovine serum and 1% penicillin/streptomycin (cat. #15140122; Gibco). For R211-Luc cells, 0.02% plasmocyn (cat. #ant-mpt; InvivoGen) and 0.05% puromycin (cat. #ant-pr; InvivoGen) were added to medium. Syngeneic or BALB/c nude mice, treated with 6-OHDA or vehicle solution (see Denervation experiments), were anesthetized two weeks after denervation using 2% isoflurane in oxygen. Lidocain (3.5 mg/kg) and Buprenorphine (0.1 mg/kg) were injected intraperitoneally. A sub-costal laparotomy was performed to exteriorize the pancreatic tail. PK4A-Luc cells (150.000 cells) or R211-Luc cells (50.000 cells) in 25 μL culture medium were injected into the pancreas using an insulin syringe. Abdominal muscle and skin closure was performed using 4-0 vicryl sutures. For each experiment, a total of ten recipient mice per treatment group were transplanted.

**Bioluminescence imaging and analysis**. Bioluminescence was recorded just after cell transplantation and every 2–3 days until day 9. Bioluminescence signals were induced by i.p. injection of 150 mg/kg Luciferin-EF (E6552; Promega, Madison, WI) in PBS, 10 min before in vivo imaging. Mice anesthetized with 2% isoflurane in oxygen were imaged with a Photon Imager (Biospace Lab, Nesles-la-Vallée, France). Data analysis was conducted as follows. First, for each mouse, we expressed the bioluminescence measures with respect to time (in days) with a Gompertz growth model[59]. The bioluminescence $y_i(t_j)$ of the $i$th mouse at time $t_j$ was modeled by

$$\log y_i(t_j) = \log b_i + e^{-a_i t_j}(z_{0i} - \log b_i) + \epsilon_{ij} \quad (1)$$

The three parameters $\log b_i$, $a_i$ and $z_{0i}$ are specific to the $i$th mouse: the first two parameters give the shape of the growth curve, the last parameter is a nuisance parameter; the variable $\epsilon_{ij}$ is a normal error term. Parameter $\log b_i$ gives the logarithm of the bioluminescence when $t_j \to \infty$ which is its maximum value, and parameter $a_i$ controls the speed at which the logarithm of the growth curve reaches its maximum value. In order to fit these growth curves to the data, and to compare sympathectomized mice (6-OHDA) with the control group (AA), a Bayesian hierarchical model was designed for estimating parameters. We assumed that

$$\log b_i = \overline{\log b_{g[i]}} + \eta_i^b \quad (2)$$

$$a_i = \overline{a_{g[i]}} + \eta_i^a \quad (3)$$

where $g[i] = $ AA or 6-OHDA was the group to which belong the $i$th mouse. The variables $\eta_i^b$ and $\eta_i^a$ are normal error terms that take into account a mouse effect. The parameters $\overline{\log b_g}$, $\overline{a_g}$ give the average shape of the growth curve specific to group $g = $ AA or 6-OHDA. Computation of the posterior distribution of all parameters (the mouse-level $\log b_i$'s, $a_i$'s and $z_{0i}$'s, the group-level $\overline{\log b_g}$'s, $\overline{a_g}$'s as well as the variance of all error terms) was performed using Stan software (New BSD License) with a NUTS sampling algorithm. Then we compared carefully the posterior distributions of the group-level parameters ($\overline{\log b_g}$, $\overline{a_g}$) of both groups (AA and 6-OHDA) to provide evidence of the discrepancy between them.

**Cell proliferation**. The IncuCyte S3 live-cell analysis system (Sartorius, Göttingen, Germany) was used for kinetic monitoring of cell proliferation of pancreatic cancer

cells. PK4A-Luc and R211-Luc cells were seeded at 10⁴/well in 24-well plates in the presence of increasing concentrations of norepinephrine (cat. #A9512; Sigma-Aldrich) ($10^{-6}$–$10^{-8}$ M) with or without the adrenergic receptor inhibitors atenolol ($10^{-6}$ M; cat. # A7655; Sigma-Aldrich) or butoxamine ($10^{-6}$ M; cat. # B1385; Sigma-Aldrich). Phase-contrast images were acquired every hour for 96 h and cell proliferation was monitored by analyzing the occupied area (% confluence) of cell images over time.

**Retrograde labeling with cholera toxin B (CTB) subunit**. Adult mice ($n = 3$) were anaesthetized using 2% isoflurane in oxygen. Buprenorphine (0.1 mg/kg) was injected intraperitoneally. A laparotomy was performed to visualize the pancreas. Seven injections of a solution of Alexa Fluor 555 Dye conjugated CTB (cat. #C34776; Life technologies) (2 mg/mL in PBS) were performed at different sites in the pancreas at a flow rate of 0.2 μL/s using a programmable nanolitre injector (Nanoject III, Drummond, Broomall, USA). Care was taken to prevent leakage of the CTB and the tissue was extensively swabbed before suturing. Mice were sacrificed 7 days after CTB injection. Dissected DRG, nodose, and celiac-superior mesenteric ganglia were post-fixed in PFA (4% in PBS) 2 h at 4 °C and kept in PBS at 4 °C for further processing.

**EdU injections**. KIC and control mice ($n = 3$ mice/group) received three i.p. injections of 15 mg/kg EdU (cat. #BCN-001-500; BaseClick) diluted in PBS at 4, 5, and 5.5 weeks of age. After the last injection, the coeliac-superior mesenteric ganglia were collected and fixed in 4% PFA in PBS. The revelation of EdU labeling was performed on 20-μm-thick cryostat tissue sections by incubation for 30 min at RT in EdU reaction buffer (0.1 M Tris-HCl pH 7.0, 4 mM CuSO4 copper (II) sulfate, 0.001 mM cyanine 3 azide, and 0.1 M ascorbic acid diluted in water).

**Digital spatial profiling**. Immune cell profiling of tumor tissues was performed using the NanoString's GeoMx digital spatial profiling (DSP) at the Centre Hospitalier Lyon-Sud. The DSP technology uses a multiplexed cocktail of primary antibodies conjugated to unique oligonucleotide tags with an ultraviolet (UV) photocleavable linker. For this experiment, 4-μm-thick paraffin-embedded tissue sections from 6.5-weeks-old KIC pancreata treated with 6-OHDA ($n = 4$ mice) or AA ($n = 5$ mice) (see Denervation experiments) were incubated with a cocktail of 30 oligonucleotide-labeled antibodies targeting markers of immune cell types and checkpoint molecules. A complete list of protein targets is available in Supplementary Data 2. Tissue sections were imaged by three-color immunofluorescence using the morphological markers CD45, pan-cytokeratin (PanCK), and nuclear stain Syto13. Images of hematoxylin and eosin-stained serial sections were demarcated by a pathologist and used alongside Nanostring immunofluorescent staining to guide ROI selection. For each section, up to 10 square ROIs of 200-μm sides were drawn at the tumor periphery, defined as the outer 300-μm rim of the tumor area. Each of these ROIs was segmented based on fluorescence into PanCK-positive tumor regions and CD45 immune cell-rich regions using the GeoMx auto-segmentation tool. UV light was used to release oligonucleotide tags from discrete ROIs. Photocleaved oligos were then captured and quantified using optical barcodes in the nCounter Flex platform. For analysis, digital counts were normalized with internal spike-in controls to account for technical variation, then normalized to the geometric mean of housekeeping controls of their defined ROIs. Subsequently, the background was subtracted using the IgG controls. Counts for each target were averaged for the PanCK and CD45 segments. Multiple Mann–Whitney tests were performed to identify proteins with significantly different counts (adjusted $P$ value <0.05) between the 6-OHDA and AA-treated groups. All DSP values for the 30 markers are included in Supplementary Data 2.

**Macrophages sorting and norepinephrine treatment**. Wild-type C57BL/6 mice ($n = 3$) were transcardially perfused with PBS. Pancreata were immediately collected and dissociated using Collagenase I (4000 U; cat. #17018-029; GIBCO), hyaluronidase type I-S (4000 U; cat. #H3506; Sigma-Aldrich), DNase I (4000 U; cat. #11284932001; Sigma-Aldrich), and 500 μL HEPES in 10 mL final volume of RPMI medium for 1 h at 37 °C with agitation. Single-cell suspensions were enriched for CD45⁺ population using CD45 magnetic microbeads (cat. #130052301; Miltenyi Biotec) on the autoMACS Pro separator. Cells were blocked in 15% normal mouse serum (cat. #015-000-120; Jackson Immunoresearch) and subsequently stained with CD45-BUV395 (clone 30-F11; cat. # 564279; BD Biosciences; RRID:AB_2651134), F4/80-BV785 (clone BM8; cat. #BLE123141; Biolegend; RRID:AB_2563667) and CD163-PE (clone TNKUPJ; cat. #12-1631-82; Thermofisher/Life technologies; RRID:AB_2716924). LIVE/DEAD™ Fixable Violet Dead Cell (cat. # L34955; Thermofisher) was used for viability staining. Live CD45⁺F4/80⁺CD163⁺ were sorted using the BD FACS ARIA III sorter (BD Biosciences). Sorted macrophages were immediately cultured at 5000 cells per well in 24-well plates in the absence or presence of norepinephrine ($1 \times 10^{-6}$ or $1 \times 10^{-8}$ M, cat. #A9512; Sigma-Aldrich) for 4 h, five replicates were done per condition. RNA was extracted using RNeasy micro kit (cat. #74004; QIAGEN) and cDNA was synthesized using SuperScript™ III First-Strand Synthesis kit (cat. #18080051; Thermofisher) following the manufacturer's instructions. *cd163* mRNA expression was analyzed by qPCR using StepOnePlus Real-Time PCR Systems (Thermofisher) and SYBR™ Select Master Mix (cat. #4472918; Thermofisher). The expression of *cd163*

was normalized against the housekeeping gene *Actb* (primers CD163: forward-ATGGGTGGACACAGAATGGTT, reverse-CAGGGAGCGTTAGTGACAGCAG; *Actb*: forward-GCTGTGCTGTCCCTGTATGCCTCT, reverse-CCTCTCAGCT GTGGTGGTGAAGC).

**Liposome preparation and injections**. Liposomes encapsulating DxR were prepared and modified for CD163 targeting as previously described[32]. KIC mice were injected with 1 mg/kg liposomes (diluted in 100 μL PBS) by retro-orbital injection of the venous sinus twice a week, with the first injection performed one day before treatment with AA or 6-OHDA. For injections, mice were anesthetized with 2% isoflurane in oxygen and a drop of ophthalmic anesthetic (1% tetracaine; Laboratoire Tvm, Lempdes, France) was placed on the injection-receiving eye. For each experiment, a total of 6–8 mice per group were treated with liposomes.

**Patients and survival analysis**. Patients (*n* = 34) included in the retrospective cohorts developed PDAC and underwent diagnostic core needle biopsy or surgical resection at North Hospital, Marseille, France. Patients were diagnosed between 2005 and 2009 and followed until March 2010. Tumors were histologically staged according to the 7th edition of the American Joint Committee on Cancer (AJCC)–Union for International Cancer Control (UICC) tumor node metastasis (TNM) classification. The study included only adenocarcinomas and histological variants, as defined by the World Health Organization (WHO) classification 2010. Data on patient demographics, diagnosis, surgical resection, and outcome were retrieved from medical records in the hospital database. Formalin-fixed, paraffin-embedded human sections were immunolabeled for CD68 and CD163 (details of antibodies are available in Supplementary Table 2) and counterstaining with Mayer's hematoxylin (cat. #S3309; Dako, Jena, Germany). Immunohistochemical images were analyzed in collaboration with the anatomical pathology laboratory of Hospital North using Calopix software (TRIBVN, Châtillon, France) and recorded in a database. The cohort was separated into two groups according to the level of CD163 and CD68 expression in their tumors (high- and low-expression groups with 0.65 as the threshold) through a procedure that maximizes differences in survival distributions. We used Kaplan–Meier survival curves, with time to death censored at the end of follow-up as the outcome.

**Statistics**. Z-score analysis was performed in MATLAB (MathWorks, Natick, MA). For each variable, the Z-score was defined as,

$$z = \frac{\bar{x} - \overline{x_c}}{\sigma_c} \quad (4)$$

where $\bar{x}$ is the mean of the variable, $\overline{x_c}$ is the mean of the variable in control tissues, and $\sigma_c$ is the standard deviation in controls. PCA and hierarchical clustering were performed in R version 3.3.3. Other statistical analyses were performed using Graphpad Prism versions 6 and 9 (GraphPad Software Inc., La Jolla, CA). Normal distribution of the data was examined using the D'Agostino–Pearson omnibus, Shapiro–Wilk, and Kolmogorov–Smirnov tests. The statistical tests used are indicated in the figure legends. All statistical tests were two-sided. For survival analysis, comparisons between Kaplan–Meier survival curves were performed using log-rank (Mantel–Cox) and hazard ratio tests.

**Reporting summary**. Further information on research design is available in the Nature Research Reporting Summary linked to this article.

## Data availability

The data generated in this study are available within the Article and Supplementary Information. Full microscopy image data sets are available upon request. Source data are provided with this paper.

## Code availability

The code used to perform the bioluminescence data analysis in Fig. 9e–i of the article is available at https://github.com/pierrepudlo/SASmacrophage.

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

## Acknowledgements

We thank Marie Falque (Institut de Biologie du Développement de Marseille, IBDM), Anaïs Bellon (IBDM), and Dolores Barea (Centre de Recherche en Cancérologie de Marseille) for their substantial help with the experiments, François Michel (INMED Imaging facility, InMAGIC, Marseille) for his help with LSFM, and Pierre Cordelier (Centre de Recherche en Cancérologie de Toulouse) for providing the R211-Luc cells. This work was supported by Centre National de la Recherche Scientifique (CNRS), France; Aix Marseille Université, France; grants from Fondation ARC (PJA 20151203159) and INSERM (Program HTE PITCHER 201609) to F.M.; grants from FRM (AJE20150633331), ANR (ANR-16-ACHN-0011) and A*midex Chaire d'excellence to S.vdP.; grants from INCa (PLBIO15-217), Fondation ARC (PJA 20181208127) and INSERM (BBG/2017) to F.G.; grant from the Novo Nordisk Foundation (NNF14OC0008781) to A.E.; grant from INCa, la Ligue contre le cancer and Fondation ARC (PAIR Pancreas 186738) to F.M., R.T., S.P., F.H., and J-Y.S. The France-BioImaging infrastructure is supported by the Agence Nationale de la Recherche (ANR-10-INBS-04-01, 'Investissements d'Avenir'). This work has received support from the French government under the Programme Investissements d'Avenir, Initiative d'Excellence d'Aix-Marseille Université via A*Midex funding (AMX-19-IET-004), and ANR (ANR-17-EURE-0029). J.G. is a recipient of doctoral fellowships from MESRI and from Fondation ARC; A.L. is a recipient of a doctoral fellowship from MESRI; H.T.T.N. is a recipient of a Program 911- VIED doctoral fellowship.

## Author contributions

Acquisition of data: J.G., C.D., A.L., H.T.T.N., A.P., and S.C. Analysis and interpretation of the data: J.G., C.D., A.L., T-T-H.N., P.P., F.H., J-Y.S., R.T., S.C., and F.M. Technical and material support: J.N., F.G., M.B., N.D., A.E., T.L., J-Y.S., S.vdP., R.T, M.H., M.S., M.R-S., J.L., and J.P. Study supervision: S.C. and F.M.

## Competing interests

The authors declare no competing interests.
