## [Peer Review File · Nature Communications]

Reviewers' Comments:

Reviewer #1:

Remarks to the Author:

This is a well-designed study examining sympathetic innervation in pancreatic cancer using a well-established mouse model with orthotopic tumors. The authors are to be commended for an elegant study yet two aspects appear missing. Discussion of Kevin Tracey's anti-inflammatory pathway involved the sympathetic system yet this is missing. Furthermore, a better discussion of TH+ staining in mouse pancreas using the recent studies from the lab of Tony Tang are also missing and would broaden the introduction as well as discussion. The image analysis section is very well detailed and easy to follow and the statistics appear appropriate.

It would be important to better emphasize the intimate relationship between TH+ axons and the vasculature. As example, the two TH+ branches described in the first results section mirror the two major arteries of the mouse pancreas.

Review of reference 16 shows some (7%) of neurons are TH+ in T8 DRG. This does not rule out possibility of TH+ sensory neurons to pancreas.

Remodeling- the authors do not discuss chronic pancreatitis yet such areas are expected in the mouse model and could result in "areas of hyperinnervation". Discussion of the literature in human chronic pancreatitis could be expanded.

Figure 2. The "hot spot" of sympathetic innervation shown in 2A may indicate uneven penetration of primary/secondary antibody. How was possible factor accounted for? In 2 I-J- the spherical regions could be islets with high endocrine cell autofluorescence picked up in the TH+ channel. Beautiful evidence of TH+ and insulin endocrine cell staining is provided. In the synaptophysin stained sections, it will also label endocrine cells. The unevenness of labeling is also appreciated in the movie such that selection of such hot spots would introduce bias. It may be that the inflammation associated with PanIN progression might result in altered (increased) diffusion of antibodies into these regions in comparison to controls.

Fig. 3 is difficult to interpret as one would expect a nearly continuum of staining within duct cells. Though labeled as Duct, the legend indicates cavities. Do the authors mean duct lumen and how were these cavities discerned? It would be convincing to show regions of the main pancreatic duct in the control mice with similar "cavities" if representing duct lumen(s). Direct labeling of the duct epithelium as with CK19 would add greater specificity to these regions of interest and excellent examples are provided in the supplementary figure panels.

Fig. 4 shows expected distribution and numbers of TH+ islet cells in A-C. However the panel in D-F is less convincing regarding two single B-cells. These cells could also represent single cells endocrine cells within the duct lining, rather than remnants of islets as the authors propose, and another marker such as GCG or CK19 would be needed. Finally, panel G shows unclear TH+ staining that may be background and confirms the presence of inflammatory CD45+ cells in the mouse pancreas but little else. The PDX tumors shown from 2 patients show reduced intratumoral TH+ with location restricted to the periphery. In patients 1 and 2, it appears that the GI tract is shown for extratumoral TH (myenteric plexus patterns) and if so, would not be considered extratumoral per se. The conclusion that the tumors stimulated growth and recruitment of pre-existing sympathetic axons from surrounding tissues might be equally claimed to represent ingrowth of vessels and accompanying sympathetic innervation, eg a secondary effect rather than primary.

Line 444- the authors propose sympathetic afferents yet the efferents would be expected to mainly release of NE at terminals.

Minor-

Verify that numbers of animals are included in methods or figure legends.

Include more details on whole-mount immunostaining including if cardiac perfusion, duration of antibody incubations, temperature.

Include objective(s) and details for LSM.

Reviewer #2:

Remarks to the Author:

In this study, Guillot et. al, investigate innervation of pancreatic tumors (pancreatic ductal adenocarcinoma or PDAC) using optical clearing and 3D light-sheet microscopy. This study is

timely in that it adds to the emerging appreciation/interest in the contribution of peripheral innervation to tumor initiation and progression. The data are overall of good quality, and the use of 3D imaging provides insight into regional differences in innervation patterns that are only possible with organ-wide imaging. The authors suggest that enhanced sympathetic innervation in PDAC comes from local sprouting of collaterals from existing axon shafts, and do not represent “new” axon growth as suggested by previous studies, or increased neurogenesis, as suggested by a recent intriguing report. Importantly, the authors report that sympathetic nerves protect against PDAC, such that chemical sympathectomy results in worse outcomes including tumor growth, enhanced macrophage accumulation in lesion sites, and overall poor survival in mice. This is a key finding that contrasts with several previous studies that support a “tumorigenic” role for nerves in cancer progression, including pancreatic tumorigenesis (Renz et. al, 2018, also see review by Zahalka and Frenette, 2020). The authors’ conclusion in this study is generally well-supported by their data, and they suggest that a critical difference between their study and that of Renz et. al is the timing of when nerves were ablated-before disease onset (as in this study) or after tumor establishment (Renz et. al). Given the significance of their finding and to resolve the contradictions in the field, it is incumbent on the authors to provide more clarity as to how their experimental paradigms could contribute to discrepant findings from the literature. See details below

1. The finding in this study that sympathectomy worsens the disease outcome in PDAC contradicts previous work where sympathetic nerves and adrenergic signaling were shown to promote tumor progression (Renz et al, 2018). The authors’ explanation that the inconsistency could be due to differences in the timing of when denervation was accomplished seems reasonable. However, the inconsistencies will add to the confusion in the field. Resolving the anti- versus pro-tumor role of innervation in PDAC is important given several studies supporting that increased nerve density correlates consistently with worse prognosis in many cancers (see review by Zahalka and Frenette, 2020). Short of asking the authors to perform late sympathectomy themselves, I would suggest that the authors clarify with more details of how their experimental paradigm differs from that of Renz et al to contribute to the different results. For example, please clarify with more details this sentence in the Discussion “This inconsistency may be explained by the time point at which denervation was performed—before disease onset in the present study and after tumor establishment in the other case.” When was denervation done in both studies? How is “disease onset” assessed?

The authors also state that Renz et. al, performed surgical nerve ablation which would remove sympathetic and sensory nerves, while the authors performed chemical sympathectomy. Given previous results that sensory nerves exert pro-tumor effects, the authors suggest that loss of sensory nerves in Renz study might partially explain the different results in the two studies. However, in Figure S4, the authors also perform surgical ablation of pancreatic nerves (which would remove both sympathetic and sensory nerves), and arrive at a different conclusion than the Renz study, i.e. surgical ablation results in poor survival outcomes and metastasis, similar to their findings with chemical sympathectomy. How do the authors reconcile the different effects of surgical nerve ablation on PDAC in both studies? Overall, it is incumbent on the authors to provide better clarity about differences between their findings and existing literature.

2. The 3D imaging analyses are elegant and provide more information about innervation of peripheral targets than imaging tissue sections as in previous studies. However, in my opinion, the authors have not conclusively demonstrated that enhanced innervation in tumors arises solely from axon collaterals. More rigorous retrograde tracing analyses using dual color labeling of terminals and co-localization in single soma or single neuron tracing are necessary to provide support for the collateral theory. In the absence of more experimental support, the authors should tone down statements that claim that axonogenesis is NOT involved, for example “These nerves have been engulfed by the tumor and their presence is therefore not the result of an active process of nerve growth and plasticity.” It is likely that increased nerve density is the result of new nerve growth and local sprouting at terminals.

3. The findings with CD163+ macrophages are interesting. However, it is surprising that depletion of such a small population (10% of all tumor-associated macrophages or TAMs, according to the authors) has such profound effects on survival outcomes in sympathectomized mice. The authors should include more controls to make the point that the CD163+ population is selectively ablated

and that other TAMs remain unaffected (the staining for F4/80 macrophages in Figure S5 is too diffuse to support the statement that other TAMs are unaffected by the chemotoxic agent).

Minor points:

1. Several graphs are missing error bars (Fig 5 I, J; Fig 6G; Fig S4Q, R). The graph in Fig S4T is missing the legend for y-axis
2. How was the tumor growth rate measured in Fig. S4T?

Reviewer #3:

Remarks to the Author:

The manuscript by Guillot J et al reports on studies on the changes that occur in sympathetic nerves in a mouse model of pancreatic cancer. The strength of this work from France is the great attention to identifying sympathetic nerves by immunostaining with detailed low power, 3D views of sympathetic nerve fiber bundles. They also undertake several studies to examine the possible function of sympathetic nerves by carrying out denervation experiments, both chemical (6-OHDA treatment) and surgical sympathectomies in their mouse model, both done at early time points (3-4 weeks of age, before PanIN lesions develop). Interestingly, they find that sympathectomy leads to increased liver metastases and decreased survival in this animal model. The effect seem to be due in part to increased intra-tumoral CD163+ macrophages.

The findings are provocative, in part because they contradict many previous studies that have shown pro-tumorigenic effects of the sympathetic nervous system, in pancreatic cancer as well as most other tumor types (e.g. prostate cancer, breast cancer, ovarian cancer, etc.). However, that said, the study would be worth publishing in some venue if the conclusions were more carefully worded, the limitations of the study noted, and the unique aspects of the model discussed a bit more. Indeed, the data do have some value, but the authors have a worrisome tendency to draw broad and sweeping conclusions based on their data, most of which are not justified. In the end, I think that they can only reach conclusions about their specific KIC model, and not pancreatic cancer more generally. Below we have outlined the major concerns.

1. Axonogenesis or neurogenesis in pancreatic cancer. The first point that the authors seem to be trying to make here is against the concept of axonogenesis or neurogenesis. The main finding here is that when comparing the pattern of TH+ nerves in 8 week old WT mice or their KIC mice, the major nerve pattern and distribution looks the same with only a few hot spots. This suggests to them that there is not much nerve growth into the tumors and it is more likely that the tumor simply grows around the pre-existing nerve bundles. While there certainly could be some truth in the notion that PDAC co-opts existing nerve fibers, conclusions about new axonogenesis or neurogenesis do not seem justified. It has been well established that human pancreatic cancer is associated with both a marked increase in nerve number, density and size (He D et al, Human Pathol, 2016; Ceyhan GO et al, Gastro 2009; Demir IE et al, Front Physiol 2012). The changes are most marked in and around pancreatic tumor sites but also fairly diffuse through much of the pancreas. In addition, changes in nerve density and size have been noted in the standard KPC model of pancreatic cancer (Renz B et al, Cancer Cell 2018), and effects of pancreatic cancer cells on the outgrowth of neurons in vitro have been documented by a number of investigators. So there is little doubt that pancreatic cancer, more so than most cancers, is able to induce axonogenesis in vitro and an expansion of neurites is observed in PDAC in vivo.

2. KIC mice. So, that brings the question back to the model that the investigators are currently studied. While the KPC mouse, developed by Hingorani and Tuveson, appears to phenocopy the axonogenesis phenotype seen in PDAC patients, one has to question whether the DePinho KIC model is the right model for addressing this question. In contrast to other well done studies, this group did not start off showing that the KIC mouse resembles human PDAC in the hypertrophy and increased nerve density that is typically found in this tumor type. This mouse model has perhaps been less studied than the KPC model, in part because the extreme rapidity of cancer development

makes it difficult to carry out therapeutic studies. Human pancreatic cancer and associated axonogenesis develop over a decade or more, and the KPC mouse, while still not a perfect model, develops cancer over a period of 6-8 months on average. Thus, given the extreme speed (<8-9 weeks) that cancer develops in KIC mice, and the very slow pace at which new axonal processes are typically able to sprout and grow, the short time course (e.g. 3-4 weeks) of cancer growth in this genetically engineered mouse would seem to preclude it as a model for the study of remodeling of the tumor microenvironment. Most well done studies on nerves and pancreatic cancer have employed multiple model systems to address a specific hypothesis, but a major weakness of this study is the complete reliance on this unusual KIC model. If they would like to reach broader conclusions about sympathetic nerves in pancreatic cancer, they would need to carry out their studies in several other mouse models, particularly KPC mice. Alternatively, perhaps the conclusion here should be that the particular genetic background of these tumors (Kras G12D, p16 null) has allowed tumor evolution to occur in the absence of strong sympathetic input. This might be worth investigating further.

3. Tumor growing around nerves rather than nerves growing towards and into tumors. This is still a very confusing point the authors are trying to make here. Since there is a strong association between the location of nerves and cancer cells in a pancreatic resection specimen, then the authors must be suggesting that tumors are preferentially arising in close proximity to nerves. I suppose this could be tested in their study – but looking at the co-localization between clusters of cancer cells or tumors and the larger nerve fibers. Are the cancer lesions located more often close to, rather than distant from, the major nerves? If the pancreas is divided at baseline into areas of high, medium and low nerve density, are the cancer lesions most often arising in areas of high nerve density? Indeed, the relationship between TH nerves and pancreatic cancer is hard for the reader to assess from the data presented. The authors appear to love their images of “solvent-cleared tissues” that mostly leave the nerves behind, removing the epithelium. However, the problem is that dysplastic and neoplastic lesions are not evident, and it is not even possible from the data presented to conclude that there is any pancreatic cancer present in the model. One would have expected to see some high power histopathology images, illustrating the frequent presence of nerves in PanIN3 and PDAC lesions. In any case, the authors seem to be asking us to accept their conclusions that the “nerve becomes embedded in the tumor as it develops” mostly on faith, which is a bit lacking here.

4. In Figure 3, the authors do a bit more detailed analysis of nerve changes in their model. This is perhaps the strongest part of the study, as most of the first two figures lack any sort of quantitation and are not highly convincing. Here, despite the limitations of their model, the investigators are able to conclude that in the area of “non-invasive pancreatic neoplastic lesions” the axons show increased density, with more (smaller) axon branches, suggesting some sprouting or axonogenesis. They suggest that the innervation of PDAC occurs through “collateral axon sprouting”, and in Fig. 4 they suggest that it is via the growth of “pre-existing axon terminals” rather than new axons. While all of this is fine, and it is agreed the notion of DCX progenitor cells from the brain invading the pancreas seems unlikely, what is missing here is any analysis of the ganglia that are the origin of the nerves innervating the pancreas. Is there any change in the celiac ganglia in this model? The development of cancer is often associated with enlargement of adjacent ganglia, and while a demonstration neurogenesis is challenging, the expansion of ganglia might suggest more than just sprouting of existing axons.

5. Sympathectomy accelerates tumor growth. The authors carry out a number of studies – chemical sympathectomy (with 6-OHDA) surgical sympathectomy in young mice, and a syngeneic model (cell line from KIC tumor) to investigate the effects of sympathetic signaling on tumor growth. The results do point to a consistent effect of sympathectomy in their KIC model, but there are a number of limitations to their data. First, the only clearly statistically significant effect on survival was in the mice given 6-OHDA at 3-4 weeks of age (although no p value is given, but I am guessing the log-rank and hazard ratio are significant). However, 6-OHDA depletes sympathetic signaling from not only pancreatic sympathetic nerves but throughout the body, and could lead to activation of other reflexes and off-target effects. The surgical sympathectomy is a stronger study but apparently showed no statistically significant differences. Finally, the syngeneic cancer cell would have been a better study but unfortunately utilized a KIC (PK4A-Luc) cell line and once again show more qualitative data with not statistically significant differences. Thus, while the

trends are consistent, the lack of statistical significance in most of the experiments is a concern, and the findings must be interpreted cautiously.

6. The data does suggest that this KIC model responds differently to sympathetic signaling compared to many/most other mouse models, it raises a number of mechanistic questions that are never really pursued or even discussed. First, do the authors believe that sympathetic nerves are modulating PDAC growth here through the release of norepinephrine or not. Adrenergic signaling has been shown nearly uniformly, including in PDAC, to promote growth, but presumably the sympathetic nervous system may influence tumor growth in this model in other ways. If so, do the authors believe that it is signaling directly to tumor cells or indirectly to the immune system. If to the immune system, then why would surgical sympathectomy to pancreatic nerves modulate macrophages in this model? What receptors are mediating the response? Are beta adrenergic receptors expressed in this particular KIC tumor in the same pattern as in most other PDAC model systems? Indeed, beta-adrenergic signaling (isoproterenol) has been shown to markedly accelerate pancreatic cancer growth in a PDX1-KrasG12D pancreas, so do the authors must believe that sympathetic nerves have an effect independent from beta-adrenergic signaling or just a unique effect in the KIC model. Finally, the data fall far short of providing any relevance to human disease, and perhaps an important experiment would be to carry out studies similar to the PK4A-Luc cell study (orthotopic injection or better, metastatic model with splenic injection) but instead with human pancreatic cancer cell lines, and combine with chemical sympathectomy.

We thank all the reviewer for their positive assessment of our manuscript and constructive remarks.

Reviewer #1:

1. This is a well-designed study examining sympathetic innervation in pancreatic cancer using a well-established mouse model with orthotopic tumors. The authors are to be commended for an elegant study yet two aspects appear missing. Discussion of Kevin Tracey's anti-inflammatory pathway involved the sympathetic system yet this is missing. Furthermore, a better discussion of TH+ staining in mouse pancreas using the recent studies from the lab of Tony Tang are also missing and would broaden the introduction as well as discussion. The image analysis section is very well detailed and easy to follow and the statistics appear appropriate.

Kevin Tracey's work on the "cholinergic anti-inflammatory pathway" has shown a role of cholinergic fibers of the vagus nerve and splenic adrenergic nerve in inhibiting cytokine production by spleen macrophages. The involvement of this pathway in models of acute pancreatitis was recently investigated and the results showed that while stimulation of the vagus nerve reduced inflammation and injury in the pancreas, this protective effect was independent of the spleen (Zang et al., *Front. Immunol.*, 2021). Thus, a contribution of the cholinergic anti-inflammatory pathway in pancreatic inflammation and the development of PDAC seems to us too speculative at the moment to be discussed in this article.

As requested by the reviewer, Tony Tang's data are now mentioned in the result section of the revised article and compared to our whole organ data.

2. It would be important to better emphasize the intimate relationship between TH+ axons and the vasculature. As example, the two TH+ branches described in the first results section mirror the two major arteries of the mouse pancreas.

To further examine the interactions between the intrapancreatic sympathetic nerves and the vascular system, we have performed immunofluorescent co-labeling with anti-TH and alpha-SMA (marker of vascular smooth muscle cells) antibodies on whole pancreases. The data confirmed that sympathetic nerves entering the pancreas run longitudinally along the two main branches of the celiac trunk that supply the pancreas: the hepatic artery, which gives rise to the superior pancreaticoduodenal arteries as it enters the head of the pancreas, and the splenic artery, which branches to form the dorsal pancreatic artery that supplies the tail of the pancreas. These data have been added in the revised manuscript (new Fig 1D-F).

3. Review of reference 16 shows some (7%) of neurons are TH+ in T8 DRG. This does not rule out possibility of TH+ sensory neurons to pancreas.

The reviewer raises an important point. The somatic DRG neurons expressing TH are C-fiber low threshold mechanoreceptors (C-LTMRs) that innervate hairy skin of the trunk and limbs, where their terminals associate with a subtype of "zigzag" hair follicles (Li et al. *Cell*, 2011). To determine directly whether visceral DRGs innervating the pancreas also express TH, we labelled sensory afferent neurons by retrograde transport of cholera toxin B subunit (CTB) injected into the pancreas. In pancreatic DRGs distributed between thoracic level 9 (T9) and T12 (see Fasanella et al., *J Comp Neurol*, 2008), none of the CTB+ neurons expressed TH. The pancreas is also innervated by vagal sensory neurons located in the nodose ganglia. Similarly, none of the CTB+ sensory neurons in the nodose ganglion were TH+. As a control, we confirmed that all CTB+ neurons in the celiac-superior mesenteric ganglion complex were TH+. These data are shown in new Fig. 1A-C and confirm that the TH+ axons observed in the pancreas are exclusively sympathetic inputs.

4. Remodeling- the authors do not discuss chronic pancreatitis yet such areas are expected in the mouse model and could result in "areas of hyperinnervation". Discussion of the literature in human chronic pancreatitis could be expanded.

To our knowledge, of the articles that describe neural remodeling in human pancreatitis, only one (Ceyhan et al., *Am. J. Gastroenterol.*, 2009) has specifically studied sympathetic innervation. This work reported changes in the

quality of intrapancreatic nerves, with lower sympathetic fiber content in nerves showing signs of damage (neuritis). Sympathetic axon terminals within the pancreatic tissue were not examined in this study. However, we share with this reviewer the idea that sympathetic sprouting and local hyperinnervation of ADM lesions could take place in chronic pancreatitis and this is now mentioned in the discussion of the revised manuscript.

5. Figure 2. The “hot spot” of sympathetic innervation shown in 2A may indicate uneven penetration of primary/secondary antibody. How was possible factor accounted for? In 2 I-J- the spherical regions could be islets with high endocrine cell autofluorescence picked up in the TH+ channel. Beautiful evidence of TH+ and insulin endocrine cell staining is provided. In the synapthophysin stained sections, it will also label endocrine cells. The unevenness of labeling is also appreciated in the movie such that selection of such hot spots would introduce bias. It may be that the inflammation associated with PanIN progression might result in altered (increased) diffusion of antibodies into these regions in comparison to controls.

To test whether inflammation could affect antibody penetration into tissues, we have measured the signal-to-background ratios (SBR) of the anti-TH antibody labeling across an optical tissue section. The results give an identical SBR in a "hot spot" and in the adjacent tissue, indicating a homogenous antibody penetration (see below).

Figure for the reviewer: Optical section through a KIC pancreas stained for TH (green) and CK19 (blue). The plots give the TH signal intensity values along the lines drawn across a control region (ROI 1) and a “hotspot” (ROI 2) using Image J. SBR was calculated as the ratio of the signal intensity (max intensity – background) to the standard deviation of the background ROI.

The autofluorescent signal of the spherical "hot spot" shown in Fig 2K indicates the presence of a PanIN. To confirm this, we performed triple staining of the KIC pancreas with anti-TH, anti-insulin, and anti-CK19 antibodies. The hot spot corresponded to a CK19+ epithelial lesion that can be distinguished from a duct by its size (about 100 µm diameter) and its alveolar morphology seen in 3D (see below this reviewer's response to

point# 6). No insulin+ or TH+ cell bodies were detected in this structure, which excludes that it could be an islet. Moreover, the reconstruction of the TH labeling in this "hot spot" reveals a network of branched fibers and does not allow the identification of cell bodies. The new data are shown in Fig 3L-O. We hope that this will convincingly demonstrate the hypoinnervation of PanIN lesions.

6. Fig. 3 is difficult to interpret as one would expect a nearly continuum of staining within duct cells. Though labeled as Duct, the legend indicates cavities. Do the authors mean duct lumen and how were these cavities discerned? It would be convincing to show regions of the main pancreatic duct in the control mice with similar "cavities" if representing duct lumen(s). Direct labeling of the duct epithelium as with CK19 would add greater specificity to these regions of interest and excellent examples are provided in the supplementary figure panels.

The structures shown in blue on the new Fig. 4 (previously Fig. 3) are indeed the cavities (= lumen) of the neoplastic "duct-like" structures present in KIC pancreas. As requested by the reviewer, we have now performed anti-CK19 staining to label the major pancreatic ducts in control mice and the "duct-like" structures in KIC mice. The 3D images shown in the new Fig. S4A-F confirm the continuum of CK19 staining between normal ducts and "duct-like" structures and further shows that these neoplastic epithelial structures are not tubular but rather have an alveolar morphology that distinguishes them from pancreatic ducts. Thus, because "duct-like" structures are not true ducts, we have changed this name to "epithelial lesions" in the legend of Fig. 4, S3, S4, and S5.

The new Fig. S4G-H also shows how the lumen of the CK19+ epithelial lesions were segmented for 3D reconstruction based on the autofluorescence signal of the tissue. These reconstructions allow us to subtract the empty spaces from the total volume of the tissue when calculating nerve/vessel density.

7. Fig. 4 shows expected distribution and numbers of TH+ islet cells in A-C. However, the panel in D-F is less convincing regarding two single B-cells. These cells could also represent single cells endocrine cells within the duct lining, rather than remnants of islets as the authors propose, and another marker such as GCG or CK19 would be needed. Finally, panel G shows unclear TH+ staining that may be background and confirms the presence of inflammatory CD45+ cells in the mouse pancreas but little else.

We agree with the reviewer that the beta-cells shown in Fig. 5D-F (previously Fig. 4D-F) could represent endocrine cells neogenerated from duct-lining progenitors. However, the question of the origin of the TH+ endocrine cells in PDAC (pre-existing islets or neogenesis), although very interesting, would require long investigations to answer it correctly. We believe that the current data allow us to conclude with certainty, that the TH+ cell bodies observed in PDACs are not neurons, but endocrine cells. We have nevertheless changed the sentence " Only rare scattered TH+ cell bodies were found in PDAC regions; these cells also expressed insulin and probably corresponded to islet remnants destroyed by invasive tumor growth " to " Only rare scattered TH+ cell bodies were found in PDAC regions, which also corresponded to insulin-expressing endocrine cells".

Fig. 5G-I (previously Fig. 4G-I) is aimed at showing that DCX+ cells in PDACs are not neural precursors or immature TH+ neurons (as proposed in prostate cancer), but correspond to hematopoietic cells. We have changed the text to make this point clearer. The image field in panel 5G includes a TH+ nerve fiber as a positive control. We agree that the TH staining is not easily visible. The revised Fig. 5G now has an insert showing the stained axon at a higher magnification.

8. The PDX tumors shown from 2 patients show reduced intratumoral TH+ with location restricted to the periphery. In patients 1 and 2, it appears that the GI tract is shown for extratumoral TH (myenteric plexus patterns) and if so, would not be considered extratumoral per se. The conclusion that the tumors stimulated growth and recruitment of pre-existing sympathetic axons from surrounding tissues might be equally claimed to represent ingrowth of vessels and accompanying sympathetic innervation, eg a secondary effect rather than primary.

Our data on autochthonous murine tumors show that sympathetic nerve fibers do not align along blood vessels in tumors, unlike in healthy tissue, suggesting that the two systems remodel independently of each other. Although some degree of angiogenesis and tissue perfusion may be necessary to sustain axonal regrowth, we found that vascular density is decreased in sympathectomized tumors, suggesting that nerve fibers may regulate vessel growth (and not the reverse). This is also suggested by previous work in prostate tumor (Zahalka et al.,

Science 2017). Thus, it seems unlikely that the presence of TH+ fibers in PDX tumors is solely a "secondary effect" resulting from neovascularization of these tumors.

We have revised the text by changing the sentence "(...) distal extensions of fibers that innervated the adjacent regions of the murine tissue" to "(...) distal extensions of fibers that innervated the adjacent gastrointestinal tract".

9. Line 444- the authors propose sympathetic afferents yet the efferents would be expected to mainly release of NE at terminals.

We have corrected this typo and changed "afferent" to "efferent".

10. Minor- Verify that numbers of animals are included in methods or figure legends.

All animal numbers are now included in the figure legends and method sections.

11. Include more details on whole-mount immunostaining including if cardiac perfusion, duration of antibody incubations, temperature. Include objective(s) and details for LSM.

We have added this information in the methods section.

Reviewer #2 (Remarks to the Author):

In this study, Guillot et. al, investigate innervation of pancreatic tumors (pancreatic ductal adenocarcinoma or PDAC) using optical clearing and 3D light-sheet microscopy. This study is timely in that it adds to the emerging appreciation/interest in the contribution of peripheral innervation to tumor initiation and progression. The data are overall of good quality, and the use of 3D imaging provides insight into regional differences in innervation patterns that are only possible with organ-wide imaging. The authors suggest that enhanced sympathetic innervation in PDAC comes from local sprouting of collaterals from existing axon shafts, and do not represent "new" axon growth as suggested by previous studies, or increased neurogenesis, as suggested by a recent intriguing report. Importantly, the authors report that sympathetic nerves protect against PDAC, such that chemical sympathectomy results in worse outcomes including tumor growth, enhanced macrophage accumulation in lesion sites, and overall poor survival in mice. This is a key finding that contrasts with several previous studies that support a "tumorigenic" role for nerves in cancer progression, including pancreatic tumorigenesis (Renz et. al, 2018, also see review by Zahalka and Frenette, 2020). The authors' conclusion in this study is generally well-supported by their data, and they suggest that a critical difference between their study and that of Renz et. al is the timing of when nerves were ablated-before disease onset (as in this study) or after tumor establishment (Renz et. al). Given the significance of their finding and to resolve the contradictions in the field, it is incumbent on the authors to provide more clarity as to how their experimental paradigms could contribute to discrepant findings from the literature. See details below

1. The finding in this study that sympathectomy worsens the disease outcome in PDAC contradicts previous work where sympathetic nerves and adrenergic signaling were shown to promote tumor progression (Renz et al, 2018). The authors' explanation that the inconsistency could be due to differences in the timing of when denervation was accomplished seems reasonable. However, the inconsistencies will add to the confusion in the field. Resolving the anti- versus pro-tumor role of innervation in PDAC is important given several studies supporting that increased nerve density correlates consistently with worse prognosis in many cancers (see review by Zahalka and Frenette, 2020). Short of asking the authors to perform late sympathectomy themselves, I would suggest that the authors clarify with more details of how their experimental paradigm differs from that of Renz et al to contribute to the different results. For example, please clarify with more details this sentence in the Discussion "This inconsistency may be explained by the time point at which denervation was performed—before disease onset in the present study and after tumor establishment in the other case." When was denervation done in both studies? How is "disease onset" assessed?

This point has been clarified in the revised text. In the present study, denervation was performed at 3.5-4 weeks of age before invasive tumor formation (that occurs by 6 weeks in KIC mice; Aguire et al., 2003), whereas Renz et al. performed denervation on animals whose tumors had reached a size of 20-60 mm³.

The authors also state that Renz et. al, performed surgical nerve ablation which would remove sympathetic and sensory nerves, while the authors performed chemical sympathectomy. Given previous results that sensory nerves exert pro-tumor effects, the authors suggest that loss of sensory nerves in Renz study might partially explain the different results in the two studies. However, in Figure S4, the authors also perform surgical ablation of pancreatic nerves (which would remove both sympathetic and sensory nerves), and arrive at a different conclusion than the Renz study, i.e. surgical ablation results in poor survival outcomes and metastasis, similar to their findings with chemical sympathectomy. How do the authors reconcile the different effects of surgical nerve ablation on PDAC in both studies? Overall, it is incumbent on the authors to provide better clarity about differences between their findings and existing literature.

We fully understand the reviewer's concern. Indeed, surgical sympathetic denervation of the pancreas consists in resecting mixed nerves, containing both sympathetic and sensory neurons, the latter having a pro-oncogenic function in PDAC. Sympathetic and sensory fibers are not present in the same proportions in the pancreas. Indeed, the exocrine pancreas initially receives dense sympathetic innervation, whereas sensory neurons are restricted around the large blood vessels and islets but are absent from the acinar tissue (Lindsay et al., Neuroscience, 2006). On the other hand, it has been reported that the proportions of autonomic and sensory fibers in pancreatic nerves switch during the progression of human pancreatic cancer (Demir et al., Nat. Rev. Gastroenterol. Hepatol., 2015). It is therefore likely that sympathetic and sensory fibers contribute differentially to the effects of surgical denervation performed before or after the development of a PDAC tumor.

According to this hypothesis, the sympathetic nervous system would have a dominant function in the early stages of PDAC development. In line with this, we have shown here that selective ablation of sympathetic neurons by 6-OHDA mimics the effects of early surgical denervation. The opposite effect of late denervation could then result from the elimination of the sensory component of the pancreatic nerves, which may have infiltrated the growing tumor. However, as the sensory innervation patterns have not been fully characterized yet in Kras mouse models, further work will be required to validate this model and understand how the pancreatic neuro-environment might evolve from antitumoral to protumoral during PDAC development.

We hope that the revised discussion will clarify this point.

2. The 3D imaging analyses are elegant and provide more information about innervation of peripheral targets than imaging tissue sections as in previous studies. However, in my opinion, the authors have not conclusively demonstrated that enhanced innervation in tumors arises solely from axon collaterals. More rigorous retrograde tracing analyses using dual color labeling of terminals and co-localization in single soma or single neuron tracing are necessary to provide support for the collateral theory.

Our data indicate increased terminal branching of sympathetic nerve fibers in noninvasive neoplastic lesions such as PanIN, and in peripheral region of PDAC tumors. However, due to the small size of PanIN structures (around 100 μm) and axon branches measured (25-40 μm), the suggested dual color retrograde tracing experiment is not feasible. Not only performing two injections in such close proximity is a challenge, but we do not have any marker to target the areas of interest in the pancreas of living animals. Nevertheless, we believe that the quantification of axonal branch size and density shown in Fig 5 and Supplementary Table 2, as well as the new Fig 3 showing hyperinnervation of PanIN lesions, convincingly support the idea of local regrowth and branching of sympathetic axons.

In the absence of more experimental support, the authors should tone down statements that claim that axonogenesis is NOT involved for example "These nerves have been engulfed by the tumor and their presence is therefore not the result of an active process of nerve growth and plasticity." It is likely that increased nerve density is the result of new nerve growth and local sprouting at terminals.

To address this issue, we have now reconstructed and quantified the TH⁺ nerve trunks in the entire pancreas of 8-weeks-old WT or KIC mice. We have found many sympathetic nerves scattered within KIC tumors (new Fig. 2C-G), confirming previous observations in human and mouse pancreatic cancer. When comparing the amount of intra- and extra-tumoral nerves, we found that the majority of KIC samples (3/4) had a higher proportion of sympathetic nerves in the tumor than in the adjacent tissue (new Fig. 2H). If these intra-tumoral nerves result from new nerve growth, one would expect the adjacent tissue to contain as many nerves as a healthy pancreas. However, we observed a significant decrease in the amount of extra-tumoral nerves compared to controls. In

fact, we found that the total amount (intra-tumoral + extra-tumoral) of TH⁺ nerves in KIC pancreases was similar to that of the total amount of TH⁺ nerves in wild-type control tissues (new Fig. 2I), indicating an overall conservation of nerves. We believe that these data further support the idea that sympathetic nerves in PDACs are not new structures that developed in the tumor, but rather correspond to preexisting nerve bundles that become embedded in the tumor during its development.

3. The findings with CD163+ macrophages are interesting. However, it is surprising that depletion of such a small population (10% of all tumor-associated macrophages or TAMs, according to the authors) has such profound effects on survival outcomes in sympathectomized mice. The authors should include more controls to make the point that the CD163+ population is selectively ablated and that other TAMs remain unaffected (the staining for F4/80 macrophages in Figure S5 is too diffuse to support the statement that other TAMs are unaffected by the chemotoxic agent).

Selective ablation of CD163+ macrophages has already been performed in other cancer models by two authors of this article, A. Etzerodt and T. Lawrence, who previously performed several control experiments to show that nanoparticles are specific and only deplete CD163+ TAMs (see Etzerodt et al., J. Exp. Med., 2019, and Etzerodt et al., J. Exp. Med., 2020). We have now added these references in the revised article. These articles previously provided evidence that targeting the minor subset of TAMs expressing CD163 is sufficient to induce tumor regression in two different tumor models.

The reviewer points out the diffuse appearance of TAMs labeling in Fig S9 (previously Fig. S5), which could be an obstacle to their accurate quantification. In contrast, we find that the inserts in Fig S9A-B show sharp F4/80 labeling that clearly delineates individual cells. The cell density being nevertheless high, the quantification of TAMs was performed on these images by measuring the F4/80 staining using Image J (contrary to the regions of lower density, e.g. PanIN, where we performed a cell count). Quantification of F4/80 fluorescence showed no difference between tumors that received control and doxorubicin-loaded nanoparticles, while the same method revealed a specific deletion of CD163+ TAMs (Fig. S9C-E).

We hope that these explanations will convince of the specific targeting and functional importance of CD163+ TAMs.

Minor points:

1. Several graphs are missing error bars (Fig 5 I, J; Fig 6G; Fig S4Q, R). The graph in Fig S4T is missing the legend for y-axis

The mentioned graphs do not have error bars because they represent the percentage of mice with distant liver/peritoneal metastases. To avoid confusion, we have now represented the data as pie charts (new Fig. 7I-J; Fig. 8G; Fig S7Q-R). The graph in Figure S4T has been replaced by the new Fig. 9, which provide a different representation of the bioluminescence analyses (see below response to point #2).

2. How was the tumor growth rate measured in Fig. S4T?

We have now performed additional grafting experiments to address points # 2 and 7 raised by the Reviewer 3, and we have better described the statistical method used to analyze tumor growth rate. Briefly, for each mouse, the bioluminescence measures with respect to time was fitted to a Gompertz curve. Two main parameters of the Gompertz equation are: $\log(b)$ (logarithm of the maximum bioluminescence value, or plateau), and a (speed at which the logarithm of the growth curve reaches its maximum value). To compare the sympathectomized mice (6-OHDA) with the control group (AA), a Bayesian hierarchical model was designed to estimate the mean values of $\log(b)$ and a . The new Fig. 9 illustrates these different steps of the analysis. Fig. 9D-E show examples of Gompertz growth curves fitted to bioluminescence measurements from representative AA and 6-OHDA treated mice. Fig. 9F-H show the 2D posterior probability distributions for the $\log(b)$ and a parameters in the AA and 6-OHDA-treated groups.

Reviewer #3 (Remarks to the Author):

The manuscript by Guillot J et al reports on studies on the changes that occur in sympathetic nerves in a mouse model of pancreatic cancer. The strength of this work from France is the great attention to identifying sympathetic nerves by immunostaining with detailed low power, 3D views of sympathetic nerve fiber bundles.

They also undertake several studies to examine the possible function of sympathetic nerves by carrying out denervation experiments, both chemical (6-OHDA treatment) and surgical sympathectomies in their mouse model, both done at early time points (3-4 weeks of age, before PanIN lesions develop). Interestingly, they find that sympathectomy leads to increased liver metastases and decreased survival in this animal model. The effect seem to be due in part to increased intra-tumoral CD163+ macrophages.

The findings are provocative, in part because they contradict many previous studies that have shown pro-tumorigenic effects of the sympathetic nervous system, in pancreatic cancer as well as most other tumor types (e.g. prostate cancer, breast cancer, ovarian cancer, etc.). However, that said, the study would be worth publishing in some venue if the conclusions were more carefully worded, the limitations of the study noted, and the unique aspects of the model discussed a bit more. Indeed, the data do have some value, but the authors have a worrisome tendency to draw broad and sweeping conclusions based on their data, most of which are not justified. In the end, I think that they can only reach conclusions about their specific KIC model, and not pancreatic cancer more generally. Below we have outlined the major concerns.

1. Axonogenesis or neurogenesis in pancreatic cancer. The first point that the authors seem to be trying to make here is against the concept of axonogenesis or neurogenesis. The main finding here is that when comparing the pattern of TH+ nerves in 8 week old WT mice or their KIC mice, the major nerve pattern and distribution looks the same with only a few hot spots. This suggests to them that there is not much nerve growth into the tumors and it is more likely that the tumor simply grows around the pre-existing nerve bundles. While there certainly could be some truth in the notion that PDAC co-opts existing nerve fibers, conclusions about new axonogenesis or neurogenesis do not seem justified. It has been well established that human pancreatic cancer is associated with both a marked increase in nerve number, density and size (He D et al, Human Pathol, 2016; Ceyhan GO et al, Gastro 2009; Demir IE et al, Front Physiol 2012). The changes are most marked in and round pancreatic tumor sites but also fairly diffuse through much of the pancreas. In addition, changes in nerve density and size have been noted in the standard KPC model of pancreatic cancer (Renz B et al, Cancer Cell 2018), and effects of pancreatic cancer cells on the outgrowth of neurons in vitro have been documented by a number of investigators. So there is little doubt that pancreatic cancer, more so than most cancers, is able to induce axonogenesis in vitro and an expansion of neurites is observed in PDAC in vivo.

We agree with this reviewer that increased nerve number, density, and size have been frequently reported in human pancreatic cancer. However, it is important to note that most studies, including two of the papers cited by this reviewer (He D et al, Human Pathol, 2016 and Ceyhan GO et al, Gastro 2009), used the pan-neuronal marker PGP9.5 to label intra-tumoral nerves, which does not allow for specific conclusions to be drawn regarding sympathetic innervation of pancreatic cancers. In contrast, in the cited review article (Demir IE et al, Front Physiol 2012) there is mention of a study by Ceyhan et al (American journal of Gastroenterology, 2009) on the neuron-type composition of pancreatic nerves analyzed using different markers, including TH. This work revealed that, despite the increase in nerve size, the amount of sympathetic fibers in these nerves was significantly reduced in human chronic pancreatitis and pancreatic cancer. We have reexamined the literature and found no evidence from human studies that pancreatic cancer is associated with increased sympathetic nerves.

The reviewer also mentioned that an increase in nerve density and size was observed in the standard KPC mouse model of pancreatic cancer (Renz B et al, Cancer Cell 2018). Again, most of the data in this study are based on immunostaining with the PNS neuronal marker peripherin. Only Fig 2E-2F of the Renz article show images and quantification of the high density of TH+ nerves in a KPC tumor, compared to a section of WT tissue almost entirely devoid of nerves. Although interesting, we believe that this result should be taken with caution. Indeed, in the Renz study, the quantification of the nerve surface was performed on 5 sections 200 μ m apart (see Materials and Methods). Thus, only a small region of the pancreas was analyzed, which does not take into account the heterogeneous distribution of sympathetic nerves within the organ. Indeed, in the revised version of this article, we have quantified sympathetic nerve density in equidistant sections spanning the entire pancreas and showed a progressive decrease in TH+ nerve density from the point of nerve entry into the pancreatic head to the distal tail regions (new Fig. 1G-H). Therefore, quantifying pancreatic nerves on tissue section is subjected to important bias depending on the level of sections within the organ and/or tumor location. To overcome this problem, we have now reconstructed and quantified TH+ nerves in the entire pancreas of 8-weeks-old WT or KIC mice. We have found many sympathetic nerves scattered within KIC tumors (new Fig. 2C-G). When comparing the amount of intra- and extra-tumoral nerves, we found that the majority of KIC samples (3/4) had a higher

proportion of sympathetic nerves in the tumor than in the adjacent tissue (new Fig. 2H). If these intra-tumoral nerves result from new nerve growth, one would expect the adjacent tissue to contain as many nerves as a healthy pancreas. However, we observed a significant decrease in the amount of extra-tumoral nerves compared to controls. In fact, we found that the total amount (intra-tumoral + extra-tumoral) of TH⁺ nerves in KIC pancreases was similar to that of the total amount of TH⁺ nerves in wild-type control tissues (new Fig. 2I), indicating an overall conservation of nerves. We hope that these new data will convince the reviewer that high intra-tumor nerve density alone is not a sufficient criterion to conclude that new nerves have been generated and grown in the tumor.

Finally, another point raised by this reviewer is the fact that pancreatic cancer cells have been reported by several investigators to stimulate the outgrowth of neurons in vitro. We would like to draw attention to the fact that all these experiments have been done using sensory neurons of the dorsal root ganglia (DRG) or enteric neurons of the myenteric plexus (He et al. 2016, Secq et al., 2015, Bressy et al., 2017, Wang et al., 2014, Renz et al., 2018, Griffin et al. 2020, Pagella et al. 2020). To our knowledge, no in vitro study has demonstrated a promoting effect of pancreatic cancer cells on the growth of sympathetic neurons in general, and of neurons of the celiac-superior mesenteric ganglia in particular.

In summary, we respectfully disagree with the reviewer's assertions about the ability of pancreatic cancers to induce sympathetic nerve expansion in vitro and in vivo. In addition, we hope that the new data included in the revised article convincingly demonstrate that the intra-tumoral nerves in KIC pancreas are pre-existing structures and not neoformations.

2. KIC mice. So, that brings the question back to the model that the investigators are currently studied. While the KPC mouse, developed by Hingorani and Tuveson, appears to phenocopy the axonogenesis phenotype seen in PDAC patients, one has to question whether the DePinho KIC model is the right model for addressing this question. In contrast to other well done studies, this group did not start off showing that the KIC mouse resembles human PDAC in the hypertrophy and increased nerve density that is typically found in this tumor type. This mouse model has perhaps been less studied than the KPC model, in part because the extreme rapidity of cancer development makes it difficult to carry out therapeutic studies. Human pancreatic cancer and associated axonogenesis develop over a decade or more, and the KPC mouse, while still not a perfect model, develops cancer over a period of 6-8 months on average. Thus, given the extreme speed (<8-9 weeks) that cancer develops in KIC mice, and the very slow pace at which new axonal processes are typically able to sprout and grow, the short time course (e.g. 3-4 weeks) of cancer growth in this genetically engineered mouse would seem to preclude it as a model for the study of remodeling of the tumor microenvironment. Most well done studies on nerves and pancreatic cancer have employed multiple model systems to address a specific hypothesis, but a major weakness of this study is the complete reliance on this unusual KIC model. If they would like to reach broader conclusions about sympathetic nerves in pancreatic cancer, they would need to carry out their studies in several other mouse models, particularly KPC mice. Alternatively, perhaps the conclusion here should be that the particular genetic background of these tumors (Kras G12D, p16 null) has allowed tumor evolution to occur in the absence of strong sympathetic input. This might be worth investigating further.

As discussed in response to point #1 raised by this reviewer, we do not believe that the currently published literature supports the conclusion that, with respect to the sympathetic nervous system, KPC mice better reproduce the axonogenesis phenotype of human PDACs than KIC mice. Indeed, Ceyhan et al (American journal of Gastroenterology, 2009) reported a reduction in sympathetic nerve fiber content in human pancreatic cancer, while Renz et al reported instead an increase in TH⁺ nerve density in KPC tumors. Nevertheless, and as noted above, these changes in nerve density and their significance must be interpreted with caution.

In this paper, we have analyzed the local remodeling of sympathetic nerve endings in both KIC and KPC mice. Our results showed broadly similar changes between the two models, indicating that despite their rapid development KIC tumors induce significant changes in sympathetic innervation to a similar extent as observed in KPC mice. During the revisions of this paper, we performed additional experiments using the R211-Luc cell line derived from a primary KPC tumor (Thibault et al., EMBO Mol Med, 2011) grafted into the pancreas of 6-OHDA-lesioned or control syngeneic mice. As with KIC tumors, we observed that sympathectomy accelerated the growth of R211-Luc KPC tumors (new Fig. 9). Taken together, these data do not support the idea that KIC tumors

would have the singular ability to grow in the absence of sympathetic input, and instead suggest that the sympathetic system contributes similarly to the control of KIC and KPC tumors.

3. Tumor growing around nerves rather than nerves growing towards and into tumors. This is still a very confusing point the authors are trying to make here. Since there is a strong association between the location of nerves and cancer cells in a pancreatic resection specimen, then the authors must be suggesting that tumors are preferentially arising in close proximity to nerves. I suppose this could be tested in their study – but looking at the co-localization between clusters of cancer cells or tumors and the larger nerve fibers. Are the cancer lesions located more often close to, rather than distant from, the major nerves? If the pancreas is divided at baseline into areas of high, medium and low nerve density, are the cancer lesions most often arising in areas of high nerve density? Indeed, the relationship between TH nerves and pancreatic cancer is hard for the reader to assess from the data presented. The authors appear to love their images of “solvent-cleared tissues” that mostly leave the nerves behind, removing the epithelium. However, the problem is that dysplastic and neoplastic lesions are not evident, and it is not even possible from the data presented to conclude that there is any pancreatic cancer present in the model. One would have expected to see some high power histopathology images, illustrating the frequent presence of nerves in PanIN3 and PDAC lesions. In any case, the authors seem to be asking us to accept their conclusions that the “nerve becomes embedded in the tumor as it develops” mostly on faith, which is a bit lacking here.

As mentioned above (see response to point#1), we have now quantified TH+ nerves in the entire pancreas of WT and KIC mice and found similar amounts of nerves in the two conditions, despite the obvious presence of nerves in all tumors examined (new Fig. 2). As requested by the reviewer, both nerves and epithelium were immunostained in these specimens to help the reader appreciate the location of tumor nodules in KIC tissues (new Fig. 2D-E). We think that these data strongly argue against the growth of new sympathetic nerve trunks inside tumors.

Although we have not suggested that tumors preferentially arise near nerves, it is known that the head of the pancreas, which receives the largest number of nerve fibers, is the most common tumor site for pancreatic cancer (D'Haese et al. 2014). This location could explain the frequent presence of nerves in human PDAC tissue. This point is now mentioned in the discussion of the article.

4. In Figure 3, the authors do a bit more detailed analysis of nerve changes in their model. This is perhaps the strongest part of the study, as most of the first two figures lack any sort of quantitation and are not highly convincing. Here, despite the limitations of their model, the investigators are able to conclude that in the area of “non-invasive pancreatic neoplastic lesions” the axons show increased density, with more (smaller) axon branches, suggesting some sprouting or axonogenesis. They suggest that the innervation of PDAC occurs through “collateral axon sprouting”, and in Fig. 4 they suggest that it is via the growth of “pre-existing axon terminals” rather than new axons. While all of this is fine, and it is agreed the notion of DCX progenitor cells from the brain invading the pancreas seems unlikely, what is missing here is any analysis of the ganglia that are the origin of the nerves innervating the pancreas. Is there any change in the celiac ganglia in this model? The development of cancer is often associated with enlargement of adjacent ganglia, and while a demonstration neurogenesis is challenging, the expansion of ganglia might suggest more than just sprouting of existing axons.

We have addressed the reviewer’s concern in two ways. First, we have imaged in 3D the whole coeliac-superior mesenteric ganglion complex of control and KIC mice. Volume measurement did not reveal significant changes between the healthy and tumor conditions (new Fig. 5J-L). Next, we have performed 5-ethynyl-2'-deoxyuridine (EdU) injections at early stages of neoplastic transformation (4, 5 and 5.5 weeks) to label proliferating cells in the coeliac-superior mesenteric complex. The same amount of EDU+ cells was observed in WT and KIC mice (new Fig. 5M-S). Furthermore, identification of EDU+ cells showed that they are endothelial cells (Pecam+), fibroblasts (Vimentin+) or glial cells (Sox10+), suggesting that this proliferation is due to normal tissue turnover (new Fig. 5S). In contrast, no TH+ sympathetic neurons had incorporated EdU even in KIC mice (new Fig. 5M-R). Taken together, these new data make it unlikely that neurogenesis occurs in the coeliac-superior mesenteric ganglion complex of KIC mice.

5. Sympathectomy accelerates tumor growth. The authors carry out a number of studies – chemical sympathectomy (with 6-OHDA) surgical sympathectomy in young mice, and a syngeneic model (cell line from KIC tumor) to investigate the effects of sympathetic signaling on tumor growth. The results do point to a

consistent effect of sympathectomy in their KIC model, but there a number of limitations to their data. First, the only clearly statistically significant effect on survival was in the mice given 6-OHDA at 3-4 weeks of age (although no p value is given, but I am guessing the log-rank and hazard ratio are significant). However, 6-OHDA depletes sympathetic signaling from not only pancreatic sympathetic nerves but throughout the body, and could lead to activation of other reflexes and off-target effects. The surgical sympathectomy is a stronger study but apparently showed no statistically significant differences. Finally, the syngeneic cancer cell would have been a better study but unfortunately utilized a KIC (PK4A-Luc) cell line and once again show more qualitative data with not statistically significant differences. Thus, while the trends are consistent, the lack of statistical significance in most of the experiments is a concern, and the findings must be interpreted cautiously.

The log-rank test is the standard statistical test for comparing two survival curves. The log-rank test provides a p-value and, to avoid confusion, we have replaced "Log-rank: xxxx" with "p=xxxx (log-rank test)" in all figure legends. Following the general convention, we used 0.05 as the significance level. Therefore, our results were statistically significant for both 6-OHDA-treated (p= 0.0132; Fig. 7B (previously Fig. 5B) and surgically sympathectomized mice (p=0.0109; Fig. S7L (previously Fig. S4L)).

Regarding the orthotopic grafting models, we have now performed additional experiments to address points # 2 and # 7 raised by this reviewer, and we have better described the statistical method of analysis. Briefly, for each mouse, the bioluminescence measures with respect to time were fitted to a Gompertz curve. Two main parameters of the Gompertz equation are: $\log(b)$ (logarithm of the maximum bioluminescence value, or "plateau"), and a (speed at which the logarithm of the growth curve reaches its maximum value). To compare the sympathectomized mice (6-OHDA) with the control group (AA), a Bayesian hierarchical model was designed to estimate the mean values of $\log(b)$ and a . Unlike frequentist statistics, Bayesian statistics do not provide p-value but a posterior probability value that tells us the degree of confidence in the estimation. A probability > 0.90 is considered as an acceptable measure of confidence.

The new Fig. 9 illustrates these different steps of the analysis. Fig. 9D-E show examples of Gompertz growth curves fitted to bioluminescence measurements from representative AA and 6-OHDA treated mice. Fig. 9F-H show the 2D posterior probability distributions for the $\log(b)$ and a parameters in the AA and 6-OHDA-treated groups. In syngeneic transplantations of PK4A-Luc and R211-Luc cells, the two groups appear clearly separated, with a notable higher plateau in the 6-OHDA-treated groups, supported by a high posterior probability value for $\log(b)$ (PK4A-Luc: $\log(b)=0.956$; R211-Luc: $\log(b)=0.996$) and a (R211-Luc: $a=0.969$) (Fig. 9F-G). In contrast, when R211-Luc cells were transplanted into immunodeficient athymic nude mice, the AA and 6-OHDA groups were no longer separated and low posterior probability values were obtained for $\log(b)$ (0.765) and a (0.857) parameters, thus indicating that the tumoral response to sympathectomy requires an intact host immune system (see response to point #7 below). We hope that these clarifications will convince the reviewer of the robustness of our analyses, which were performed by experts in mathematical modeling of cancer and Bayesian statistics, both co-authors of this article (F. Hubert and P. Pudlo, Institute of Mathematics of Marseille).

6. The data does suggest that this KIC model responds differently to sympathetic signaling compared to many/most other mouse models, it raises a number of mechanistic questions that are never really pursued or even discussed.

First, do the authors believe that sympathetic nerves are modulating PDAC growth here through the release of norepinephrine or not.

Adrenergic signaling has been shown nearly uniformly, including in PDAC, to promote growth, but presumably the sympathetic nervous system may influence tumor growth in this model in other ways.

If so, do the authors believe that it is signaling directly to tumor cells or indirectly to the immune system.

If to the immune system, then why would surgical sympathectomy to pancreatic nerves modulate macrophages in this model?

What receptors are mediating the response?

Are beta adrenergic receptors expressed in this particular KIC tumor in the same pattern as in most other PDAC model systems?

Indeed, beta-adrenergic signaling (isoproterenol) has been shown to markedly accelerate pancreatic cancer growth in a PDX1-KrasG12D pancreas, so do the authors must believe that sympathetic nerves have an effect independent from beta-adrenergic signaling or just a unique effect in the KIC model.

Here, the reviewer raises the point that ADRB2 signaling has been shown to promote PDAC growth, as demonstrated, for example, by Renz et al. (2018). This study reported that chronic restraint stress promotes *Kras*-induced pancreatic tumorigenesis through elevation of circulating adrenal-derived catecholamines (epinephrine and, to a lesser extent, other catecholamines such as norepinephrine) and stimulation of ADRB2-dependent pancreatic epithelial growth. The authors also explored the function of local norepinephrine delivery by pancreatic sympathetic nerves in a “neural model for catecholamine delivery during in vitro ADM”. However, it is important to highlight that this model, adapted from Ceyhan et al. (2008), used sensory DRG neurons. If a small percentage of DRG neurons are TH⁺, they lack dopamine beta-hydroxylase (DBH), the enzyme that converts dopamine into norepinephrine, and therefore do not produce norepinephrine (Sapio et al., *Front Neurosci.* 2020). Moreover, we have now shown by retrograde tracing (new Fig. 1C) that TH⁺ DRG neurons do not project their axons into the pancreas. Thus, whether and how local norepinephrine supply from sympathetic axons regulates PDAC growth remains unknown, even in conventional PDX1-*Kras*G12D models. This is now better explained in the introduction of the revised article.

In order to explore the role of norepinephrine in our model, we have now tested the in vitro activity of norepinephrine on PK4A-Luc pancreatic cancer cells that exhibit increased orthotopic tumor growth in sympathectomized mice. At all concentrations tested, norepinephrine had no effect on cell growth assessed for 96 h with the IncuCyte system. These new data are shown in Fig. 9B of the revised manuscript and suggest that sympathetic neurons do not affect tumor growth through direct signaling to cancer cells.

We have next tested whether norepinephrine could suppress CD163 expression in macrophages. This has required the development of an appropriate in vitro model. Indeed, CD163 mRNA is not expressed by most human and mouse macrophage cell lines (Buechler et al., *Journal of Leukocyte Biology*, 2000), but some investigators have reported upregulation of CD163 in the RAW 264.7 monocyte/macrophage cell line after treatment with anti-inflammatory cytokines (de Araujo Junior, *Mol Cell Biochem*, 2020). We repeated these experiments but did not detect CD163 mRNA by qRT-PCR. Next, CD163 mRNA was investigated in mouse bone marrow-derived monocytes, but the very low expression level of CD163 we observed was a concern when we tried to detect a decreased CD163 expression. Finally, we set up cultures of FACS-sorted CD45⁺/CD163⁺/F4/80⁺ mouse pancreatic macrophages (new Fig. S8M,N). The new Fig. 10H shows that incubation with norepinephrine induced a significant decrease in CD163 mRNA expression, suggesting a direct effect of norepinephrine in inhibiting the pro-tumor phenotype of pancreatic macrophages.

Finally, the data fall far short of providing any relevance to human disease, and perhaps an important experiment would be to carry out studies similar to the PK4A-Luc cell study (orthotopic injection or better, metastatic model with splenic injection) but instead with human pancreatic cancer cell lines, and combine with chemical sympathectomy.

We agree with the reviewer that the proposed experiments with human pancreatic cell lines would be important to extend our current results. However, xenograft experiments must be performed with immunodeficient mice, in which the tumor response to sympathectomy could be compromised given the potential importance of the immune system in this response. To directly test this point, we repeated the orthotopic transplantation experiments of the KPC-derived R211 murine cancer cells, this time using athymic nude mice instead of syngeneic mice. The results are shown in Fig. 9H and confirmed that the promoting effect of sympathectomy on PDAC growth requires an intact host immune system, which prevented us from performing the requested xenograft experiments with human cell lines.

To further explore the role of the immune system in our model, we used Nanostring’s GeoMx Digital Spatial Profiling (DSP) to characterize the immune landscape of control and sympathectomized KIC tumors. The results confirmed the increase in CD163 at the tumor periphery. CD163⁺ TAMs can have suppressive effects on tumor-infiltrating T cells (Etzerodt et al., *J. Exp. Med.*, 2020). While markers of T-cell infiltration (e.g., CD3, CD8, CD4 and forkhead box P3 (FoxP3)) did not vary between sympathectomized and control tumors, the immune checkpoint CTLA4 (cytotoxic T-lymphocyte-associated antigen 4) emerged as the most strongly upregulated marker in denervated KIC tumors. Interestingly, we found a positive and significant correlation between CD163 and CTLA4 expression in tumor tissues. These data are shown in Fig. 10A-B of the revised article and provide further insight into the immune mechanisms involved in the tumor response to sympathectomy.

Reviewers' Comments:

Reviewer #1:

Remarks to the Author:

The authors have addressed all concerns and inclusion of additional data expands interest in the article.

Reviewer #2:

Remarks to the Author:

The authors have satisfactorily addressed my concerns with the revised Discussion and additional data.

Reviewer #3:

Remarks to the Author:

The revised manuscript by Guillot J et al continues its focus on sympathetic nerves in pancreatic cancer (PDAC) and has been strengthened a bit with additional data over the last year. While the paper contains some useful data that could provide useful insights regarding the varied roles of the sympathetic nervous system in PDAC, there continues to be major concerns about the broad claims and conclusions based on very limited model systems, which have been minimally changed. The strength of this paper remains the careful histologic analysis of nerve fibers in mice, and the interesting and unexpected finding of increased cancer progression in their KIC mouse model. However, analysis of neural pathways is complicated and the author's desired conclusions from this study and the proposed paradigm ignore the accumulated data on adrenergic signaling, beta-blockers and pancreatic cancer, which has indicated a largely promotional role for adrenergic signaling in PDAC. Below are the major concerns that remain largely unaddressed.

1. KIC mouse model of PDAC. The authors have not provided a sufficient response to the concern about whether the KIC mouse model is an adequate model to evaluate neural remodeling in pancreatic cancer, and their responses seem to have missed the larger point. Remodeling of the tumor microenvironment, and neurogenesis in general, occurs very slowly. Human pancreatic cancer develops over a decade or two, thus allowing for the gradual increase in nerve number, density and size that are routinely seen in human PDAC. KPC tumors can reproduce some of this, although even these tumors develop too quickly (e.g. over 6 months) to perfectly mimic the neural remodeling seen in human tumors. Importantly, such neural remodeling is generally poorly reproduced in syngeneic or orthotopic cancer cell line model systems which have a very short (4-5 week) time course. Thus, it is unlikely that analysis of nerve number or density over a short 6-8 week genetic model of PDAC allows for sufficient time for neural remodeling, based on all that is known about such processes. While the authors claim to have "reproduced" the KIC findings in KPC mice (p. 11), this analysis is not convincing and carried out in KPC mice younger (e.g. 14 week) than typically studied. Notably, while the axonal density was higher in the KPC mice, overall nerve density was not compared or quantified. In their response, the authors have focused mainly on whether an increase in sympathetic nerves has been shown in human or other mouse models, but the question not answered is whether the KIC mouse model reproduces the neural remodeling in human and other models, or whether it is an appropriate model to address this question. Thus, while the authors can and should describe their results and reach conclusions from their interesting KIC mouse model, they must qualify these findings with the phrase "in the KIC mouse model". They should not assume such findings are representative of "pancreatic cancer in general", as they are likely not.

2. PDAC induction of axonogenesis. There is no question that there is an expansion of total nerve density in human PDAC. It is fine for the authors to raise questions about whether TH+ nerves are expanded in human PDAC, but on the other hand, the authors have provided no new human data to address this. In addition, the authors question whether PDAC cells induce sympathetic neuron outgrowth and appropriately note that studies have been done with DRG and not sympathetic ganglia. On the other hand, the authors certainly could have tested this question and performed

such studies themselves, but did not. Thus, they have provided no data that pancreatic cancer cells, and specifically human pancreatic cancer cells known to express neurotrophins, induce or not the outgrowth of sympathetic neurons. It would be surprising if they did not, but would require a slower (>6 month) model of pancreatic tumorigenesis to best study this question.

3. Nerve-cancer crosstalk with KIC tumor cells. Prior studies showed a clear cross talk between nerves and pancreatic cancer cells, both in the KPC model and with human PDAC cells. Importantly it was shown that most PDAC cells overexpress neurotrophins, which are upregulated in response to ADRB2 signaling. Thus, in addition to issues regarding the too rapid time course in KIC mice of tumor development (with insufficient time for neural remodeling), another explanation for the KIC model is that the cells do not express neurotrophins or other axonal guidance molecules, or that they don't express appropriate beta-adrenergic receptors. While the PK4A-Luc cells don't show proliferation in response to epinephrine, this is only one cell line and no positive controls are shown. Multiple studies in the literature have reported proliferative effects by PDAC and other cancer cell lines in response to adrenergic agonists. Furthermore, multiple studies have shown outgrowth of neurites from DRG ganglia. Do the KIC cell lines (e.g. PK4A-Luc) induce an outgrowth similar to that shown for human pancreatic cancer lines? If the authors can show neurite outgrowth from DRGs with PDAC cell lines, but not with celiac or superior mesenteric ganglia, they could certainly make the argument that perhaps sympathetic axons don't expand. Otherwise I think based on the KIC model they have insufficient evidence to make this argument.

4. Pancreatic ganglia and neurogenesis. The authors go on to try to disprove the induction of any neurogenesis or proliferation by neural progenitors in their rapid KIC mouse model. However, their study approach suggests a lack of appreciation of the difficulties of the question and the long history involved in demonstrating neurogenesis, given the very slow normal cell division by neural progenitors in the CNS and enteric nervous system. For decades the concept of neurogenesis was not accepted because CNS progenitors did not label with short-term BrdU. It took decades of careful studies to demonstrate that the brain does in fact contain neural progenitors/stem cells but this is now widely accepted. Many in the field now estimate that neural progenitors divide extremely slowly, perhaps once every 2-3 months. The investigators use an unreferenced protocol, where EdU is given 3 times over 10 days and mice are immediately sacrificed and superior mesenteric ganglia analyzed. This is unfortunately insufficient to label neural progenitors in the CNS or ENS in normal mice. Is there any evidence or rationale to justify such limited labeling approach? Studies of the proliferation by ENS progenitors used the implantation of BrdU pumps over several months, followed by a washout period to identify such progenitors. Thus, the section on p.12 on EdU labeling provides little useful insight and certainly cannot be used as evidence again axonogenesis, and thus should likely be deleted.

5. CD163+ macrophages. The evidence suggests that in the KIC model, sympathetic nerves suppress cancer spread but not through direct contacts with PDAC cells. Instead, studies suggest that this is immune mediated and immunologic analysis appears to point towards changes in this M2 macrophage population with sympathetic denervation, but the mechanism appears obscure. First, where is this occurring? They suggest it occurs mainly in intrapancreatic macrophages, but they do not show proximity of macrophages to TH+ nerves in their model system. In addition, they do not provide convincing data that such macrophages are directly suppressed by sympathetic signaling. The only data provided are some in vitro studies showing that gene expression of CD163 is suppressed by short-term incubation with epinephrine, but the significance of CD163 gene downregulation is unclear and in any case it is a long way from showing that adrenergic signaling reduces the T cell suppressive properties of macrophages. Indeed, abundant published data has shown in many cancer model systems that beta-adrenergic signaling upregulates TGF-beta and other molecules and induces M2 macrophages to become more immunosuppressive. There is no in vivo data here linking adrenergic signaling to the suppression of CD163+ macrophages. Furthermore, how do the authors explain all the beta-blocker studies which have shown a beneficial effect on PDAC? How is it that beta-adrenergic agonists accelerate PDAC, which has been proven through in vivo studies, if at the same time it presumably suppresses CD163+ macrophages.

6. Possibility of systemic effects of interventions through non-beta-adrenergic pathways. While the authors focus on the possible effects in prior studies of surgical sympathetic denervation, and the

possible role of sensory neurons to explain the findings, this explanation is inadequate as it would not account for the beta-blocker and *Adrb2*^{-/-} studies. One unique aspect of this current study is its reliance on the 6-OHDA model which involves systemic sympathetic denervation in a very short-term mouse genetic model that lacks a significantly remodeled TME or neural microenvironment. One possibility is that there is a much stronger role in the KIC mouse model for the alpha-adrenergic system, which was not modulated in most of the earlier studies but is undoubtedly affected by 6-OHDA. Blocking adrenergic signaling in the KIC mouse may possibly have resulted in activation of other feedback pathways and perhaps in the KIC mouse, that may have stimulated PDAC progression. Another possibility is that there may be other off-target effect of 6-OHDA. Prior studies in humans and mice demonstrated quite clearly a strong role of beta-adrenergic signaling and *Adrb2* in promoting pancreatic cancer progression. The investigators here did not investigate any adrenergic receptors and specifically did not test the effects of broad beta-adrenergic agonists such as isoproterenol which was shown in published studies to accelerate pancreatic cancer. Indeed, the absence of any studies documenting adrenergic receptors or signaling, it is hard to understand the reason for the discrepant findings in this study.

7. Overall, the work while comprising some interesting data, falls far short of supporting the broad conclusions proposed. Thus statements in the Abstract, "our findings revealed properties of the sympathetic nervous system in PDAC immunity", and in the Discussion, "The present study revealed a protective role of the sympathetic nervous system against PDAC," are far too broad given that there is no human data to support this, and the study was done entirely with an incredibly fast genetic mouse model that likely does not rely on neural remodeling for its progression.

Manuscript ID: NCOMMS-20-14656
Response to reviewer comments

Reviewer #1:

The authors have addressed all concerns and inclusion of additional data expands interest in the article.

We thank the reviewer for the positive evaluation of our manuscript.

Reviewer #2:

The authors have satisfactorily addressed my concerns with the revised Discussion and additional data.

We thank the reviewer for the positive evaluation of our manuscript.

Reviewer #3:

1. KIC mouse model of PDAC. The authors have not provided a sufficient response to the concern about whether the KIC mouse model is an adequate model to evaluate neural remodeling in pancreatic cancer, and their responses seem to have missed the larger point. Remodeling of the tumor microenvironment, and neurogenesis in general, occurs very slowly. Human pancreatic cancer develops over a decade or two, thus allowing for the gradual increase in nerve number, density and size that are routinely seen in human PDAC. KPC tumors can reproduce some of this, although even these tumors develop too quickly (e.g. over 6 months) to perfectly mimic the neural remodeling seen in human tumors. Importantly, such neural remodeling is generally poorly reproduced in syngeneic or orthotopic cancer cell line model systems which have a very short (4-5 week) time course. Thus, it is unlikely that analysis of nerve number or density over a short 6-8 week genetic model of PDAC allows for sufficient time for neural remodeling, based on all that is known about such processes. While the authors claim to have "reproduced" the KIC findings in KPC mice (p. 11), this analysis is not convincing and carried out in KPC mice younger (e.g. 14 week) than typically studied. Notably, while the axonal density was higher in the KPC mice, overall nerve density was not compared or quantified. In their response, the authors have focused mainly on whether an increase in sympathetic nerves has been shown in human or other mouse models, but the question not answered is whether the KIC mouse model reproduces the neural remodeling in human and other models, or whether it is an appropriate model to address this question. Thus, while the authors can and should describe their results and reach conclusions from their interesting KIC mouse model, they must qualify these findings with the phrase "in the KIC mouse model". They should not assume such findings are representative of "pancreatic cancer in general", as they are likely not.

Our response: The reviewer raises an important point. Peripheral nerve growth in an adult organism is a relatively slow process (about 1-2 mm/day; Höke, JCI, 2011). Thus, in humans, after a distal nerve injury in the forearm, nerve regeneration over several tens of cm usually takes 6 months to 1 year. There are currently no data available on the time of tumor development required for an increase in nerve number and density to be observed in human PDACs.

The average size of human pancreatic tumors at diagnosis is 30 mm (Agarwal et al., Pancreas, 2008), and tumor sizes range from 4 to 20 mm in the KPC and KIC mouse models (our personal observations). Thus, theoretically existing nerves located near a growing tumor (or neural progenitor cells in the tumor) could have formed new nerves infiltrating an entire tumor within a few weeks.

Sprouting at nerve terminals is also a process that develops over a period of weeks after an injury or under inflammatory conditions. For example, sympathetic nerve sprouting and hyperinnervation is typically observed 7 days after injury in mouse models of myocardial infarction (Wernli et al, Basic Research in Cardiology, 2009; Yin et al, J Cell Mol Med, 2017).

Thus, based on what is known about the time required for axonal processes to sprout and grow, the KIC model, despite rapid tumor development (<8-9 weeks), appears sufficiently long and therefore appropriate to study neuronal remodeling. In support of this, we found significant changes in the innervation patterns of KIC tumors that are comparable to those observed in humans and in other models:

1) we reported a high proportion of nerve bundles within KIC tumors, as previously described in the KPC model (Renz et al., 2018) and human tumors.

Our 3D analysis of KIC tumors further showed that the high proportion of intratumoral nerves results from engulfment of preexisting nerves and is not due to total pancreatic nerve expansion, even though new nerves would theoretically have had time to grow into these tumors.

Whether or not this result mimics the mechanisms that lead to increased intratumoral nerve density in human PDAC and other models is not known. Indeed, no previous studies have traced the origin of intratumoral nerves or quantified total nerves volume in diseased pancreases. Thus, the hypothesis that total nerve expansion occurs in slower models of PDAC is interesting, but remains to be formally demonstrated.

2) we described and quantify extensive sprouting and growth of sympathetic axons in pre-neoplastic and early neoplastic lesions both in KIC and in KPC models.

Commenting on these results in his/her previous review, this reviewer wrote: *"This is perhaps the strongest part of the study, as most of the first two figures lack any sort of quantitation and are not highly convincing. Here, despite the limitations of their model, the investigators are able to conclude that in the area of "non-invasive pancreatic neoplastic lesions" the axons show increased density, with more (smaller) axon branches, suggesting some sprouting or axonogenesis."*

In the second review, the reviewer now finds this same analysis in KPC mice "not convincing", without specifying details. He/she points out that we used KPC mice younger (14 weeks) than typically studied, which is not quite exact. Other studies used KPC mice at 10-12 weeks (for analysis of PanIN innervation) and 16 weeks (for PDAC) (Stopczynski et al., Cancer Res, 2014; Sinha et al., Cancer Res., 2017). 14 weeks is therefore an intermediate age at which advanced PanIN lesions and fibrosis were evident, and early PDAC was observed in all animals we analyzed.

Changes made in the revised manuscript: Since we have not quantified overall nerve density in KPC pancreas, we cannot rule out that nerve expansion occurs in this slower or other tumor models. This is now clearly indicated in the discussion of the revised article which states: *"In contrast to this model, the present study showed that wild-type and KIC mouse pancreata contain similar amounts of sympathetic nerves and that nerve trunks found within tumors are pre-existing pancreatic nerves. These nerves have been engulfed by the tumor and their presence is therefore not the result of an active process of nerve growth and plasticity. Our data on KIC mice do not exclude that nerve expansion may occur in the human PDAC or in other mouse models, although de novo nerve formation in PDAC remains to be formally demonstrated."*

2. PDAC induction of axonogenesis. There is no question that there is an expansion of total nerve density in human PDAC.

Our response: We respectfully disagree with this statement which is not supported by any scientific evidence.

Expansion of total nerves is one possible mechanistic explanation for the frequent presence of nerves observed on histological tissue sections of human PDAC. However, an increase in total nerves in diseased pancreas has not been formally demonstrated in humans, nor in animal models. Furthermore, de novo innervation of PDACs is a hypothesis that is not shared by all investigators. For example, a recent study from the Hondermarck laboratory reported: “*The data presented indicate that the number of nerves between [pancreatic cancer] and normal adjacent tissue is the same*”, “*Hence, it is likely that the phenomenon being observed is nerve hypertrophy, not de novo innervation as has been described in other tumors*” (Ferdoushi et al., Scientific reports, 2021).

In PDAC and other cancers, some authors have used GAP43 or NF-H (also known as NF200) as markers of growing neurons to prove that intratumoral nerves are novel structures. However, NF200 and GAP43 are well-established markers of mature sensory neurons, that innervate peripheral organs, including the healthy pancreas (see below Fig. 1 for the reviewer; Gebhart et al., Encyclopedia of Pain. Springer, 2013 ; Su et al., J. Comp. Neurol. 523:1505-1528, 2015). Thus, immunostaining for NF200 or GAP43 can provide information about the identity of intratumoral nerves, but not on their origin (pre-existing versus de novo). There is currently no good marker to distinguish between mature/existing and new/growing axons in pancreatic cancer.

Figure 1 for the reviewer: Fluorescent immunolabeling of GAP43⁺ and NF200⁺ nerve fibers in sections of wild-type adult mouse pancreas.

The alternative possibility that intratumoral nerves are preexisting nerves has been largely ignored so far and may seem counterintuitive since most studies have documented a virtual absence of nerves in sections of healthy pancreas. However, the pancreas contains numerous nerves whose totality cannot be appreciated on tissue microarray or classical histological sections, which represent only a very small volume of the whole organ. Moreover, the distribution of the nerves being heterogeneous in the organ, nerves are not visible at all section levels. In particular, localization analysis of nerves in the mouse pancreas have found a higher density in the head region where PDACs are most frequently formed than in the pancreatic head (Saricaoglu et al., Neurogastroenterol Motil., 2020; and our present study).

In conclusion, our study is the first to quantify total nerves in a whole pancreas and our results do not support total nerve expansion in the KIC mouse model. Although the mechanisms that lead to tumor innervation could be different in human cancers, nerve expansion in human and other PDAC models is a hypothesis that remains to be demonstrated.

It is fine for the authors to raise questions about whether TH⁺ nerves are expanded in human PDAC, but on the other hand, the authors have provided no new human data to address this.

Our response: Data on TH⁺ nerves on human PDAC sections do exist in the literature (Ceyhan et al., Am J Gastroenterol 2009). We are not convinced of the value of replicating or extending such data, since, as discussed above, analysis on TMAs or resected tumor sections does not allow to trace the origin of

intratumoral nerves or to perform a global quantification of all pancreatic nerves. Very recently, a method for quantitative 3D molecular imaging of the entire human pancreas has been published (Hahn et al., Communications biology, 2021) and this method may be applicable to nerve detection. Nevertheless, access to donated pancreases from untreated PDAC patients remains a challenge.

In addition, the authors question whether PDAC cells induce sympathetic neuron outgrowth and appropriately note that studies have been done with DRG and not sympathetic ganglia. On the other hand, the authors certainly could have tested this question and performed such studies themselves, but did not. Thus, they have provided no data that pancreatic cancer cells, and specifically human pancreatic cancer cells known to express neurotrophins, induce or not the outgrowth of sympathetic neurons. It would be surprising if they did not, but would require a slower (>6 month) model of pancreatic tumorigenesis to best study this question.

Our response: In the first review, the reviewer stated that “ (...) *effects of pancreatic cancer cells on the outgrowth of neurons in vitro have been documented by a number of investigators. So there is little doubt that pancreatic cancer, more so than most cancers, is able to induce axonogenesis in vitro and an expansion of neurites is observed in PDAC in vivo*”. We have previously pointed out that the in vitro studies to which the reviewer referred were performed with DRG sensory neurons, but not with sympathetic neurons.

The reviewer now regrets that we did not perform co-culture experiments with sympathetic neurons and cancer cells, although these experiments were not requested in the first revision of the manuscript. In his/her opinion, cancer cells derived from a “slow” PDAC model (i.e., a model assumed by the reviewer to induce de novo nerve formation) should be able to promote the growth of sympathetic axons in vitro.

Although we did not find evidence of de novo nerve formation in the “rapid” KIC model, we observed significant sprouting and growth of sympathetic axons. We also reported sympathetic axon sprouting and infiltration in the “slower” KPC and PDX models. These events certainly require the activity of growth promoting signals for sympathetic axons.

Thus, the requested neuron/cancer cell co-culture experiments will not inform about the ability of growing tumors to induce de novo nerve formation, as opposed to sprouting and growth of existing axon terminals.

The results will indicate whether the signals promoting sympathetic axon growth are produced by the cancer cells themselves. On the other hand, a lack of growth promoting effect by PDAC cells would suggest that other cell types (e.g., fibroblasts, glial cells, immune cells...) regulate sympathetic axon growth, as suggested by findings that axonal remodeling is already evident at preneoplastic lesions. The identification of the cells and molecular mechanisms that regulate sympathetic axon remodeling during PDAC development is an important area of research that we are currently pursuing in the laboratory.

3. Nerve-cancer crosstalk with KIC tumor cells. Prior studies showed a clear cross talk between nerves and pancreatic cancer cells, both in the KPC model and with human PDAC cells. Importantly it was shown that most PDAC cells overexpress neurotrophins, which are upregulated in response to ADRB2 signaling. Thus, in addition to issues regarding the too rapid time course in KIC mice of tumor development (with insufficient time for neural remodeling), another explanation for the KIC model is that the cells do not express neurotrophins or other axonal guidance molecules, or that they don't express appropriate beta-adrenergic receptors.

Our response: This assumption that KIC tumors lack growth signals for sympathetic axons is inconsistent with the sprouting and growth of sympathetic terminals we observed in KIC tumors, and that we also observed in KPC (expressing NGF) and PDX models.

While the PK4A-Luc cells don't show proliferation in response to epinephrine, this is only one cell line and no positive controls are shown.

Multiple studies in the literature have reported proliferative effects by PDAC and other cancer cell lines in response to adrenergic agonists.

Our response : As mentioned by the reviewer, several studies have reported that activation of beta adrenoreceptors with the synthetic agonist isoproterenol promotes PDAC cell proliferation (see for example, Wan et al, Cancer letters, 2016; Askari et al, J Cancer Res Clin Oncol, 2005; Lin et al, Hepatogastroenterology, 2012). In contrast, only a few studies have directly assessed the activity of the endogenous ligand norepinephrine on PDAC cells, and the results are contrasted. Some investigators reported an increase in the proliferation of BxPC-3 and Panc-1 cells in response to high doses (10^{-5} M) of norepinephrine (Qian et al, Oncology report, 2018). Others instead reported a bidirectional effect of norepinephrine on MIA PaCa-2 and BxPC-3 cells, with 10^{-8} M concentration having a trend toward stimulation and 10^{-6} M toward suppression, whereas norepinephrine at 10^{-5} M had significant suppressive effect (Wang et al., Plos One, 2012).

We previously reported a lack of proliferative effect by PK4A-Luc cells in response to 10^{-8} and 10^{-6} M norepinephrine. We have now repeated these experiments using the R211-Luc cancer cell line derived from a primary KPC tumor. Our results indicate a significant suppression of R211-Luc cell growth at a concentration of 10^{-6} M (but no effect at 10^{-8} M). This suppression was blocked by the beta1-blocker atenolol and only partially blocked by the beta2-blocker butoxamine. The observed suppression is nevertheless relatively weak and, while this may contribute in part to the inhibitory activity of the sympathetic nervous system on PDAC development, the lack of response of R211-Luc tumors to sympathectomy in nude mice suggests a greater impact of the sympathetic nervous system on the immune environment than on the tumor cells themselves.

These differences in cellular responses to synthetic/endogenous adrenergic agonists may be due to different selectivity for adrenergic receptors, as well as to a property of GPCRs known as "functional selectivity" (or ligand bias), whereby ligands preferentially activate or inhibit different signaling pathways via the same receptor (reviewed in Evans et al, Br J Pharmacol, 2010). Indeed, it is known that norepinephrine produces only partial and biased beta-2 AR signaling compared to the "full" agonist isoproterenol. (Heubach et al, Mol Pharmacol, 2004; Reiner et al, J Biol Chem, 2010).

Changes made in the revised manuscript: The new data are now presented in Fig. 9B of the revised manuscript, and the results section states: *"Taken together, these data indicate that while direct noradrenergic signaling from the sympathetic nervous system to cancer cells may contribute in part to the inhibitory activity of the sympathetic nervous system on PDAC development, the tumor response to sympathectomy is more likely mediated indirectly by changes in the immune environment."*

Figure 9B: Cell confluence measured using IncuCyte live-cell imaging for R211-Luc cells incubated with 10⁻⁶M norepinephrine (NE), with or without atenolol (At) or butoxamine (buto). Data are presented as mean ± SEM. control versus NE, p<0.001, control versus NE+ buto, p<0.001, NE versus NE+buto, p<0.001 (Two-way ANOVA).

Furthermore, multiple studies have shown outgrowth of neurites from DRG ganglia. Do the KIC cell lines (e.g. PK4A-Luc) induce an outgrowth similar to that shown for human pancreatic cancer lines? If the authors can show neurite outgrowth from DRGs with PDAC cell lines, but not with celiac or superior mesenteric ganglia, they could certainly make the argument that perhaps sympathetic axons don't expand. Otherwise I think based on the KIC model they have insufficient evidence to make this argument.

Our response: As detailed above (see response to point #2), whether or not sympathetic neurons behave similarly to DRG neurons in co-cultured with PDAC cells will not help to clarify the issue of sympathetic nerve expansion in humans and other PDAC models.

4. Pancreatic ganglia and neurogenesis. The authors go on to try to disprove the induction of any neurogenesis or proliferation by neural progenitors in their rapid KIC mouse model. However, their study approach suggests a lack of appreciation of the difficulties of the question and the long history involved in demonstrating neurogenesis, given the very slow normal cell division by neural progenitors in the CNS and enteric nervous system. For decades the concept of neurogenesis was not accepted because CNS progenitors did not label with short-term BrdU. It took decades of careful studies to demonstrate that the brain does in fact contain neural progenitors/stem cells but this is now widely accepted. Many in the field now estimate that neural progenitors divide extremely slowly, perhaps once every 2-3 months. The investigators use an unreferenced protocol, where EdU is given 3 times over 10 days and mice are immediately sacrificed and superior mesenteric ganglia analyzed. This is unfortunately insufficient to label neural progenitors in the CNS or ENS in normal mice. Is there any evidence or rationale to justify such limited labeling approach? Studies of the proliferation by ENS progenitors used the implantation of BrdU pumps over several months, followed by a washout period to identify such progenitors. Thus, the section on p.12 on EdU labeling provides little useful insight and certainly cannot be used as evidence against axonogenesis, and thus should likely be deleted.

Our response: Neurogenesis and expansion of the coeliac ganglia was raised by this reviewer during the first revision as a mechanism that might contribute to the changes in sympathetic innervation observed in 6.5-week-old KIC mice. Because these changes occur over a short period of time (first pancreatic ductal lesions appear around 4 weeks; Aguirre et al., Genes and Dev, 2003), we reasoned that only a "burst of neurogenesis" (as observed for example after traumatic nerve injury) occurring between 4 and 5.5 weeks could lead to hyperinnervation of pancreatic lesions a few days/weeks later. This justified the limited EdU injection protocol we used.

The reviewer now points out that neurogenesis is a very slow process, which in itself seems to rule out an involvement of this mechanism in the rapid KIC model. The mentioned protocol of BrdU infusion over several months to detect proliferation of ENS progenitors cannot be applied to KIC mice.

We have kept the EdU data in this revised version of the paper to show the absence of a "burst of neurogenesis" detected with our labeling protocol that allows the detection of other proliferating cell types (glial, endothelial cells...) during the study period. However, these data can be deleted as suggested by the reviewer.

5. CD163+ macrophages. The evidence suggests that in the KIC model, sympathetic nerves suppress cancer spread but not through direct contacts with PDAC cells. Instead, studies suggest that this is immune mediated and immunologic analysis appears to point towards changes in this M2 macrophage population with sympathetic denervation, but the mechanism appears obscure. First, where is this occurring? They suggest it occurs mainly in intrapancreatic macrophages, but they do not show proximity of macrophages to TH+ nerves in their model system.

Our response: The regions in which changes in CD163⁺ macrophages were identified by DSP are described in the Method section: "*For each section, up to 10 square ROIs of 200 μm sides were drawn at the tumor periphery, defined as the outer 300 μm rim of the tumor area*". Sympathetic fibers infiltrated in PDAC tumors are indeed largely restricted to these peripheral areas. Immunostaining of sympathetic fibers and macrophages (stained for F4/80 and CD163) in ADM and PanIN lesions is shown in Figure S8 and indicates close proximity between sympathetic axons and pancreatic macrophages.

Changes made in the revised manuscript: On panels E and I of Figure S8, we have added inserts to help visualize the proximity of sympathetic fibers and macrophages and added a new supplementary figure (Figure S9) to show this proximity in PDAC tissue.

Supplementary Figure 9. Proximity between macrophages and the sympathetic innervation of PDAC tissues. Immunostaining for F4/80, CD163, and TH in PDAC tissue of a 6.5-week-old KIC mouse. Scale bar= 50 μm.

In addition, they do not provide convincing data that such macrophages are directly suppressed by sympathetic signaling. The only data provided are some in vitro studies showing that gene expression of CD163 is suppressed by short-term incubation with epinephrine, but the significance of CD163 gene downregulation is unclear and in any case it is a long way from showing that adrenergic signaling reduces the T cell suppressive properties of macrophages. Indeed, abundant published data has shown

in many cancer model systems that beta-adrenergic signaling upregulates TGF-beta and other molecules and induces M2 macrophages to become more immunosuppressive. There is no in vivo data here linking adrenergic signaling to the suppression of CD163+ macrophages.

Our response: In the present study, we used two approaches in vivo (digital spatial profiling (DSP) and immunohistochemistry) to demonstrate that deletion of sympathetic adrenergic fibers increased the number of CD163⁺ macrophages in KIC tumors, and we further showed that selective ablation of the CD163⁺ macrophage subset reversed the adverse effect of sympathectomy on survival of KIC mice. In addition, to address a previous question raised by this reviewer regarding the role of NE signaling on immune cells, we provided proof of principle that NE suppresses CD163 expression in isolated pancreatic macrophages. Together our data indicate a role of macrophages as cellular mediators of sympathetic nervous system activity in PDAC.

The reviewer thinks these data are not convincing. The main reason appears to be that our results do not fit with the common view that adrenergic signaling promotes the induction of M2 immunosuppressive macrophages. According to this view, one would expect the interruption of adrenergic signaling to decrease, rather than increase, CD163⁺ macrophages in tumors. However, two recent studies indicated that the beta-blocker propranolol increased CD163 expression in central nervous system macrophages in rodent models of neuroinflammation (Pilipovic et al., *Neurobiol Dis*, 2020; Lin et al., *Cells*, 2020). Thus, there is a converging set of findings that support a suppressive role of adrenergic signaling on CD163⁺ macrophages. However, this does not exclude that adrenergic signaling may have distinct functions on other macrophage subtypes, and indeed our DSP analyses revealed an increase in the MHCII marker (frequently upregulated in M1-type macrophages) in sympathectomized tumors.

The reviewer also pointed out that we did not explore how adrenergic signaling reduces the T cell suppressive properties of macrophages. This was indeed not the objective of this article. The immunosuppressive function of CD163⁺ macrophages has already been studied in great detail in other articles written by co-authors of this study (see for example Etzerodt et al., *J Exp Med*, 2019).

Furthermore, how do the authors explain all the beta-blocker studies which have shown a beneficial effect on PDAC?

Our response: Our results are not inconsistent with the known effects of beta-blocker on PDAC.

Beta-blockers have been shown to have a beneficial effect on PDAC when used in combination with other treatments (e.g., chemotherapy) or in models of chronic stress-induced tumorigenesis, where systemic catecholamine levels are increased (Renz et al., *Cancer Cell*, 2018; Partecke et al., *Pancreatol*, 2016). However, beta-blockers can also exert an opposite effect and abolish the antitumoral effect of the enriched environment ("eustress") on PDAC (Song et al., *Cancer Res.*, 2017). Thus, beta-adrenergic signaling does not always exert beneficial effect on PDAC, but may exert dual effects depending on environmental stressors/conditions.

Furthermore, the link between pharmacological inhibition of adrenergic signaling and tumorigenesis is more complex than a simple consequence of blocking neuroendocrine-released catecholamines. Indeed, the adrenal medulla and sympathetic innervation are not the only source of catecholamines, as PDAC cells also have the capacity to synthesize and release their own catecholamine neurotransmitters (Al-Wadei et al., *Mol Cancer Res.*, 2011). In addition, it is also important to consider that adrenergic receptors are often upregulated in cancers, which may result in enhanced "constitutive" activity even without an activating ligand. Therefore, adrenergic antagonists may mediate their effects not only by preventing binding of endogenous agonists, but also by decreasing the propensity of the receptor to assume an active state (inverse agonism) (reviewed in Berg and

Clarke, IJNP, 2018). This may explain why beta-blockers suppress the in vitro growth of several PDAC cell lines (MIA PaCa-2 and BxPC-3, see Zang et al., Cancer Biology & Therapy, 2010) and human PDAC organoids (Renz et al., Cancer Cell, 2018) in the absence of any exogenously added catecholamines.

Therefore, given the multiple sources of endogenous catecholamines and the mechanisms of action of beta-blockers (antagonist and inverse agonist acting at the same receptor), it is not possible to derive conclusive information on the specific role of sympathetic adrenergic fibers solely from experiments using beta-blockers in vivo.

In the present study we used 6-OHDA to specifically interfere with sympathetic nerve signaling in tumorigenesis. To our knowledge, only one other study has used 6-OHDA in a PDAC model (Song et al., Cancer Res., 2017). The authors reported that *"6OHDA-induced sympathectomy tended to promote the growth of Panc02 tumors in standard environment mice and largely abolished the tumor inhibitory effect of enriched environment, demonstrating a critical role for the sympathetic nervous system (SNS) in enriched environment-induced tumor inhibition"*. Thus, there is no major discrepancy between our data demonstrating a protective function of sympathetic nerve fibers in KIC tumors and the current literature on PDAC.

How is it that beta-adrenergic agonists accelerate PDAC, which has been proven through in vivo studies, if at the same time it presumably suppresses CD163+ macrophages.

Our response: Our results are not inconsistent with the known effect of beta-adrenergic agonists on PDAC.

The stimulatory effect of beta-adrenergic agonists such as isoproterenol on PDAC results from the modulation of various biological processes exerted at different levels: i) on cancer cells, potentially increasing their proliferation, ii) on the tumor microenvironment, where most cells (endothelial cells, fibroblasts, macrophages and other immune cells) express beta-adrenergic receptors, and iii) on the tumor macro-environment. Furthermore, as discussed above (see our response to point #3 raised by this reviewer), the concept of "functional selectivity" in adrenergic signaling systems (in addition to the traditional receptor selectivity) must be taken into account when interpreting the effects of adrenergic agonist drugs. It is known, for example, that norepinephrine causes partial and biased beta2-adrenergic receptor signaling compared to the standard "full" agonist isoproterenol. Thus, the effect of systemic beta-adrenergic receptor activation certainly has a greater, and potentially different, impact on tumor outcome than the effect sympathetic signaling exert on cells located in their immediate vicinity.

6. Possibility of systemic effects of interventions through non-beta-adrenergic pathways. While the authors focus on the possible effects in prior studies of surgical sympathetic denervation, and the possible role of sensory neurons to explain the findings, this explanation is inadequate as it would not account for the beta-blocker and Adrb2-/- studies.

Our response: The Wang laboratory previously showed that treatment with the specific beta2-blocker ICI 118,551 and surgical removing of mixed sympathetic/sensory nerves both inhibit the growth of established PDAC tumors (Renz et al., Cancer Cell, 2018). We proposed that the sensory nervous system may contribute to the observed effects of tumor denervation. As the reviewer points out, a link between sensory neurons and adrenergic signaling is difficult to conceive, nevertheless such a link is clearly suggested by the authors of the study. Indeed, the authors used DRG neurons, instead of sympathetic neurons, for their in vitro nerve-cancer cell co-culture experiments. They wrote *"In this Matrigel-embedded model, adapted from one previously described (Ceyhan et al., 2008), murine embryonic dorsal root ganglia (DRGs) were implanted adjacent to Kras mutant spheres (Figure 3I).*

After 7 days, there were significantly more spheres in the DRG co-cultures than in controls (Figures 3J and 3K). Furthermore, pretreatment of these co-cultures with ICI blocked the increase in sphere number.” These findings linking sensory neurons activity and adrenergic signaling were not discussed further in the article. However, in a recent review article from the Wang laboratory, DRG neurons were schematized as acting on cancer cells via norepinephrine release and activation of the beta-2 adrenoceptor (see the figure below taken from White and Wang, Cell Res, 2021).

Fig. 1: Therapeutic strategies to modulate nerve–cancer crosstalk in PDAC.

Figure 2 for the reviewer: scheme taken from White and Wang, Cell Res, 2021 suggesting a link between sensory neurons and beta-2 AR signaling in PDAC

In the present article, we did not detect the expression of the norepinephrine biosynthetic enzyme tyrosine hydroxylase in sensory afferents of the pancreas. Thus, alternative explanations could be that DRG neurons stimulate autocrine catecholamine production and/or constitutive beta-2 adrenergic receptor activity (via receptor upregulation) in cancer cells. Both of these effects could be blocked by the antagonist/inverse agonist activity of ICI 118,551.

One unique aspect of this current study is its reliance on the 6-OHDA model which involves systemic sympathetic denervation in a very short-term mouse genetic model that lacks a significantly remodeled TME or neural microenvironment.

Our response: The claim that KIC tumors lack a significantly remodeled TME or neural microenvironment is not supported by experimental data: 1) we and others (Renz et al., 2018) have shown that KIC and KPC tumors have a similar neural microenvironment (i.e., high proportion of intratumoral nerves and increased sprouting of individual axon terminals), 2) a recent single-cell RNA seq analysis has revealed similar cancer and stromal cell (immune cells, fibroblasts) evolution and heterogeneity between the Kras mutant models with distinct secondary driver mutations, including the KIC, KPC, and KPFC models (Hosein et al, JCI, 2019).

One possibility is that there is a much stronger role in the KIC mouse model for the alpha-adrenergic system, which was not modulated in most of the earlier studies but is undoubtedly affected by 6-OHDA. Blocking adrenergic signaling in the KIC mouse may possibly have resulted in activation of other feedback pathways and perhaps in the KIC mouse, that may have stimulated PDAC progression.

Our response: This is an interesting point. However, since the role of the alpha-adrenergic system has not been studied in PDAC, it is not possible to say whether it plays a more important/different role in the KIC model compared with others.

Another possibility is that there may be other off-target effect of 6-OHDA.

Our response: Some off-target effect of 6-OHDA cannot be ruled out; however, we have addressed this issue by confirming the effects of systemic 6-OHDA-mediated denervation by surgical sympathectomy performed at the same stage.

Prior studies in humans and mice demonstrated quite clearly a strong role of beta-adrenergic signaling and *Adrb2* in promoting pancreatic cancer progression. The investigators here did not investigate any adrenergic receptors and specifically did not test the effects of broad beta-adrenergic agonists such as isoproterenol which was shown in published studies to accelerate pancreatic cancer. Indeed, the absence of any studies documenting adrenergic receptors or signaling, it is hard to understand the reason for the discrepant findings in this study.

Our response: Apparent discrepancies between our results and effects of beta-adrenergic agonists/antagonists are discussed and reconciled in our response to point # 5 raised by this reviewer.

7. Overall, the work while comprising some interesting data, falls far short of supporting the broad conclusions proposed. Thus statements in the Abstract, "our findings revealed properties of the sympathetic nervous system in PDAC immunity", and in the Discussion, "The present study revealed a protective role of the sympathetic nervous system against PDAC," are far too broad given that there is no human data to support this, and the study was done entirely with an incredibly fast genetic mouse model that likely does not rely on neural remodeling for its progression.

Changes made in the revised manuscript: As requested by the reviewer, the abstract has been modified and now ends with: *"Altogether, our findings revealed new insights into the mechanisms by which the sympathetic nervous system exerts cancer-protective properties in a mouse model of PDAC."* In the discussion, the sentence *"The present study revealed a protective role of the sympathetic nervous system against PDAC"* has been replaced by *"The present study revealed a cancer-protective function of the sympathetic nervous system in the KIC mouse model of PDAC."* Finally, we have throughout the text replaced the general term "PDAC" with "KIC (or KPC) tumor" when necessary to avoid overly broad conclusions.

Reviewers' Comments:

Reviewer #3:

Remarks to the Author:

The revised manuscript by Guillot et al on sympathetic nerves and pancreatic cancer has been improved somewhat in response to the previous critique. Several of the earlier statements have been modified and/or toned down a bit. I think that the overall message of the paper, which is that much of the pancreatic cancer -nerve interactions occur with existing nerves in areas of higher nerve density, is a reasonable conclusion.

However, there are still several areas where this paper is quite confusing and where conclusions are overstated.

First, while tumors grow preferentially in areas where nerves already exist, the question of whether nerves are changed and remodeled remains unclear. The authors do note that there is an active, localized sprouting of nerve terminal that occurs in the setting of PDAC. And the authors do show in Fig. 2 that there is no significant change in the total volume of TH+ nerves in KIC pancreas. However, there are still several issues here. While the authors in their response that the association between PDAC and nerves "is not due to total pancreatic nerve expansion", and that "our results do not support total nerve expansion in the KIC mouse model", I was not able to find a simple graph or figure or statement in the paper showing this. On re-reading this paper, I only found data on "TH+ nerves" and not on total nerve density. This needs to be clarified. Please show and comment on the "total nerve density" if it exists. If the analysis was only done on TH+ nerves, then the manuscript needs to be substantially edited and "nerve" replaced with "TH+ nerve."

In addition, the evidence that there is no increase in TH+ nerve density, while explainable by the very early time points in the study, is also weak. Figure 2I does show a slight increase in total TH staining volume but it is not significant. However, there were only 4 mice in each group which is extremely low. What were the Power Calculations for this study? If one assume a say 10-15% increase in nerve volume or density, what numbers of animals would be needed to show this difference?

Further, it is clear in published images of histologic sections of PDAC mouse models, as well as my own experience looking at human histologic sections, that nerve bundles are often hypertrophied or enlarged in pancreatic cancer. Is this not present in the KIC mouse pancreas? Unfortunately, the authors did not provide any comparable sections from their KIC mouse model for validation. Or are the authors suggesting that similar very large nerve bundles are present in the WT pancreas if one looked in the same region. If so, such data should be provided and this comparison could be shown.

In any case, assuming there is some enlargement of nerve bundles, it becomes confusing as to the mechanism. The authors have acknowledged in the text that the development of PDAC "may involve substantial growth and remodeling of individual nerve fibers". In addition, their data indicate that "early and progressive sprouting and growth of sympathetic axon branches is a common characteristic of PDAC."

For some investigators, this is the definition of axonogenesis (i.e. growth of axons). The investigators, though, would like to distinguish this from "neurogenesis", which might be useful, but the authors fail to really define their terms or distinguish their terms or questions. They should make this distinction of axonogenesis (growth of axons) from neurogenesis (increase in nerve cell bodies) at the start. However, they do not have enough data to comment on neurogenesis.

I think the authors are a bit distracted here by the Magnon DCX paper which has not been widely accepted. Furthermore, they stain only for TH+ cell bodies and do not look broadly at neural progenitors in associated ganglia. Finally, the limitations of their proliferation were previously raised, and again the study focused on TH+ sympathetic neurons.

- I would recommend cutting the sentence on proliferation (lines 258-259) as this is misleading. If you do want to mention the proliferation studies, you would need to state that "EdU was given 3

times over 10 days and did not show a burst of proliferation, but of course this approach does not rule out a slower process of neurogenesis.”

- Furthermore, the concluding sentence “Together, these results rule out a model of neurogenesis in PDAC ...branching of existing axon terminals.” The conclusion is completely unsubstantiated and also in many ways superfluous. The authors should certainly argue that the latter mechanism seems to be predominant in their KIC model without making such a broad and unproven claim.

Another problematic statement is on page 14, lines 309-311. I would suggest that the sentence be modified to state, “Altogether, these results demonstrate that the sympathetic nervous system can exert a protection function during the early development and progression of pancreatic cancer in the KIC mouse.”

Similarly, on page 15, line 340, would change this to “...on PDAC development in the KIC mouse.”

I think the effect of sympathetic nerves on macrophages is interesting, and worth studying further, but it does appear that the effect of nerves is model, stage and context dependent. While the methods used are probably better than some, the spectrum of models is quite limited, and I would still be cautious with any sweeping conclusions.

Reviewer #3

We would like to thank the reviewer for his/her careful consideration of our manuscript and insightful comments.

The revised manuscript by Guillot et al on sympathetic nerves and pancreatic cancer has been improved somewhat in response to the previous critique. Several of the earlier statements have been modified and/or toned down a bit. I think that the overall message of the paper, which is that much of the pancreatic cancer -nerve interactions occur with existing nerves in areas of higher nerve density, is a reasonable conclusion.

However, there are still several areas where this paper is quite confusing and where conclusions are overstated.

First, while tumors grow preferentially in areas where nerves already exist, the question of whether nerves are changed and remodeled remains unclear. The authors do note that there is an active, localized sprouting of nerve terminal that occurs in the setting of PDAC.

And the author do show in Fig. 2 that there is no significant change in the total volume of TH+ nerves in KIC pancreas. However, there are still several issues here. While the authors in their response that the association between PDAC and nerves "is not due to total pancreatic nerve expansion", and that "our results do not support total nerve expansion in the KIC mouse model", I was not able to find a simple graph or figure or statement in the paper showing this. On re-reading this paper, I only found data on "TH+ nerves" and not on total nerve density. This needs to be clarified. Please show and comment on the "total nerve density" if it exists. If the analysis was only done on TH+ nerves, then the manuscripts needs to be substantially edited and "nerve" replaced with "TH+ nerve."

Our response: All the work presented in our article concerns the sympathetic nervous system. We have omitted in some places in the text to recall the nerves studied. As requested by the reviewer, we have modified the manuscript to replace the term "nerve" with "TH+ nerves" when necessary.

In addition, the evidence that there is no increase in TH+ nerve density, while explainable by the very early time points in the study, is also weak. Figure 2I does show a slight increase in total TH staining volume but it is not significant. However, there were only 4 mice in each group which is extremely low. What were the Power Calculations for this study? If one assume a say 10-15% increase in nerve volume or density, what numbers of animals would be needed to show this difference?

Our response: Sample size calculation indicates that we would need 126 mice (n=61 per group) to show a 10-15% increase in nerve volume in the KIC pancreas (with the following parameters: mean TH⁺ nerve volume in WT = $1.67 \times 10^9 \mu\text{m}^3$, SD = $0.29 \times 10^9 \mu\text{m}^3$, alpha = 0.05, power (1-beta) = 0.9).

This sample size not only raises ethical concerns but a 10-15% increase in nerve volume represents an insignificant effect compared to what has been reported in the field. Indeed, given that in 8-week-old KIC mice, tumors occupy about 1/3 of the total organ volume, this difference would correspond to an approximate 30-50% increase in nerve volume or density within PDAC (1.3x or 1.5x more nerves). This is far below the 1000% increase in TH⁺ nerve density (10x more nerves in the tumor) reported previously in 2D analysis. See, for example, Fig.2F of Renz et al. (2018) below.

In this study, sample size (8 mice, n= 4 animals per group) was determined to show an approximately 100% (2x more) increase in sympathetic nerves in KIC tumors with power set at 0.9. A smaller effect is unlikely to be significant in revealing de novo nerve formation, since slight variations in total TH⁺ nerve volume result from the invasion of the extrapancreatic nerve plexus by tumors of the pancreatic head (3/4 cases) and enlargement of intra-tumoral nerves (see below).

Further, it is clear in published images of histologic sections of PDAC mouse models, as well as my own experience looking at human histologic sections, that nerve bundles are often hypertrophied or enlarged in pancreatic cancer. Is this not present in the KIC mouse pancreas? Unfortunately, the authors did not provide any comparable sections from their KIC mouse model for validation. Or are the authors suggesting that similar very large nerve bundles are present in the WT pancreas if one looked in the same region. If so, such data should be provided and this comparison could be shown.

In any case, assuming there is some enlargement of nerve bundles, it becomes confusing as to the mechanism. The authors have acknowledged in the text that the development of PDAC “may involve substantial growth and remodeling of individual nerve fibers”. In addition, their data indicate that “early and progressive sprouting and growth of sympathetic axon branches is a common characteristic of PDAC.”

Our response: The reviewer asks a new question about nerve hypertrophy and the mechanisms underlying this phenomenon, which had not been raised in previous rounds of examination.

Since increased size (hypertrophy) and increased number (de novo innervation/axonogenesis) of nerves have often been reported concomitantly in PDAC, one might think that they are related events. However, the two phenomena can occur independently, and a recent study reported nerve hypertrophy, but not increased nerve number, in human PDAC samples (Ferdoushi et al., Scientific reports, 2021).

The question of nerve hypertrophy has not been studied in our article so far, but we do not dispute it.

Indeed, in our analyses, we observed that the integrity of intratumoral nerves was sometimes altered, with fiber bundles appearing "defasciculated". This phenomenon, which results in increased nerve size, is likely due to neural invasion by malignant cells, which is a common and characteristic feature of PDAC.

For some investigators, this is the definition of axonogenesis (i.e. growth of axons). The investigators, though, would like to distinguish this from “neurogenesis”, which might be useful, but the authors fail to really define their terms or distinguish their terms or questions. They should make this distinction of axonogenesis (growth of axons) from neurogenesis (increase in nerve cell bodies) at the start.

Our response: The question of the relative contribution of axonogenesis and neurogenesis in PDAC is clearly posed at the outset of our article (see Introduction section, pages 5-6, lines 93-106). To ensure that there is no ambiguity in the definition of these terms for the broad readership of Nature

Communications, we propose that the text be modified to include the following definitions, "Nevertheless, the relative contribution of axonogenesis (i.e., **the growth of axons from existing neurons**) and neurogenesis (i.e., **the de novo generation of neuronal cells**) to the neuroplastic changes accompanying the development and progression of PDAC remains to be explored."

However, they do not have enough data to comment on neurogenesis.

I think the authors are a bit distracted here by the Magnon DCX paper which has not been widely accepted. Furthermore, they stain only for TH+ cell bodies and do not look broadly at neural progenitors in associate ganglia.

Finally, the limitations of their proliferation were previously raised, and again the study focused on TH+ sympathetic neurons.

- I would recommend cutting the sentence on proliferation (lines 258-259) as this is misleading. If you do want to mention the proliferation studies, you would need to state that "EdU was given 3 times over 10 days and did not show a burst of proliferation, but of course this approach does not rule out a slower process of neurogenesis."

Our response: The data on the proliferation study was reinstated in the manuscript and the reviewer's proposed text change was made.

- Furthermore, the concluding sentence "Together, these results rule out a model of neurogenesis in PDAC ...branching of existing axon terminals." The conclusion is completely unsubstantiated and also in many ways superfluous. The authors should certainly argue that the latter mechanism seems to be predominant in their KIC model without making such a broad and unproven claim.

Our response: We have made the changes suggested above.

Another problematic statement is on page 14, lines 309-311. I would suggest that the sentence be modified to state, "Altogether, these results demonstrate that the sympathetic nervous system can exert a protection function during the early development and progression of pancreatic cancer in the KIC mouse."

Similarly, on page 15, line 340, would change this to "...on PDAC development in the KIC mouse."

Our response: We have made the changes suggested above.

I think the effect of sympathetic nerves on macrophages is interesting, and worth studying further, but it does appear that the effect of nerves is model, stage and context dependent. While the methods used are probably better than some, the spectrum of models is quite limited, and I would still be cautious with any sweeping conclusions.

Our response: We thank the reviewer for recognizing the interest of our results and the quality of our methodological approaches. We agree that further studies using other experimental and preclinical model systems will be needed to improve our understanding of neuroimmune interactions in PDAC, which are only beginning to be elucidated.